# Occupancy-based Policy Gradient: Estimation, Convergence, and Optimality

**Audrey Huang**
Department of Computer Science
University of Illinois Urbana-Champaign
Champaign, IL 61820
`audreyh5@illinois.edu`

**Nan Jiang**
Department of Computer Science
University of Illinois Urbana-Champaign
Champaign, IL 61820
`nanjiang@illinois.edu`

## Abstract

Occupancy functions play an instrumental role in reinforcement learning (RL) for guiding exploration, handling distribution shift, and optimizing general objectives beyond the expected return. Yet, computationally efficient policy optimization methods that use (only) occupancy functions are virtually non-existent. In this paper, we establish the theoretical foundations of model-free policy gradient (PG) methods that compute the gradient through the occupancy for both online and offline RL, without modeling value functions. Our algorithms reduce gradient estimation to squared-loss regression and are computationally oracle-efficient. We characterize the sample complexities of both local and global convergence, accounting for both finite-sample estimation error and the roles of exploration (online) and data coverage (offline). Occupancy-based PG naturally handles arbitrary offline data distributions, and, with one-line algorithmic changes, can be adapted to optimize any differentiable objective functional.

## 1 Introduction

Value-based methods have been the dominant paradigm in model-free reinforcement learning, with a solid theoretical foundation in large state spaces under function approximation [CJ19; JYWJ20a; ZLKB20; JLM21; XJ21; XFBJK22]. In contrast, a model-free RL paradigm based on their natural counterparts—the *occupancy functions*—remains largely under-investigated. Occupancy functions are densities that describe a policy's state visitation, and play instrumental roles in guiding exploration [HKSVS19; AFK24], handling distribution shift [HM17; NCDL19; CJ22], and optimizing general objectives beyond the expected return [ZBWK20; MDSDBR22]. Despite this, they are seldom modeled directly in learning algorithms and appear only in the analyses, except in conjunction with value functions in marginalized importance sampling [LLTZ18; NDKCLS19; UHJ20; ZH-HJL22; HJ22a]. Recently, [HCJ23] developed algorithms in online and offline RL that model only occupancies via density function classes, spotlighting their roles in handling non-exploratory offline data and in online exploration. However, their focus was on statistical guarantees, and computationally efficient policy optimization for occupancy-based methods remained an open problem.

In answer, we develop model-free policy gradient (PG) algorithms that compute the gradient through occupancy functions, without estimating any values. By leveraging a Bellman-like recursion, we reduce occupancy-based gradient estimation to solving a series of squared-loss minimization problems, which can be done in a computationally oracle-efficient manner. Our analysis captures the effects of gradient estimation error, exploration (in online PG, which is characterized by the initial state distribution), and offline data quality (in offline PG) on the sample and iteration complexity required for local and global convergence. In the online setting, our results complement previous works on the optimality of value-based PG [AKLM21; BR24] and extend past their scope to in-

clude general objectives of occupancy functions, such as entropy maximization for pure exploration and risk-sensitive functionals in safe RL [MDSDBR22]. These objectives generally cannot be optimized using value-based policy gradients because they do not admit value functions or Bellman-like equations with which to estimate them [ZBWK20; HDGP24].

In the offline setting, we handle gradient estimation from fixed datasets of poor coverage, which departs from most existing (value-based) off-policy PG estimators that assume an exploratory dataset [KU20; XYWL21; NZJZW22]. Learning with non-exploratory data is a core consideration in recent offline RL [XCJMA21; ZHHJL22], and gives rise to unique challenges in our setting: occupancies are converted into density ratios for learning purposes, but these ratios become unbounded when the data lacks coverage. [HCJ23] used clipping to handle occupancy estimation under poor coverage, which we show is insufficient for gradient estimation (Prop. 4.2). Instead, a novel smooth-clipping mechanism (Sec. 4.2) is developed to provide statistically robust gradient estimates.

App. A includes a full discussion of related work, and our contributions are organized as follows:

1. **Online PG** (Sec. 3) We propose OCCUPG, an occupancy-based PG algorithm that reduces gradient estimation to squared-loss minimization, based on a recursive Bellman flow-like update for the occupancy gradient. We analyze the sample complexities for both local and global convergence, and, notably, our algorithm and analyses extend straightforwardly to the optimization of general objective functionals.

2. **Offline PG** (Sec. 4) For offline RL we develop and analyze OFF-OCCUPG, which optimizes only the portions of a policy's return that are adequately covered by offline data. Conceptually, our algorithm is based on combining the methods in Sec. 3 with (a smoothed version of) the recursively clipped occupancies from [HCJ23]. As a result, our estimation and convergence guarantees do not require assumptions on data coverage, which relaxes the restrictions of previous works.

## 2 Preliminaries

**Finite-horizon Markov decision process (MDP).** Finite-horizon MDPs are defined by the tuple $\mathcal{M} = (\mathcal{S}, \mathcal{A}, P, R, H, d_0)$, where $\mathcal{S}$ is the state space, $\mathcal{A}$ is the action space, and $H$ is the horizon. We use $[H] = \{0, \dots, H\}$ and when clear from the context, use $\{\Box_h\} = \{\Box_h\}_{h \in [H]}$. For notational compactness we assume that $\mathcal{S} = \dot{\cup}_h \mathcal{S}^h$ is the union of $H$ disjoint sets $\{\mathcal{S}^h\}$, each of which is the set of states reachable at timestep $h$. This is WLOG as we can always augment the state space with $[H]$ at the cost of only $H$ factors [JKALS17; MBFR24].

Since each state can only be visited at a single timestep, we can now define the (non-stationary) transitions as $P : \mathcal{S} \times \mathcal{A} \to \Delta(\mathcal{S})$, and the initial state distribution as $d_0 \in \Delta(\mathcal{S}^0)$. We assume the reward function $R : \mathcal{S} \to [0, 1]$ is bounded on the unit interval and (for simplicity) state-wise deterministic. This sufficiently captures the challenges of our setting since the occupancies are densities over states, and it will be easily seen later that our results generalize to per-state-action rewards. A policy $\pi : \mathcal{S} \to \Delta(\mathcal{A})$ interacting with $\mathcal{M}$ observes trajectories $\{(s_h, a_h, s_{h+1}, r_{h+1})\}_{h=0}^{H-1}$, and has expected return $J(\pi) = \mathbb{E}_\pi[\sum_{h=1}^H R(s_h)]$. At any $(h, s, a)$, its expected return-to-go is encoded in the value function $Q_h^\pi(s, a) = \mathbb{E}_\pi[\sum_{h' > h} R(s_{h'}) | s_h = s, a_h = a]$.

For each $h \in [H]$, a policy's occupancy function $d_h^\pi \in \Delta(\mathcal{S})$ is a p.d.f. describing its state visitation, $d_h^\pi(s) = \mathbb{P}_\pi(s_h = s)$. In combination with the policy, the MDP dynamics dictate the evolution of the occupancy over timesteps. This is encoded in the recursive Bellman flow equation, which mandates that $d_h^\pi = \mathbf{P}^\pi d_{h-1}^\pi$ for all $h \in [H]$. Here, $\mathbf{P}^\pi$ is the Bellman flow operator with $(\mathbf{P}^\pi f)(s') := \sum_{s,a} P(s'|s,a)\pi(a|s)f(s) \in \mathbb{R}^\Box$, for any function $f : \mathcal{S} \to \mathbb{R}^\Box$.

**Policy optimization.** For an objective function $f : \Pi_\Theta \to \mathbb{R}$, the general goal of this work is to find $\operatorname{argmax}_{\pi_\theta \in \Pi_\Theta} f(\pi_\theta)$ over a policy class $\Pi_\Theta = \{\pi_\theta : \theta \in \Theta, \theta \in \mathbb{R}^p, \|\theta\| \leq B\}$, parameterized by a convex and closed parameter class $\Theta$ with dimension p. One example of $f$ is the expected return $J(\pi_\theta)$. Projected gradient ascent (PGA) will be our base algorithm for policy optimization. For a fixed learning rate $\eta$ and iterations $t \in [T]$, it iteratively updates $\theta^{(t+1)} = \operatorname{Proj}_\Theta(\theta^{(t)} + \eta \nabla f(\pi_{\theta^{(t)}}))$. Here, $\nabla f(\pi_\theta) = [\frac{\partial f(\pi_\theta)}{\partial \theta^p}]_{p \in [p]} \in \mathbb{R}^p$ is the gradient with respect to $\theta$, where superscript $p$ indexes the $p$-th entry of a vector. We will assume that the gradient of the policy's log-probability is bounded, as is ubiquitous in the PG literature [LSAB19; AKLM21].

**Assumption 2.1.** For all $\pi_\theta \in \Pi_\Theta$, $\max_{s,a} \|\nabla \log \pi_\theta(a|s)\|_\infty \leq G$.

We will later analyze the convergence rate of our algorithms to stationary points with (approximately) zero gradient, and refer to $\pi^{(t)} = \pi_{\theta^{(t)}}$ for short. For PGA, stationarity will be measured using the standard gradient mapping $\|G^\eta(\pi^{(t)})\|$ with $G^\eta(\pi^{(t)}) := \frac{1}{\eta}(\theta^{(t)} - \theta^{(t+1)})$, the parameter change between iterations [Bec17]. Note that if $\Theta = \mathbb{R}^p$ and no projection is required, then $\|G^\eta(\pi^{(t)})\| = \|\nabla f(\pi^{(t)})\|$ reduces to the gradient magnitude.

**Computational oracles.** As is common in the literature, we analyze computational efficiency in terms of the number of calls to the following oracles, which serve as computational abstractions. We desire a polynomial number of such calls in terms of problem-relevant parameters. Given an i.i.d. dataset $\mathcal{D} = \{(x,y)\}$ and function class $\mathcal{F}$, the maximum likelihood estimation oracle outputs $\text{argmax}_{f \in \mathcal{F}} \mathbb{E}_{\mathcal{D}}[\log f(x)]$. The squared-loss regression oracle finds $\text{argmin}_{f \in \mathcal{F}} \mathbb{E}_{\mathcal{D}}[(f(x) - y)^2]$. Both can be approximated efficiently whenever optimizing over $\mathcal{F}$ is feasible [MHKL20; FR20].

# 3  Online Occupancy-based PG

We now develop our occupancy-based policy optimization algorithm for the online RL setting, where the policy can continuously interact with the environment to gather new trajectories. Our gradient estimation routine is based on a recursive Bellman flow-like equation that can be approximately solved using squared-loss regression, not unlike those used to estimate occupancy functions in FORC [HCJ23] or value functions in FQI [ASM07]. The intuitions established for our online algorithm form the foundation for our later offline methods.

## 3.1  Occupancy-based Policy Gradient

The expected return of a policy $\pi$ can be expressed as the expectation over its occupancy of the per-state rewards, $J(\pi) = \sum_h \sum_{s_h} d_h^\pi(s_h) R(s_h)$. The gradient of $J(\pi)$ then passes through $d^\pi$,

$$\nabla J(\pi) = \sum_h \sum_{s_h} \nabla d_h^\pi(s_h) R(s_h) = \sum_h \mathbb{E}_{s_h \sim d_h^\pi} [\nabla \log d_h^\pi(s_h) R(s_h)].$$

We use the grad-log trick above to write $\nabla J(\pi)$ as an expectation over $d^\pi$, which makes it amenable to estimation from online samples as long as we can calculate $\nabla \log d_h^\pi : \mathcal{S} \to \mathbb{R}^p$. We make the key observation that $\nabla \log d_h^\pi$ can be expressed as a function of $\nabla \log d_{h-1}^\pi$, which involves a time-reversed conditional expectation over the previous timestep's $(s_{h-1}, a_{h-1})$ given the current $s_h$.

**Lemma 3.1.** *For any $\pi$ and $h \in [H]$, $\nabla \log d_h^\pi$ satisfies the recursion*

$$\nabla \log d_h^\pi = \mathbf{E}_{h-1}^\pi \left( \nabla \log \pi + \nabla \log d_{h-1}^\pi \right), \tag{1}$$

*where* $[\mathbf{E}_{h-1}^\pi f](s') := \mathbb{E}_\pi[f(s_{h-1}, a_{h-1})|s_h = s'] = \sum_{s,a} \frac{P(s'|s,a)\pi(a|s)d_{h-1}^\pi(s)}{d_h^\pi(s')} f(s,a)$[1], *for any function* $f : \mathcal{S} \times \mathcal{A} \to \mathbb{R}^p$. *Further, under Asm. 2.1*, $\max_{s,h} \|\nabla \log d_h^\pi(s)\|_\infty \leq hG$.

Eq. (1) is derived by propagating the gradient through the Bellman flow equation, and we can solve it from $h = 1$ to $H$ to compute $\nabla \log d_h^\pi$ (with $\nabla \log d_0^\pi = \mathbf{0}$ by definition). While related observations have been made throughout the rich history of PG literature [CC97; MT01; KU20; XYWL21], the expression in Eq. (1) is adapted to our unique pursuit of modeling $\nabla \log d^\pi$ with general function approximators. In particular, the conditional expectation ($\mathbf{E}^\pi$) immediately hints that $\nabla \log d^\pi$ is amenable to estimation using squared-loss regression, a technique that is well-understood for value functions [SB18] and, more recently, for occupancy functions [HCJ23].

Formally, to solve the dynamic programming equation of Eq. (1) in a computationally efficient manner, we reduce it to minimizing a squared-loss regression problem. Consider the standard (supervised learning) regression setup. The solution to $\text{argmin}_f \mathbb{E}_{(x,y)\sim Q}[(f(x)-y)^2]$ maps $x \mapsto \mathbb{E}_Q[y|x]$, the conditional expectation given $x$ of the target $y$ under the joint $Q$. As a result (see Lem. B.2),

$$\nabla \log d_h^\pi = \text{argmin}_{g:\mathcal{S}\to\mathbb{R}^p} \mathbb{E}_\pi \left[ \left\| g(s_h) - \left( \nabla \log \pi(a_{h-1}|s_{h-1}) + \nabla \log d_{h-1}^\pi(s_{h-1}) \right) \right\|^2 \right]. \tag{2}$$

---

[1]We use the convention $0/0 = 0$ for ratios between two functions.

---

**Algorithm 1** OCCUPG: Online Occupancy-based Policy Gradient

---

**Input:** Samples $n$; iterations $T$; policy class $\Pi_\Theta$; gradient function class $\{\mathcal{G}_h\}$; learning rate $\eta$
1: **for** $t = 0, \ldots, T-1$ **do**
2:     Collect $n$ trajectories with $\pi^{(t)}$. Set $\mathcal{D}_h^{\text{reg}} = \{(s_h, a_h, s_{h+1})\}_{i=1}^n$ for all $h$. Repeat for $\{\mathcal{D}_h^{\text{grad}}\}$.
3:     Initialize $g_0 = \mathbf{0}$.
4:     **for** $h = 1, \ldots, H$ **do**
5:         Let $\mathcal{L}_{h-1}^{(t)}(g_h; g_{h-1}) := \frac{1}{n} \sum_{(s,a,s') \in \mathcal{D}_{h-1}^{\text{reg}}} \left\| g_h(s') - \left(\nabla \log \pi^{(t)}(a|s) + g_{h-1}(s)\right) \right\|^2$. Set

$$\widehat{g}_h^{(t)} = \operatorname{argmin}_{g_h \in \mathcal{G}_h} \mathcal{L}_{h-1}^{(t)}(g_h; \widehat{g}_{h-1}^{(t)}). \tag{3}$$

6:     **end for**
7:     Estimate $\widehat{\nabla} J(\pi^{(t)}) = \frac{1}{n} \sum_{h=1}^{H} \sum_{(s,a,s',r') \in \mathcal{D}_{h-1}^{\text{grad}}} \widehat{g}_h^{(t)}(s') \cdot r'$
8:     Update $\theta^{(t+1)} = \operatorname{Proj}_\Theta \left( \theta^{(t)} + \eta \widehat{\nabla} J(\pi^{(t)}) \right)$.
9: **end for**

---

Here, $g$ is a vector-valued function, and the norm $\|\cdot\|^2$ is equivalent to the sum of $\mathsf{p}$ scalar-valued squared-losses for each parameter dimension. The RHS only requires sampling $(s_{h-1}, a_{h-1}, s_h) \sim \pi$ from online rollouts. Then, given finite samples, we can robustly estimate $\nabla \log d^\pi$ by minimizing an empirical version of Eq. (2) using regression oracles.

## 3.2 Online policy gradient algorithm and analyses

Alg. 1 (OCCUPG) displays our full online occupancy-based PG procedure. For each iteration $t \in [T]$, we first collect two independent datasets: $\{\mathcal{D}_h^{\text{reg}}\}$ for $\nabla \log d^{\pi^{(t)}}$ estimation, and $\{\mathcal{D}_h^{\text{grad}}\}$ for $\nabla J(\pi^{(t)})$ estimation. The former occurs in Line 5, where we recursively solve an empirical version of Eq. (2); the latter is computed in Line 7, then used to update the policy (Line 8).

**Gradient estimation guarantee.** In the following, we establish that our regression-based estimation procedure produces accurate estimates of $\nabla J(\pi)$. Our guarantee holds under the requirement that the gradient function classes $\{\mathcal{G}_h\}$ can express the population gradient update (Lem. 3.1) for any target function. It is analogous to the Bellman completeness assumption that is required for regression-based value or occupancy function estimation [CJ19; HCJ23].

**Assumption 3.1** (Gradient function class completeness)**.** For all $h \in [H]$, $\sup_{g \in \mathcal{G}_h, s \in \mathcal{S}} \|g_h(s)\|_\infty \leq hG$. Further, for all $\pi \in \Pi_\Theta$, we have $\mathbf{E}_{h-1}^\pi (\nabla \log \pi + g_{h-1}) \in \mathcal{G}_h$, for all $g_{h-1} \in \mathcal{G}_{h-1}$.

Next, since we allow $\mathcal{G}$ to be a continuous function class, our sample complexity bound for gradient estimation is expressed in terms of its pseudodimension $:= \mathsf{d}_\mathcal{G}$ (Def. F.1). Examples of $\mathcal{G}$ parameterizations and their $\mathsf{d}_\mathcal{G}$ are discussed in Rem. 3.1 below. Finally, Thm. 3.1 shows that OCCUPG produces accurate gradient estimates given the following polynomial sample size.

**Theorem 3.1.** *Fix $\delta \in (0,1)$ and $\pi \in \Pi_\Theta$. Under Asm. 2.1 and Asm. 3.1, we have that w.p. $\geq 1 - \delta$, $\|\nabla J(\pi) - \widehat{\nabla} J(\pi)\| \leq \varepsilon$ when $n = \widetilde{O}\left( \frac{\mathsf{pd}_\mathcal{G} H^6 G^2 \log(1/\delta)}{\varepsilon^2} \right)$.*

*Remark* 3.1. Lastly, we provide examples of $\mathcal{G}$ for Asm. 3.1 in representative MDP structures. Low-rank MDPs (Def. B.1) are a well-studied setting where the transition function admits a low-rank decomposition into two features of rank $k$, i.e., there exists $\phi : \mathcal{S} \times \mathcal{A} \to \mathbb{R}^k$ and $\mu : \mathcal{S} \to \mathbb{R}^k$ such that $P(s'|s,a) = \langle \phi(s,a), \mu(s') \rangle$ [JYWJ20b]. Tabular MDPs are a special case with one-hot features. Due to the bilinear transitions, both the occupancy and its gradient are linear functions of $\mu$, i.e., $d^\pi = \mu(s)^\top \psi$ and $\nabla d^\pi(s) = \mu(s)^\top \Psi$ for some $\Psi \in \mathbb{R}^{k \times p}, \psi \in \mathbb{R}^k$, and all $s \in \mathcal{S}$. When $\mu$ is known, we can set $\mathcal{G}_h$ to be a linear-over-linear function class $\mathcal{G}_h = \left\{ g_h(s) = \frac{\mu(s)^\top \Psi}{\mu(s)^\top \psi} : \Psi \in \mathbb{R}^{k \times \mathsf{p}}, \psi \in \mathbb{R}^k, \max_s \|g_h(s)\|_\infty \leq hG \right\}$, which has $\mathsf{d}_\mathcal{G} = k\mathsf{p}$ (Prop. B.1).

**Stationary convergence.** Next, we analyze the convergence rate of OCCUPG to a stationary policy, i.e., one that has near-zero gradient. Note that, in general, stationary policies are not necessarily optimal as the objective function is non-convex. As is standard in the literature, we will assume that the objective has a smooth gradient [LSAB19; AKLM21].

**Assumption 3.2** ($\beta$-smooth objective). *For a function $f : \Pi_\Theta \to \mathbb{R}$, there exists $\beta > 0$ such that $\|\nabla f(\pi_\theta) - \nabla f(\pi_{\theta'})\|_2 \leq \beta \|\theta - \theta'\|_2$ for all $\theta, \theta' \in \Theta$.*

Cor. 3.1 shows that, in expectation, OCCUPG with $T = O(\beta H / \varepsilon)$ iterations outputs a $\varepsilon$-stationary point, as measured by $\|G^\eta(\pi^{(t)})\| = \frac{1}{\eta}\|\theta^{(t)} - \theta^{(t-1)}\|$. The proof relies on Thm. 3.1, i.e., with enough samples the statistical noise of the gradient estimates are sufficiently small to enable convergence.

**Corollary 3.1.** *Under Asm. 2.1, Asm. 3.1, and Asm. 3.2, the iterates of OCCUPG with $T = O(\beta H \varepsilon^{-1})$ and $n = \widetilde{O}(\mathsf{pd}_\mathcal{G} H^6 G^2 \log(T/\delta)\varepsilon^{-1})$ satisfy $\frac{1}{T}\sum_{t=1}^{T} \mathbb{E}[\|G^\eta(\pi^{(t)})\|^2] \leq \varepsilon$.*

**Computational efficiency.** OCCUPG is not only statistically efficient but computationally oracle-efficient as well, since it reduces to a series of squared-loss minimization problems. In each iteration, it makes $H$ calls to a regression oracle to compute the occupancy gradient (Line 5). Then to converge to a $\varepsilon$-stationary point, from Cor. 3.1 we require a total of $O(\beta H^2 / \varepsilon)$ such calls.

**Optimality.** Lastly, we analyze when the policies recovered by OCCUPG are also approximately optimal. The key inequality is an upper bound on the suboptimality of any policy in terms of its gradient magnitude (or stationarity), and a *coverage coefficient* $\mathcal{C}^{\pi^*}$ with respect to the optimal policy.

**Lemma 3.2.** *For any $\pi$ and $\pi'$, define $B^\pi(\pi') := \sum_{h,s,a} d_h^\pi(s)\pi'(a|s)Q_h^\pi(s,a)$. Suppose $\forall \pi \in \Pi_\Theta$,*

1. *(Policy completeness) There exists $\pi^+ \in \Pi_\Theta$ such that $\pi^+ \in \arg\max_{\pi'} B^\pi(\pi')$.*
2. *(Gradient domination) $\max_{\pi' \in \Pi_\Theta} B^\pi(\pi') - B^\pi(\pi) \leq m \max_{\theta' \in \Theta} \langle \nabla B^\pi(\pi), \theta' - \theta \rangle$*

*Given $\nu \in \Delta(\mathcal{S})$, define the coverage coefficient $\mathcal{C}^{\pi^*} := \left\| \sum_h d_h^{\pi^*} / \nu \right\|_\infty$ for $\pi^* = \arg\max_\pi J(\pi)$. Then for any $\pi_\theta \in \Pi_\Theta$,*

$$J(\pi^*) - J(\pi_\theta) \leq m\, \mathcal{C}^{\pi^*} \max_{\theta' \in \Pi_\Theta} \langle \nabla J_\nu(\pi_\theta), \theta' - \theta \rangle, \tag{4}$$

*where $J_\nu(\pi) := \mathbb{E}_{s_0 \sim \nu, \pi}[\sum_h r_h]$ is the expected return of $\pi$ in $\mathcal{M}$ with initial state distribution $\nu$.*

The lemma preconditions are identical to those required for value-based analysis [BR24]. $B^\pi(\pi')$ is a one-step improvement objective with respect to the occupancies and value functions of $\pi$, and we require (1) the policy class to be expressive enough that it contains any maximizer; and (2) the one-step objective to itself have optimality gap upper-bounded by the one-step policy gradient magnitude, for which the constant $m$ is determined wholly by the policy parameterization. For example, the tabular policy $\pi_\theta(a|s) = \theta_{sa}$ has $m = 1$ [AKLM21].

The coverage coefficient $\mathcal{C}^{\pi^*}$ is the finite-horizon counterpart to the infinite-horizon "exploratory initial distribution" salient to the analysis of [AKLM21] and [BR24] (which lists developing it as future work). In RL, a small gradient magnitude alone does not guarantee optimality, as it can also occur when the policy rarely visits rewarding states. The coverage coefficient quantifies both how policy performance can suffer from insufficient exploration, as well as how exploratory initializations mitigates this problem. Finally, combining Lem. 3.2 with the stationary convergence result in Cor. 3.1 shows that, on average, the best-iterate of OCCUPG is near-optimal.

**Corollary 3.2.** *Under the preconditions of Lem. 3.2 and Cor. 3.1, running OCCUPG[2] with initial distribution $\nu$ satisfies $\mathbb{E}[\min_t J(\pi^*) - J(\pi^{(t)})] \leq \varepsilon$ when $T = \widetilde{O}\left( \frac{\beta B^2 (\mathcal{C}^{\pi^*})^2 m^2 H^2}{\varepsilon^2} \right)$ and $n = \widetilde{O}\left( \frac{B^2 (\mathcal{C}^{\pi^*})^2 m^2 \mathsf{pd}_\mathcal{G} H^6 G^2 \log(T)}{\varepsilon^2} \right)$.*

## 3.3 Optimization of general functionals

One standout feature of OCCUPG is that it can, *with a one-line change*, be adapted for policy optimization of any (differentiable) objective function involving occupancies. We work with $J_F(\pi) = \sum_h F_h(d_h^\pi)$ as a representative formula, where $F_h : \Delta(\mathcal{S}) \to \mathbb{R}$ is a general functional. Such objectives often evade value-based PG optimization because they do not admit value functions or Bellman-like recursions with which to compute them. Examples include entropy maximization

---

[2]This means that trajectories are generated by first sampling the initial state $s_1 \sim \nu$, then rolling out the policy according to the true MDP's dynamics.

where $F_h(d) = -\langle d, \log d \rangle$; imitation learning where $F_h(d) = -\|d - d_h^{\pi_E}\|_2^2$ for an expert policy $\pi_E$; and the expected return with $F_h(d) = \langle d, R \rangle$ [MDSDBR22].

The policy gradient is then $\nabla J_F(\pi) = \sum_h \mathbb{E}_{s \sim d_h^\pi} \left[ \frac{\partial F_h(d)}{\partial d(s)} |_{d=d_h^\pi} \nabla \log d_h^\pi(s) \right]$. Implementation-wise, we need only change Line 7 in OccuPG to accommodate the new gradient formula, to $\widehat{\nabla} J_F(\pi) = \frac{1}{n} \sum_h \sum_{s \in \mathcal{D}_h} \widehat{g}_h^\pi(s) \frac{\partial F_h(d)}{\partial d(s)} |_{d=\widehat{d}_h^\pi}$. The partial derivative of $F_h$ is evaluated with a plug-in occupancy estimate $\widehat{d}^\pi$ that can be obtained using maximum likelihood estimation (App. D). Notably, the occupancy gradient estimation module for $\widehat{g}_h^\pi \approx \nabla \log d_h^\pi$ (Line 5) is reused verbatim. Given their resemblance to those in Sec. 3.2, the full algorithm and analyses are deferred to App. B.5.

## 4 Offline Occupancy-based PG

In this section, we develop an algorithm for occupancy-based policy optimization in the offline setting, where only fixed datasets are available for learning. A direct modification of OccuPG, e.g., by converting occupancies to density ratios over the offline data distribution, will fail unless the data covers *all possible policies*, otherwise the density ratio may be unbounded. In-line with recent state-of-the-art offline RL algorithms, our goal is to establish an offline PG algorithm that adapts to and retains meaningful guarantees under arbitrary offline datasets, for which our key consideration is establishing an offline gradient estimation method. We begin by defining these offline datasets.

**Definition 4.1.** The offline dataset is $\mathcal{D} = \{\mathcal{D}_h\}$, where $\mathcal{D}_h = \{(s_h, a_h, s_{h+1}, r_{h+1}))\}_{i=1}^n$ is generated i.i.d. as $s_h \sim d_h^D$ for some $d_h^D \in \Delta(\mathcal{S})$ and $a_h \sim \pi_h^D(\cdot|s_h)$ in $\mathcal{M}$, for a known behavior policy $\pi_h^D$. The marginal next-state distribution in $\mathcal{D}_h$ is denoted as $d_h^{D,\dagger}(s_{h+1})$.

Def. 4.1 is more general than the typical i.i.d. trajectory setting [KU20; NZJZW22], where $d_h^D = d_{h-1}^{D,\dagger}$. Crucially, unlike previous works that require lower-bounded $d^D$ or all-policy coverage [KU20; XYWL21; NZJZW22], we will make no assumptions about the quality of $\mathcal{D}$ with respect to $\Pi_\Theta$.

**Additional notation.** For short, we say $\mathbb{E}_{\mathcal{D}_h}[\cdot] \equiv \mathbb{E}_{(s_h, a_h, s_{h+1}, r_{h+1}) \sim \mathcal{D}_h}[\cdot]$, and use $(s, a, s', r') \sim \mathcal{D}_h$ when clear from the context. For any $g : \mathcal{S} \times \mathcal{A} \to \mathbb{R}^p$ and reweighting function $\rho : H \times \mathcal{S} \times \mathcal{A} \to \mathbb{R}_+$, we define an *offline reweighted* analog to $\mathbf{E}_h^\pi$ (Lem. 3.1) for all $h \in [H]$ to be

$$[\mathbf{E}_h^{D,\rho} g](s') := \mathbb{E}_{(s,a,s') \sim \mathcal{D}_h \cdot \rho_h}[g(s,a)|s'] = \sum_{s,a} \frac{[\mathcal{D}_h \cdot \rho_h](s,a,s')}{\sum_{s,a}[\mathcal{D}_h \cdot \rho_h](s,a,s')} g(s,a). \tag{5}$$

The (time-reversed) conditional expectation is taken over $[\mathcal{D}_h \cdot \rho_h](s,a,s') := P(s'|s,a)d_h^D(s)\pi_h^D(a|s)\rho_h(s,a)$, the joint offline distribution re-weighted by $\rho_h$. While this may not be a valid density, its induced conditional distribution on $(s,a|s')$ always is, i.e., $\sum_{s,a} \frac{[\mathcal{D}_h \cdot \rho_h](s,a,s')}{\sum_{s,a}[\mathcal{D}_h \cdot \rho_h](s,a,s')} = 1$. As an example, for a given $\pi$ we have $\mathbf{E}_h^{D,\rho} = \mathbf{E}_h^\pi$ when $\rho_h(s,a) = \frac{d_h^\pi(s)\pi(a|s)}{d_h^D(s)\pi_h^D(a|s)}$ is the policy's density ratio and is well-defined.

### 4.1 Offline density-based policy gradient

A policy's occupancy $d^\pi$ may not be covered by arbitrary offline data (Def. 4.1), so neither its expected return $J(\pi) = \sum_h \langle d_h^\pi, R \rangle$ nor its gradient $\nabla J(\pi)$ will be estimatable from $\mathcal{D}$. As a result, there is no hope of recovering $\arg\max_{\pi \in \Pi_\Theta} J(\pi)$. Our solution is to instead *maximize return only on areas of the state space that are sufficiently covered by offline data*, which is captured exactly by the recursively clipped occupancy $\bar{d}^\pi$ from [HCJ23]. It clamps the policy occupancy to preset multiples $C_h^{\mathbf{s}}, C_h^{\mathbf{a}}$ of the offline data distribution, thereby representing only the "sufficiently covered" portion.

**Definition 4.2** (Recursively clipped occupancy). Let $(\Box \wedge \Box) := \min\{\Box, \Box\}$. Given clipping constants $\{C_h^{\mathbf{s}}, C_h^{\mathbf{a}}\} \geq 1$, define the clipped policy to be $\bar{\pi}_h = (\pi \wedge C_h^{\mathbf{a}} \pi_h^D)$, and recursively define

$$\bar{d}_h^\pi = \mathbf{P}^{\bar{\pi}_{h-1}} \left( \bar{d}_{h-1}^\pi \wedge C_{h-1}^{\mathbf{s}} d_{h-1}^D \right), \forall h \in [H]. \tag{6}$$

Eq. (6) resembles the Bellman flow equation with clipped policy $\bar{\pi}$, and acts on the previous-timestep $\bar{d}_{h-1}^\pi$ clipped to at most $C_{h-1}^{\mathbf{s}} d_{h-1}^D$. Above this threshold the occupancy is considered to be insufficiently covered for estimation, and $C^{\mathbf{s}}$ strikes a bias-variance tradeoff between the amount of

clipped mass vs. distribution shift. The clipped occupancy's density ratio is always well-defined and bounded as $\bar{d}_h^\pi / d_{h-1}^{D,\dagger} \leq C_{h-1}^{\mathbf{s}} C_{h-1}^{\mathbf{a}}$, and we use it to define our (now learnable) offline objective,

$$\bar{J}(\pi) = \sum_h \sum_{s_h} \bar{d}_h^\pi(s_h) R(s_h) = \sum_h \mathbb{E}_{\mathcal{D}_{h-1}} \left[ \frac{\bar{d}_h^\pi(s_h)}{d_{h-1}^{D,\dagger}(s_h)} R(s_h) \right] .$$

For any "fully covered" policy with $d_h^\pi \leq C_{h-1}^{\mathbf{s}} C_{h-1}^{\mathbf{a}} d_{h-1}^{D,\dagger}$ for all $h \in [H]$, we have $\bar{d}^\pi = d^\pi$ and $\bar{J}(\pi) = J(\pi)$. In this sense, $\arg\max_\pi \bar{J}(\pi)$ will be at least as good as the best policy fully covered by offline data. Next, define the density ratio be $\bar{w}_h^\pi := \bar{d}_h^\pi / d_{h-1}^{D,\dagger}$. The gradient of $\bar{J}(\pi)$ is

$$\nabla \bar{J}(\pi) = \sum_h \mathbb{E}_{\mathcal{D}_{h-1}} \left[ \bar{w}_h^\pi(s_h) R(s_h) \nabla \log \bar{d}_h^\pi(s_h) \right] .$$

To calculate this gradient we must compute both $\bar{w}^\pi$ and $\nabla \log \bar{d}^\pi$; for the former, [HCJ23] provides a method that we will later call as a subroutine. Our focus is on computing $\nabla \log \bar{d}_h^\pi$, which is enabled by the following recursive equation, which is an offline analog of Lem. 3.1.

**Lemma 4.1.** *For any $\pi$ and all $h \in [H]$, define $\bar{\rho}_h^\pi(s,a) := \frac{(\bar{d}_h^\pi(s) \wedge C_h^{\mathbf{s}} d_h^D(s))}{d_h^D(s)} \frac{\bar{\pi}_h(a|s)}{\pi_h^D(a|s)}$. Then*

$$\nabla \log \bar{d}_h^\pi = \mathbf{E}_{h-1}^{D,\bar{\rho}^\pi} \left( \nabla \log \pi \odot \mathbf{1}[\pi \leq C_h^{\mathbf{a}} \pi_{h-1}^D] + \nabla \log \bar{d}_{h-1}^\pi \odot \mathbf{1}[\bar{d}_{h-1}^\pi \leq C_h^{\mathbf{s}} d_{h-1}^D] \right), \quad (7)$$

*where $\mathbf{E}_{h-1}^{D,\bar{\rho}^\pi}$ is from Eq. (5), and $[M \odot v](\cdot) := v(\cdot) M(\cdot) \in \mathbb{R}^{\mathsf{p}}$ for $M : \square \to \mathsf{p}$ and $v : \square \to \mathbb{R}$.*

Lem. 4.1 is derived from applying the chain rule to Def. 4.2, and the clipped occupancies play an instrumental role in handling insufficient offline coverage. Notably, the indicator function zeroes-out both the gradients $\nabla \log \pi$ and $\nabla \log \bar{d}_{h-1}^\pi$ where they are insufficiently covered, e.g., $\bar{d}_{h-1}^\pi(s) > C_{h-1}^{\mathbf{s}} d_{h-1}^D(s)$. Further, under full offline coverage we recover Lem. 3.1 and $\nabla \log \bar{d}^\pi = \nabla \log d^\pi$.

Because the rewards are nonnegative, $\nabla \log \bar{d}^\pi$ induces a *pessimistic policy gradient* that shifts policies away from out-of-distribution actions, even if they generate high return. This is seen more clearly in Prop. 4.1, that rearranges the resulting expression for $\nabla \bar{J}(\pi)$ into a value-based form:

**Proposition 4.1.** *We can equivalently write*

$$\nabla \bar{J}(\pi) = \sum_h \mathbb{E}_{\mathcal{D}_h} [\bar{\rho}_h^\pi(s,a) \nabla \log \pi_h(a|s) \bar{Q}_h^\pi(s,a)],$$

*where $\bar{Q}^\pi$ is a pessimistic value function that obeys the Bellman-like recursion $\bar{Q}_h^\pi(s,a) = \mathbf{1}[\pi \leq C_h^{\mathbf{a}} \pi_h^D](a|s) \sum_{s'} P(s'|s,a) \left( R(s') + \mathbf{1}[\bar{d}_{h+1}^\pi \leq C_{h+1}^{\mathbf{s}} d_{h+1}^D](s') \bar{Q}_{h+1}^\pi(s', \bar{\pi}_{h+1}) \right)$.*

In $\bar{Q}^\pi$, future returns are zeroed out at states and actions that exceed the threshold of data coverage, due to indicators functions that are inherited from $\nabla \log \bar{d}^\pi$. Prop. 4.1 can be seen as a pessimistic offline analog to the classical PG theorem $\nabla J(\pi) = \sum_h \mathbb{E}_{s,a \sim d_h^\pi} [\nabla \log \pi(a|s) Q_h^\pi(s,a)]$ [SMSM99], entirely induced by the definition of the clipped occupancy.

**Non-robustness of $\nabla \log \bar{d}^\pi$ estimation to plug-in densities.** With finite samples, however, it turns out that consistent estimates of $\nabla \log \bar{d}_h^\pi$ in Eq. (6) cannot be computed. To make this argument, we first outline the high-level gradient estimation procedure for a fixed policy:

- Estimate occupancies $\{\widehat{d}_h^\pi\}$ and $\{\widehat{d}_h^D\}$
- Compute $\widehat{\nabla} \log \bar{d}_h^\pi$ using Eq. (7) with plug-in indicator function estimate $\mathbf{1}[\widehat{d}_{h-1}^\pi \leq C_h^{\mathbf{s}} \widehat{d}_h^D]$

The problem arises in step two, as $\mathbf{1}[\cdot]$ is a stepwise function and not smooth. Even if $\widehat{d}^\pi$ is vanishingly close to $d^\pi$, the gradient calculated from plug-in occupancy estimates can have constant error.

**Proposition 4.2.** *There exists an MDP and policy $\pi$ such that, for any $\varepsilon > 0$, $\max_{h,s} \| \nabla \log d_h^\pi(s) - \widehat{\nabla} \log \bar{d}_h^\pi(s) \| = O(1)$ when $\| \bar{d}_h^\pi - \widehat{d}_h^\pi \|_1 \leq \varepsilon$ and $\| \widehat{d}_h^D - d_h^D \|_1 \leq \varepsilon$ for all $h$.*

### 4.2 Smooth clipping

To resolve this issue, we will use a "smooth-clipping" function $\sigma(x,c)$ to approximate the "hard"-clipping $(x \wedge c)$ in Eq. (6), whose non-smooth gradient was the source of our estimation problems. Figure 1 plots 1-D examples of $\sigma(x,c)$ against $(x \wedge c)$ as reference (dashed), and Asm. 4.1 describes the properties of $\sigma$ that enable our later estimation and convergence guarantees.

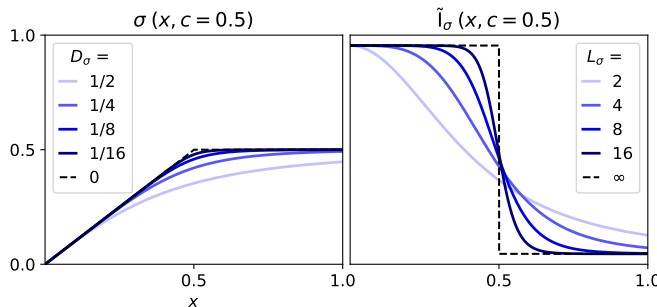

Figure 1: We plot $\sigma(x, c)$ from Prop. 4.3 for different $b$, that trade-off between clipping approximation error and smoothness ($D_\sigma \propto 1/L_\sigma$).

**Assumption 4.1.** Assume that $\sigma$ satisfies $\forall x, x', c, c' \in \text{dom}(\sigma)$,

1. (Approximate clipping) $\exists D_\sigma \geq 0$ such that $0 \leq (x \wedge c) - \sigma(x, c) \leq D_\sigma(x \wedge c)$.

2. (Monotonicity) $\sigma(x', c) \leq \sigma(x, c)$ if $x' \leq x$; $\sigma(x, c') \leq \sigma(x, c)$ if $c' \leq c$; and vice versa.

3. (Smooth gradient) Define the smoothed indicator $\tilde{\mathbf{1}}(x, c) := x\, \partial_x \log \sigma(x, c)$, where $\partial_x$ is the partial derivative w.r.t. $x$. Then $\tilde{\mathbf{1}}(x, c) \in [0, 1]$ and $\exists L_\sigma \geq 0$ s.t. $\forall x, x', c, c' \in \text{dom}(\sigma)$,
$$c|\tilde{\mathbf{1}}(x, c) - \tilde{\mathbf{1}}(x', c)| \leq L_\sigma|x - x'|, \text{ and } x|\tilde{\mathbf{1}}(x, c) - \tilde{\mathbf{1}}(x, c')| \leq L_\sigma|c - c'|.$$

Note that $\sigma(x, c) = (x \wedge c)$ is a special case with $\tilde{\mathbf{1}}(x, c) = \mathbf{1}[x \leq c]$, thus $D_\sigma = 0$ and $L_\sigma = \infty$. The following choice of $\sigma$, which is plotted in Fig. 1, fulfills Asm. 4.1.

**Proposition 4.3.** *For any $b > 1$, $\sigma(x, c) = \left(x^{-b} + c^{-b}\right)^{-1/b}$ has $L_\sigma = b$ and $D_\sigma = 1/b$.*

Next, we define the smooth-clipped occupancy function $\widetilde{d}_h^\pi$, which is no larger than $\bar{d}_h^\pi$.

**Definition 4.3** (Recursively smooth-clipped occupancy)**.** For smooth-clipping function $\sigma$ satisfying Asm. 4.1 and clipping constants $\{C_h^{\mathbf{s}}, C_h^{\mathbf{a}}\}$, define $\widetilde{\pi}_h := \sigma\left(\pi, C_h^{\mathbf{a}}\pi_h^D\right)$, and inductively set
$$\widetilde{d}_h^\pi = \mathbf{P}^{\widetilde{\pi}_{h-1}}\left(\sigma\left(\widetilde{d}_{h-1}^\pi, C_{h-1}^{\mathbf{s}}d_{h-1}^D\right)\right), \forall h \in [H]. \tag{8}$$

Then letting $\widetilde{w}_h^\pi := \widetilde{d}_h^\pi / d_{h-1}^{D,\dagger}$, our new objective is $\widetilde{J}(\pi) = \sum_h \mathbb{E}_{\mathcal{D}_{h-1}}[\widetilde{w}_h^\pi(s_h)R(s_h)]$ with gradient $\nabla\widetilde{J}(\pi) = \sum_h \mathbb{E}_{\mathcal{D}_{h-1}}[\widetilde{w}_h^\pi(s_h)R(s_h)\nabla\log\widetilde{d}_h^\pi(s_h)]$, where $\nabla\log\widetilde{d}_h^\pi$ obeys the following recursion.

**Lemma 4.2.** *For $\sigma$ satisfying Asm. 4.1, recall $\tilde{\mathbf{1}}(x, c) := x\, \partial_x \log \sigma(x, c)$. Then for all $h \in [H]$,*
$$\nabla\log\widetilde{d}_h^\pi = \mathbf{E}_{h-1}^{D,\widetilde{\rho}^\pi}\left(\nabla\log\pi \odot \tilde{\mathbf{1}}\left(\pi, C_{h-1}^{\mathbf{a}}\pi_{h-1}^D\right) + \nabla\log\widetilde{d}_{h-1}^\pi \odot \tilde{\mathbf{1}}\left(\widetilde{d}_{h-1}^\pi, C_{h-1}^{\mathbf{s}}d_{h-1}^D\right)\right), \tag{9}$$

*where* $\widetilde{\rho}_{h-1}^\pi(s, a) := \frac{\sigma\left(\widetilde{d}_{h-1}^\pi(s), C_{h-1}^{\mathbf{s}}d_{h-1}^D(s)\right)}{d_{h-1}^D(s)} \frac{\widetilde{\pi}_{h-1}(a|s)}{\pi_{h-1}^D(a|s)}$ *and* $\mathbf{E}_{h-1}^{D,\widetilde{\rho}^\pi}$ *is defined in Eq. (5). Further, under Asm. 2.1, $\max_{s,h} \|\nabla\log\widetilde{d}_h^\pi(s)\|_\infty \leq hG$.*

Eq. (9) replaces the (non-smooth) indicator function in $\nabla\log\bar{d}^\pi$ (Lem. 4.1) with its smooth approximation $\tilde{\mathbf{1}}$, which, as we will show shortly, enables robust gradient estimation with plug-in occupancy estimates. As before, we can reduce it to squared-loss regression (Eq. (11)). Further, by optimizing $\widetilde{J}(\pi)$, we also approximately maximize our target objective $\bar{J}(\pi)$, with bias proportional to $D_\sigma$.

**Proposition 4.4.** *Under Asm. 4.1, $0 \leq \max_{\pi \in \Pi_\Theta} \bar{J}(\pi) - \max_{\pi \in \Pi_\Theta} \widetilde{J}(\pi) \lesssim H^2 D_\sigma$.*

### 4.3 Offline smooth-clipped gradient estimation

Alg. 2 describes the offline PG algorithm for optimizing $\widetilde{J}(\pi)$. To reduce clutter, we have used $\nabla\log\widetilde{\pi}_h := \nabla\log\pi \odot \tilde{\mathbf{1}}\left(\pi, C_h^{\mathbf{a}}\pi_h^D\right)$. First, OFF-OCCUPG estimates $d_{h-1}^D$ using MLE (details in App. D due to space constraints). Then, for each iteration $t$, it estimates the smooth-clipped occupancy $\widetilde{d}_h^{\pi^{(t)}}$ using FORC (adapted from [HCJ23], see App. E). This is plugged into a squared-loss regression problem approximating Eq. (9) to learn $\nabla\log\widetilde{d}_h^{(t)}$ (lines 8 to 10), then estimate $\nabla\widetilde{J}(\pi^{(t)})$ (line 12).

---
**Algorithm 2** OFF-OCCUPG: Offline Occupancy-based Policy Gradient
---
**Input:** data $\mathcal{D}$; iters $T$; learning rate $\eta$; function classes $\Pi_\Theta, \mathcal{F}, \mathcal{W}, \mathcal{G}$; clipping constants $\{C_h^{\mathbf{s}} \, C_h^{\mathbf{a}}\}$

1: Split $\mathcal{D}$ equally into $\mathcal{D}^{\mathrm{mle}}, \mathcal{D}^{\mathrm{FORC}}, \mathcal{D}^{\mathrm{reg}}, \mathcal{D}^{\mathrm{grad}}$, each with $n$ samples.

2: Estimate $\{\widehat{d}_h^D, \widehat{d}_h^{D,\dagger}\} \leftarrow \mathrm{MLE}\left(\mathcal{D}^{\mathrm{mle}}, \mathcal{F}\right)$            `// Alg. 4`

3: **for** $t = 0, \ldots, T-1$ **do**

4:     Estimate $\{\widehat{w}_h^{(t)}\} \leftarrow \mathrm{FORC}\left(\pi^{(t)}, \mathcal{D}^{\mathrm{FORC}}, \mathcal{W}, \{\widehat{d}_h^D, \widehat{d}_h^{D,\dagger}\}\right)$      `// Alg. 5`

5:     Set occupancy estimate $\widehat{d}_h^{(t)} = \widehat{w}_h^{(t)} \, \widehat{d}_{h-1}^{D,\dagger}$ for all $h \in [H]$.

6:     Initialize $\widehat{g}_0^{(t)} = \mathbf{0}$.

7:     **for** $h = 1, \ldots, H$ **do**

8:        Set density ratio $\widehat{\rho}_{h-1}^{(t)} = \frac{\widetilde{\pi}_{h-1}^{(t)}}{\pi_{h-1}^D} \frac{\sigma\left(\widehat{d}_{h-1}^{(t)}, C_{h-1}^{\mathbf{s}} \widehat{d}_{h-1}^D\right)}{\widehat{d}_{h-1}^D}$.

9:        Set gradient regression target $\widehat{y}_{h-1}^{(t)} = \widehat{g}_{h-1}^{(t)} \odot \widetilde{\mathbf{1}}\left(\widehat{d}_{h-1}^{(t)}, C_{h-1}^{\mathbf{s}} \widehat{d}_{h-1}^D\right)$.

10:        Let $\widetilde{\mathcal{L}}_{h-1}^{(t)}(g; y, \rho) := \frac{1}{n} \sum_{(s,a,s') \in \mathcal{D}_{h-1}^{\mathrm{reg}}} \rho(s,a) \|g(s') - (\nabla \log \widetilde{\pi}_{h-1}^{(t)}(a|s) + y(s))\|^2$. Solve

$$\widehat{g}_h^{(t)} = \mathrm{argmin}_{g_h \in \mathcal{G}_h} \ \widetilde{\mathcal{L}}_{h-1}^{(t)}(g_h; \widehat{y}_{h-1}^{(t)}, \widehat{\rho}_{h-1}^{(t)}) \tag{10}$$

11:     **end for**

12:     Set $\widehat{\nabla} \widetilde{J}(\pi^{(t)}) = \frac{1}{n} \sum_{h=1}^{H} \sum_{(s,a,s',r') \in \mathcal{D}_h^{\mathrm{grad}}} \widehat{w}_h^{(t)}(s') \cdot \widehat{g}_h^{(t)}(s') \cdot r'$

13:     Update $\theta^{(t+1)} = \mathrm{Proj}_\theta(\theta^{(t)} + \eta \widehat{\nabla} \widetilde{J}(\pi^{(t)}))$.

14: **end for**

---

Before stating the estimation guarantee for $\nabla \widetilde{J}(\pi)$, we first introduce the required assumptions. For simplicity, we assume that the function classes used in MLE and FORC are finite, and defer their guarantees to the respective appendices, as they have been well-established in previous papers [AKKS20; HCJ23]. We focus on discussing Asm. 4.2 for the offline gradient function class, which requires a stronger level of expressiveness. Since the regression target in OFF-OCCUPG involves plug-in occupancy estimates, the completeness condition naturally requires $\mathcal{G}_h$ to express the gradient update in Lem. 4.2 for all possible targets composed of functions from $\mathcal{F}_{h-1}, \mathcal{W}_{h-1}, \mathcal{G}_{h-1}$. As a result, Asm. 4.2 is generally stronger than Asm. 3.1 for OCCUPG.

**Assumption 4.2.** For all $h$, $\sup_{g \in \mathcal{G}_h} \|g_h\|_\infty \leq hG$; and for all $(\pi, g, f, f', w) \in \Pi_\Theta \times \mathcal{G}_{h-1} \times \mathcal{F}_h \times \mathcal{F}_{h-1} \times \mathcal{W}_{h-1}$, we have $\mathbf{E}_{h-1}^{D,\rho}(\nabla \log \widetilde{\pi}_{h-1} + g \odot \widetilde{\mathbf{1}}(wf', C_{h-1}^{\mathbf{s}}f)) \in \mathcal{G}_h$, where $\rho = \frac{\sigma(wf', C_h^{\mathbf{s}}f)}{f} \frac{\widetilde{\pi}_{h-1}}{\pi_{h-1}^D}$.

When the underlying MDP has favorable structure, however, we can expect that $d_\mathcal{G}$ is not much larger than was required for OCCUPG. This is indeed the case in low-rank MDPs, where the $\mathcal{G}$ defined in Rem. 3.1 also satisfies Asm. 4.2 (proof in Prop. C.1). Due to the bilinear transition structure, the offline gradient update (Lem. 4.2) applied to any target remains a linear-over-linear function.

The guarantees for the MLE (Alg. 4) and weight estimation (Alg. 5) subroutines require Asm. D.1 and Asm. E.1, respectively, which are included in the preconditions of the main result below. Briefly, Asm. D.1 requires $\mathcal{F}$ to realize the true data distributions $d_h^D$ and $d_h^{D,\dagger}$, which is standard in supervised learning. Asm. E.1 requires $\mathcal{W}$ to be closed under the Bellman flow operator, and can be viewed as a 1-dimensional version of Asm. 4.2 where $\rho = 1$. In this sense both assumptions are weaker requirements on expressivity than that of the gradient class in Asm. 4.2, and more detailed discussions are left to App. D and App. E.

Having established its preconditions, we now present our main estimation guarantee for OFF-OCCUPG, which pays additional factors for the coverage of offline data ($\sum_h C_h^{\mathbf{s}} C_h^{\mathbf{a}}$) and the smoothness of $\sigma$.

**Theorem 4.1.** *Suppose* $\widetilde{J}(\cdot)$ *satisfies* [Asm. 3.2](#) *and fix* $\pi \in \Pi_\Theta$. *Under* [Asm. 2.1](#), [Asm. 4.1](#), [Asm. 4.2](#), [Asm. D.1](#), *and* [Asm. E.1](#), *w.p.* $\geq 1 - \delta$ *we have* $\|\nabla\widetilde{J}(\pi) - \widehat{\nabla}\widetilde{J}(\pi)\| \leq \varepsilon$ *when*
$$n = \widetilde{O}\left(\frac{\mathrm{pd}_\mathcal{G} H^6 G^2 \left(\sum_h C_h^\mathbf{s} C_h^\mathbf{a}\right)^2 L_\sigma^2 \log(|\mathcal{W}||\mathcal{F}|/\delta)}{\varepsilon^2}\right).$$

**Stationary convergence & computational efficiency.** Similar to OCCUPG, OFF-OCCUPG with $T = O(\beta H^2/\varepsilon^2)$ converges to an $\varepsilon$-stationary point. The formal statement is given in [Cor. C.1](#) and is based on the estimation guarantee in [Thm. 4.1](#). As a result, OFF-OCCUPG is also computationally oracle-efficient. Each invocation of MLE involves $2H$ calls to a likelihood maximization oracle (see [Alg. 4](#)), and each invocation of FORC requires $H$ calls to a squared-loss regression oracle (see [Alg. 5](#)). Then local convergence is still achieved with $O(\beta H^2/\varepsilon^2)$ such calls, as increasing $T$ further cannot reduce error from statistical noise (that depends only on the fixed $n$).

**Optimality.** Analyzing the conditions under which offline PG recovers global optima is more challenging, as we can no longer utilize exploratory initialization (from [Cor. 3.2](#)). However, since all occupancies have been clipped to the data distribution, we show in [App. C.5](#) that the offline data itself can sometimes suffice as an exploratory initial distribution, and the corresponding bound is in terms of $\{C_h^\mathbf{s}\}$ (instead of the online $\mathcal{C}^{\pi^*}$). However, this is not guaranteed in general and our current result only holds under strong all-policy offline data coverage. Briefly, some hardness comes from the fact that clipping causes gradient signals to vanish, so a stationary policy might be far off-support, rather than optimal. Investigating the possibility of more relaxed conditions for offline PG convergence (or, conversely, refining hardness results) are especially interesting directions for future work.

## 5 Conclusion

For the first time, we demonstrate how policy optimization can be conducted with (only) occupancy functions for both online and offline RL, and comprehensively analyze both local and global convergence. In the online setting our method directly extends to optimizing general objective functionals that cannot be optimized using value-based methods, and in the offline setting the occupancy-based gradient naturally handles incomplete offline data coverage. As our work is the first in this line of research and theoretical in nature, for future work we plan to launch empirical investigations of our methods, especially those for optimizing general functionals. Additionally, the conditions under which offline PG can converge to global optima is not well-understood, and we hope that our preliminary results here encourage greater interest and investigation into this question.

## Acknowledgements

Nan Jiang acknowledges funding support from NSF IIS-2112471, NSF CAREER IIS-2141781, Google Scholar Award, and Sloan Fellowship.

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

## Appendix

## A  Related work

In this section, we discuss related works in greater detail that concern the convergence and estimation of policy gradient in RL.

While a handful of recent papers have similarly observed that the gradient of the log density can be utilized to compute the policy gradient, especially in the context of using it to optimize general functionals, none of them have analyzed methods that are sample-efficient under general function approximation. In particular, [KJDC24] requires on-policy sampling from the time-reversed transition $(s, a|s')$, which, as they note, is highly restrictive. To overcome this issue they propose a min-max algorithm that converges under linear approximation, which is computationally a far more difficult to solve (under a more stringent structural assumption) than the regression objective in Alg. 1. Similarly, [BFH23] consider only online policy gradient, and to handle large state spaces they use linear function approximation, which may incur a large error through bias in many settings. [ZBWK20] approach the problem of optimizing risk functionals through a primal-dual approach that involves occupancies as dual variables, but they only analyze convergence in tabular settings.

A number of works on off-policy gradient optimization utilize all three of the density ratio, value, and policy class functions to compute the gradient [NDKCLS19; HJ22b; UIJKSX21; XYWL21]. This is because the density ratios are required to handle distribution mismatches with offline data. The downside, however, is that even max-min optimization is difficult, so performing optimization over all three functions requires complex optimization loops. By using simply projected gradient ascent on a policy class, our algorithms avoid such complexities and are amenable to classical convergence analysis that allow us to focus on the role of coverage coefficients in our final results.

Because the density ratio is generally not well-defined with arbitrary offline data, all of these works require some form of all-policy coverage for both estimation and convergence guarantees. The weight gradient calculation in [UIJKSX21] exhibits a recursive decomposition that is related to ours. However, their formulation is not compatible with our data assumptions and they require policy coverage to be well-defined. A follow-up paper in [XYWL21] uses squared-loss regression on the same updates, which is similar in flavor to our gradient estimation objectives. However, they use linear function approximation for weight functions, which is not realizable in general, and also require all-policy coverage for their convergence results.

One close work of comparison is [LSAB19], whose PG algorithm uses learned density ratios to reweight the data distribution and approximate the expression of the policy gradient theorem [SMSM99]. To handle coverage issues in offline PG, they "zero out" portions of the trajectory that exceed data coverage, but only do this for $(s, a)$ such that $d^D(s)\pi^D(a|s) = 0$. This is done (in the infinite horizon setting) by resampling the dataset based on an augmented MDP where such $(s, a)$ transition to an absorbing state However, this does not control the (potentially extremely large) variance of the estimator, e.g., on states where $d^D(s) \approx 0$. Their objective can be seen as a special case of $\bar{J}(\pi)$ with finite but extremely large choices of $C^{\mathbf{s}}$ and $C^{\mathbf{a}}$. They show convergence to a stationary point in terms of weight and value estimation errors that are left implicit, and leave the high variance and coverage issues with offline data implicit.

Another is [DWS12], that takes the complementary approach and simply calculates the gradient averaged on $d^D$. However, this is a biased gradient object and does not express the policy gradient of any specific function, which means its stationary point may not even exist, thus precluding convergence analysis.

The PSPI algorithm in [XCJMA21] is a policy optimization algorithm based on pessimistic value functions. Their setting is somewhat orthogonal to ours in the sense that they study values and we study occupancies, and we note that they do not perform policy optimization with respect to a standalone policy class but rather an implicit one induced by the value functions, which an be extremely large. In the value function sphere, [NZJZW22] leverage the linear structure of linear-MDPS to develop closed-form gradient estimators through the value functions. They largely only analyze estimation errors and additionally require a form of all-policy coverage for their results.

Lastly, our optimality analysis builds off the results in [BR24] and [AKLM21] that analyze global optimality in the (infinite horizon) online setting.

# B    Additional results and proofs for Sec. 3

## B.1    Proofs for Sec. 3.1

**Proof of Lem. 3.1** First we expand $\nabla d_h^\pi$ using the Bellman flow equation, $d_h^\pi(s') = \sum_{s,a} P(s'|s,a)\pi(a|s)d_{h-1}^\pi(s)$:

$$\nabla d_h^\pi(s') = \sum_{s,a} P(s'|s,a)(\nabla\pi(a|s)d_{h-1}^\pi(s) + \pi(a|s)\nabla d_{h-1}^\pi(s))$$
$$= \sum_{s,a} P(s'|s,a)\pi(a|s)d_{h-1}^\pi(s)(\nabla\log\pi(a|s) + \nabla\log d_{h-1}^\pi(s)).$$

In the last line above we use the grad-log trick. Note that $\nabla\log d^\pi(s)$ is not well-defined when $d^\pi(s) = 0$, but the two terms will cancel out in the above expression for this case. From Bayes' theorem, $\mathbb{P}^\pi(s_{h-1} = s, a_{h-1} = a|s_h = s') = P(s'|s,a)\pi(a|s)d_{h-1}^\pi(s)/d_h^\pi(s')$, thus

$$\nabla\log d_h^\pi(s') = \nabla d_h^\pi(s')/d_h^\pi(s')$$
$$= \mathbb{E}^\pi[\nabla\log\pi(a|s) + \nabla\log d_{h-1}^\pi(s)|s_h = s']$$
$$= [\mathbf{E}_{h-1}^\pi(\nabla\log\pi + \nabla\log d_{h-1}^\pi)](s').$$

We use the convention that $0/0 = 0$, thus $\nabla\log d_h^\pi$ is always well-defined.

Lastly, the second statement results from Lem. B.1, which shows that $\|\nabla\log d_h^\pi(s')\|$ is always bounded and well-defined under Asm. 2.1.

**Lemma B.1.** *Under Asm. 2.1, we have* $\max_{s,h} \|\nabla\log d_h^\pi(s)\| \le hG$.

*Proof.* The lemma statement can be derived inductively starting from the observation that Eq. (1) is an expectation over its target functions. As a result, the maximum gradient magnitude should accrue

additively over horizons. More concretely, fix $h$ and $s$. Then

$$
\begin{aligned}
\|\nabla \log \widetilde{d}_h^\pi(s')\| &= \|\mathbb{E}^\pi[\nabla \log \pi(a|s) + \nabla \log d_{h-1}^\pi(s)|s_h = s']\| \\
&\leq \|\nabla \log \pi(a|s)\| + \|\nabla \log d_{h-1}^\pi(s)\| \\
&\leq G + \|\nabla \log d_{h-1}^\pi(s)\|,
\end{aligned}
$$

using Asm. 2.1 in the last line. Since $\nabla \log d_0^\pi = \mathbf{0}$ by definition, unrolling the above recursion through timesteps gives the stated result. $\square$

**Lemma B.2.** *For $g : \mathcal{S} \to \mathbb{R}^p$ and $f : \mathcal{S} \times \mathcal{A} \to \mathbb{R}^p$, define the squared loss*

$$
L_h(g; f, \pi) = \mathbb{E}_\pi[\|g(s_{h+1}) - f(s_h, a_h)\|^2].
$$

*Then for any such $f$,*

$$
\mathbf{E}_h^\pi(f) = \underset{g:\mathcal{S}\to\mathbb{R}^p}{\operatorname{argmin}} L_h(g; f, \pi)
$$

*and*

$$
\nabla \log d_{h+1}^\pi = \underset{g:\mathcal{S}\to\mathbb{R}^p}{\operatorname{argmin}} L_h\left(g; \nabla \log \pi + \nabla \log d_h^\pi, \pi\right).
$$

*Proof of Lem. B.2.* Since the objective is convex, can solve for the minimizer in closed form by taking the derivative and setting it to 0 in an element-wise manner. Fix $s'$. Taking the gradient of $L_h(g; f, \pi)$ with respect to $g(s')$, we have that

$$
0 = d_h^\pi(s')g(s') - \sum_{s,a} P(s'|s,a)\pi(a|s)d_{h-1}^\pi(s)f(s,a).
$$

Rearranging and using the definition of $\mathbf{E}_h^\pi$ gives the result. The second statement follows from Lem. 3.1. $\square$

## B.2 Proofs for Sec. 3.2 : Estimation and local convergence

**Proof of Thm. 3.1** First we split up the errors contributed by regression and the estimation. Fix $\pi$, then $\mathbb{E}_{\mathcal{D}^{\mathrm{reg}}}[\widehat{\nabla} J(\pi)] = \sum_h \mathbb{E}_{s \sim d_h^\pi}[\widehat{g}_h^\pi(s)R_h(s)]$ and

$$
\|\nabla J(\pi) - \widehat{\nabla} J(\pi)\| \leq \|\nabla J(\pi) - \mathbb{E}_{\mathcal{D}^{\mathrm{reg}}}[\widehat{\nabla} J(\pi)]\| + \|\mathbb{E}_{\mathcal{D}^{\mathrm{reg}}}[\widehat{\nabla} J(\pi)] - \widehat{\nabla} J(\pi)\|
$$

The first term is related to the regression error in $\widehat{g}_h^\pi$ approximating $\nabla \log d_h^\pi$,

$$
\begin{aligned}
\|\nabla J(\pi) - \mathbb{E}_{\mathcal{D}^{\mathrm{reg}}}[\widehat{\nabla} J(\pi)]\| &= \left\|\sum_h \mathbb{E}_{d_h^\pi}[\nabla \log d_h^\pi(s)R_h(s)] - \mathbb{E}_{d_h^\pi}[\widehat{g}_h^\pi(s)R_h(s)]\right\| \\
&\leq \sum_h \left\|\mathbb{E}_{d_h^\pi}[\nabla \log d_h^\pi(s) - \widehat{g}_h^\pi(s)]\right\| \\
&= \sum_h \sqrt{\sum_{p=1}^p \|\nabla^p \log d_h^\pi - [\widehat{g}_h^\pi]^p\|_{1,d_h^\pi}^2}.
\end{aligned}
$$

For a fixed $h$ and $p$, we recursively decompose

$$
\begin{aligned}
\|\nabla^p \log d_h^\pi - [\widehat{g}_h^\pi]^p\|_{1,d_h^\pi} &\leq \|\nabla^p \log d_h^\pi - [\mathbf{E}_{h-1}^\pi(\nabla \log \pi + \widehat{g}_{h-1}^\pi)]^p\|_{1,d_h^\pi} \\
&\quad + \|[\mathbf{E}_{h-1}^\pi(\nabla \log \pi + \widehat{g}_{h-1}^\pi)]^p - [\widehat{g}_h^\pi]^p\|_{1,d_h^\pi} \\
&\leq \|\nabla^p \log d_{h-1}^\pi - [\widehat{g}_{h-1}^\pi]^p\|_{1,d_{h-1}^\pi} + \|[\mathbf{E}_{h-1}^\pi(\nabla \log \pi + \widehat{g}_{h-1}^\pi)]^p - [\widehat{g}_h^\pi]^p\|_{2,d_h^\pi},
\end{aligned}
$$

using the fact that $\nabla \log d_h^\pi = \mathbf{E}_{h-1}^\pi(\nabla \log \pi + \nabla \log d_{h-1}^\pi)$ in the second line. Then unrolling the recursion, we have

$$
\|\nabla^p \log d_h^\pi - [\widehat{g}_h^\pi]^p\|_{1,d_h^\pi} \leq \sum_h \|[\mathbf{E}_{h-1}^\pi(\nabla \log \pi + \widehat{g}_{h-1}^\pi)]^p - [\widehat{g}_h^\pi]^p\|_{2,d_h^\pi}
$$

Applying Lem. F.2 (more exactly, this is an offline version but we invoke it with $\rho = 1$ and no clipping for the online setting) with $\delta' = \delta/2Hp$ and a union bound over all $h$ and $p$, we have

$$
\|[\mathbf{E}_{h-1}^\pi(\nabla \log \pi + \widehat{g}_{h-1}^\pi)]^p - [\widehat{g}_h^\pi]^p\|_{2,d_h^\pi}^2 = \mathbb{E}[\mathcal{L}_{\mathcal{D}_{h-1}^{\mathrm{reg}}}^p(\widehat{g}_h^\pi, \widehat{g}_{h-1}^\pi) - \mathcal{L}_{\mathcal{D}_{h-1}^{\mathrm{reg}}}^p(\mathbf{E}_{h-1}^\pi(\nabla \log \pi + \widehat{g}_{h-1}^\pi), \widehat{g}_{h-1}^\pi)]
$$

$$\leq 2(\varepsilon_{h-1}^{\text{reg}})^2,$$

where $\varepsilon_{h-1}^{\text{reg}} = \sqrt{\frac{c d_{\mathcal{G}} h^2 G^2 \log(2H\mathsf{p}/\delta)}{n}}$. Then for any $h, p$ we have

$$\|\nabla^p \log d_h^\pi - [\widehat{g}_h^\pi]^p\|_{1, d_h^\pi} \leq \sqrt{2} \sum_{g \leq h} \varepsilon_g^{\text{reg}} \leq \sqrt{2} h \varepsilon_h^{\text{reg}} = \sqrt{\frac{2 c d_{\mathcal{G}} h^4 G^2 \log(2H\mathsf{p}/\delta)}{n}}$$

Implying

$$\|\nabla J(\pi) - \mathbb{E}_{\mathcal{D}^{\text{reg}}}[\widehat{\nabla} J(\pi)]\| \leq \sqrt{\mathsf{p}} H \|\nabla^p \log d_H^\pi - [\widehat{g}_H^\pi]^p\|_{1, d_H^\pi} = \sqrt{\frac{2 c \mathsf{p} d_{\mathcal{G}} H^6 G^2 \log(2H\mathsf{p}/\delta)}{n}}$$

For the second term,

$$|\mathbb{E}_{\mathcal{D}^{\text{reg}}}[\widehat{\nabla}^p J(\pi)] - \widehat{\nabla}^p J(\pi)| \leq \sum_h |\mathbb{E}_{s \sim d_h^\pi}[\widehat{g}_h^\pi(s) R_h(s)] - \frac{1}{n} \sum_{s \in \mathcal{D}_h} \widehat{g}_h^\pi(s) R_h(s)| \leq H^2 G \sqrt{\frac{\log(2\mathsf{p}H/\delta)}{n}},$$

where we use Hoeffding's inequality with union bound, for all $h \in [H]$ and $p \in \mathsf{p}$ in the last line, given that the randomness of $\widehat{g}$ is independent given $\mathcal{D}_h^{\text{reg}}$. Thus

$$\|\mathbb{E}_{\mathcal{D}^{\text{reg}}}[\widehat{\nabla} J(\pi)] - \widehat{\nabla} J(\pi)\| \leq H^2 G \sqrt{\frac{\mathsf{p} \log(2\mathsf{p}H/\delta)}{n}}$$

Combining the two terms, our final bound is

$$\|\nabla J(\pi) - \widehat{\nabla} J(\pi)\| \lesssim \sqrt{\frac{\mathsf{p} d_{\mathcal{G}} H^6 G^2 \log(2H\mathsf{p}/\delta)}{n}},$$

with the regression error dominating.

**Proof of Cor. 3.1** For any fixed run of Alg. 1, calling Thm. 3.1 for $\pi^{(t)}$ with $\delta' = \delta/T$ and taking a union bound over $T$ gives with probability at least $1 - \delta$ that

$$\|\nabla J(\pi^{(t)}) - \widehat{\nabla} J(\pi^{(t)})\| \lesssim \sqrt{\frac{\mathsf{p} d_{\mathcal{G}} H^6 G^2 \log(2H\mathsf{p}T/\delta)}{n}}, \ \forall t \in [T].$$

Then setting $\delta = 1/\sqrt{n}$, we have

$$\mathbb{E}\left[\|\nabla J(\pi^{(t)}) - \widehat{\nabla} J(\pi^{(t)})\|\right] \lesssim \sqrt{\frac{\mathsf{p} d_{\mathcal{G}} H^6 G^2 \log(2H\mathsf{p}Tn)}{n}},$$

where the expectation is over random samples in $\mathcal{D}^{\text{reg}}, \mathcal{D}^{\text{est}}$. Finally, plugging this into the PGD stationary convergence bound in Lem. G.1 gives

$$\frac{1}{T} \sum_t \mathbb{E}\left[\|G^\eta(\pi^{(t)}, \nabla J(\pi^{(t)}))\|^2\right] \leq \frac{4\beta H}{T} + \frac{6\mathsf{p} d_{\mathcal{G}} H^6 G^2 \log(2H\mathsf{p}Tn)}{n}$$

Setting the RHS to $\varepsilon$ and setting $T, n$ appropriately gives the result.

### B.3 Proofs for Sec. 3.2: Global convergence

We will establish the conditions under which $J(\pi)$ satisfies a gradient domination property, meaning that for any $\theta \in \Theta$, the suboptimality of $\pi_\theta$ is bounded by some function $S$ that includes a measure of its stationarity, i.e., $\max_{\pi' \in \Pi_\Theta} \widetilde{J}(\pi') - J(\pi_\theta) \leq S(\nabla \widetilde{J}(\pi_\theta))$. This combined with the sample complexity bounds for stationary convergence established in Cor. 3.1 enables our global optimality result in Cor. 3.2.

Though we are concerned with optimizing $J(\pi)$ induced by running $\pi$ starting from initial distribution $d_0$, it will be useful to consider performing Alg. 1 using a different exploratory initial distribution $\mu \in \Delta(\mathcal{S})$. By exploratory, we mean that we allow $\mu(s) > 0$ for all $s \in \mathcal{S}$, unlike $d_0 \in \Delta(\mathcal{S}^0)$. In the (stationary) infinite horizon this is a common trick for obtaining well-defined gradient domination bounds [AKLM21], but its finite-horizon (nonstationary) counterpart is nontrivial and to our knowledge has not previously been formalized (it is listed as future work in [BR24]).

We state and prove a more general version of Lem. 3.2:

**Lemma B.3.** *For any $\pi$ and $\pi'$, define $B^\pi(\pi') := \sum_{h,s,a} d_h^\pi(s)\pi'(a|s)Q_h^\pi(s,a)$. Suppose $\forall \pi \in \Pi_\Theta$,*

1. *(Policy completeness) There exists $\pi^+ \in \Pi_\Theta$ such that $\pi^+ \in \mathrm{argmax}_{\pi'} B^\pi(\pi')$.*
2. *(Gradient domination) $\max_{\pi' \in \Pi_\Theta} B^\pi(\pi') - B^\pi(\pi) \le m \max_{\theta' \in \Theta} \langle \nabla B^\pi(\pi), \theta' - \theta \rangle$.*

*Given $\nu \in \Delta(\mathcal{S})$, define the coverage coefficient $\mathcal{C}^{\pi^*} := \left\| \sum_h d_h^{\pi^*} / \nu \right\|_\infty$ for $\pi^* = \mathrm{argmax}_\pi J(\pi)$. Then for any $\pi_\theta \in \Pi_\Theta$,*

$$J(\pi^*) - J(\pi_\theta) \le m \left\| \frac{\sum_h d_h^{\pi^*}}{\sum_h d_{\mu,h}^{\pi_\theta}} \right\|_\infty \max_{\theta' \in \Pi_\Theta} \langle \nabla J_\mu(\pi_\theta), \theta' - \theta \rangle$$

$$\le \mathcal{C}^{\pi^*} \max_{\theta' \in \Pi_\Theta} \langle \nabla J_\nu(\pi_\theta), \theta' - \theta \rangle,$$

*where $J_\nu(\pi) := \mathbb{E}_{s_0 \sim \nu, \pi}[\sum_h r_h]$ is the expected return of $\pi$ in $\mathcal{M}$ with initial state distribution $\nu$.*

**Proof of Lem. B.3** First we note two facts that hold regardless of $\mathcal{M}$. We have $Q_g^\pi(s^h, a^h) = Q_h^\pi(s^h, a^h)$ for any $g \le h$, and $d_g^\pi(s^h) = 0$ if $g > h$.

$$J(\pi^*) - J(\pi_\theta) = \sum_{h=0}^{H-1} \sum_{s,a} d_h^{\pi^*}(s)\left(\pi^*(a|s) - \pi_\theta(a|s)\right) Q_h^{\pi_\theta}(s,a)$$

Then we will write $Q^\pi(s,a) \equiv Q_h^\pi(s,a)$, thus

$$J(\pi^*) - J(\pi_\theta) = \sum_{h=0}^{H-1} d_h^{\pi^*}(s)\left(\pi^*(a|s) - \pi_\theta(a|s)\right) Q^{\pi_\theta}(s,a)$$

$$= \sum_{s,a} \left( \sum_h d_h^{\pi^*}(s) \right) \left(\pi^*(a|s) - \pi_\theta(a|s)\right) Q^{\pi_\theta}(s,a)$$

$$\le \max_{\pi^+} \sum_{s,a} \left( \sum_h d_h^{\pi^*}(s) \right) \left(\pi^+(a|s) - \pi_\theta(a|s)\right) Q^{\pi_\theta}(s,a)$$

$$\le \max_{\pi^+} \sum_{s,a} \frac{\sum_h d_h^{\pi^*}(s)}{\sum_h d_{\mu,h}^{\pi_\theta}(s)} \left( \sum_h d_{\mu,h}^{\pi_\theta}(s) \right) \left(\pi^+(a|s) - \pi_\theta(a|s)\right) Q^{\pi_\theta}(s,a)$$

$$\le \left\| \frac{\sum_h d_h^{\pi^*}}{\mu} \right\|_\infty \max_{\pi^+} \sum_{s,a} \left( \sum_h d_{\mu,h}^{\pi_\theta}(s) \right) \left(\pi^+(a|s) - \pi_\theta(a|s)\right) Q^{\pi_\theta}(s,a)$$

For the RHS, observe that $d_{\mu,g}^{\pi_\theta}(s^h) = 0$ for $g > h$. Then

$$\sum_h \sum_{s,a} d_{\mu,h}^{\pi_\theta}(s)\left(\pi^+(a|s) - \pi_\theta(a|s)\right) Q^{\pi_\theta}(s,a)$$

$$= \sum_h \sum_{s,a} d_{\mu,h}^{\pi_\theta}(s)\left(\pi^+(a|s) - \pi_\theta(a|s)\right) Q_h^{\pi_\theta}(s,a)$$

$$= \sum_h \sum_{s,a} d_{\mu,h}^{\pi_\theta}(s)\left(\pi^+(a|s) - \pi_\theta(a|s)\right) Q_{\mu,h}^{\pi_\theta}(s,a)$$

$$= B^{\pi_\theta}(\pi^+) - B^{\pi_\theta}(\pi_\theta)$$

$$\le m \max_{\theta' \in \Theta} \langle \nabla B^{\pi_\theta}(\pi_\theta), \theta' - \theta \rangle$$

$$= m \max_{\theta' \in \Theta} \langle \nabla J_\mu(\pi_\theta), \theta' - \theta \rangle$$

Combining the two inequalities results in the final guarantee.

**Proof of Cor. 3.2** Fix $\{\pi^{(t)}\}_{t \in [T]}$ from Alg. 1. Then for any $t \in [T]$, from Lem. 3.2 we have

$$J(\pi^*) - J(\pi^{(t)}) \le m\mathcal{C}^{\pi^*} \max_{\theta' \in \Pi_\Theta} \left\langle \nabla J(\pi^{(t)}), \theta' - \theta \right\rangle$$

$$\leq Bm\mathcal{C}^{\pi^*}\|G^{\eta}(\pi^{(t)})\| \qquad \text{(Lem. G.4)}$$

Then summing through $T$ and taking an expectation over the randomness in the algorithm, we have

$$\mathbb{E}\left[\frac{1}{T}\sum_t J(\pi^*) - J(\pi^{(t)})\right] \leq Bm\mathcal{C}^{\pi^*}\mathbb{E}\left[\frac{1}{T}\sum_t \|G^{\eta}(\pi^{(t)})\|\right]$$

$$\leq Bm\mathcal{C}^{\pi^*}\left(\frac{4\beta H}{T} + \frac{6\mathsf{pd}_{\mathcal{G}}H^6G^2\log(2H\mathsf{p}Tn)}{n}\right). \qquad \text{(Cor. 3.1)}$$

### B.4 Examples of gradient function class $\mathcal{G}$

This section contains formal statements of the claims in Rem. 3.1, and their proofs. We begin by defining the low-rank MDP, noting that for notational compactness we have dropped the features' $h$-dependence given our assumption that there is a one-to-one correspondence between states and the timestep at which they are visited.

**Definition B.1** (Low-rank MDP). We say $\mathcal{M}$ is a low-rank MDP with dimension $k$ if $\forall h \in [H]$, there exists $\phi : \mathcal{S} \times \mathcal{A} \to \mathbb{R}^k$ and $\mu_h : \mathcal{S} \to \mathbb{R}^k$ such that $(s, a, s')$, we have $P(s'|s, a) = \langle \phi(x, a), \mu(x')\rangle$. Further, $\|\phi\|_\infty \leq C^\phi$ and $\sum_s \mu(s) \leq C^\mu$.

Prop. B.1 shows that, in low-rank MDPs, a linear-over-linear parameterization for the gradient function class satisfies the completeness requirement in Asm. 3.1, with pseudo-dimension linear in the low-rank dimension and the parameter dimension, i.e., $\mathsf{d}_{\mathcal{G}_h} = O(k\mathsf{p})$.

**Proposition B.1.** *Suppose $\mathcal{M}$ is a low-rank MDP (Def. B.1), and suppose $\mu$ is known. For each layer $h$, define the function class*

$$\mathcal{G}_h = \left\{g_h = \frac{\mu^\top \Psi}{\mu^\top \psi} : \Psi \in \mathbb{R}^{k\times\mathsf{p}}, \psi \in \mathbb{R}^k, \|g_h\|_\infty \leq hG, \forall h \in [H]\right\}.$$

*Then $\{\mathcal{G}_h\}$ satisfies Asm. 3.1 and has pseudodimension (Def. F.1) $\mathsf{d}_{\mathcal{G}_h} = O(k\mathsf{p})$.*

*Proof of Prop. B.1.* It suffices to show that, for any function $f : \mathcal{S} \times \mathcal{A} \to \mathbb{R}^\mathsf{p}$ and policy $\pi$, its gradient update from Lem. 3.1 is $\mathbf{E}_h^\pi(\nabla \log \pi + f) \in \mathcal{G}_{h+1}$.

Since $[\mathbf{E}_h^\pi(\nabla \log \pi + f)](s') = \mathbb{E}_\pi[\nabla \log \pi(s, a) + f(s)|s']$, from Bayes' rule and the definition of the Bellman flow operator (see proof of Lem. 3.1), we have

$$[\mathbf{E}_h^\pi(\nabla \log \pi + f)](s') = \frac{[\mathbf{P}_{h-1}^\pi(\nabla \log \pi + f)](s')}{d_h^\pi(s')}.$$

First, we will show that $[\mathbf{P}_h^\pi f](s') = \mu(s')^\top \Psi$ for some $\Psi \in \mathbb{R}^{k\times\mathsf{p}}$ and all $s' \in \mathcal{S}$. Below, we use $f^p(s)$ to denote the $p$-th parameter of $f(s) \in \mathbb{R}^\mathsf{p}$. For fixed $p \in [\mathsf{p}]$,

$$[\mathbf{P}_h^\pi f]^p(s') = \sum_{s,a} P(s'|s, a)\pi(a|s)\left(\nabla^p \log \pi(a|s) + f^p(s)\right)$$

$$= \mu(s')^\top \left(\sum_{s,a} \phi(s, a)\pi(a|s)\left(\nabla^p \log \pi(a|s) + f^p(s)\right)\right)$$

$$= \mu(s')^\top \psi,$$

where $\psi = \sum_{s,a} \phi(s, a)\pi(a|s)\left(\nabla \log \pi(a|s) + f^p(s)\right) \in \mathbb{R}^k$. Stacking this result for each $p$ into the matrix $\Psi$ shows the desired statement that $[\mathbf{P}_h^\pi f](s') = \mu(s')^\top \Psi$.

We can apply similar reasoning as above in the Bellman flow equation to show that $d_h^\pi(s') = \langle \mu(s'), \psi\rangle$ for some $\theta \in \mathbb{R}^k$. Combined with the above, this shows that

$$\nabla \log d_h^\pi(s') = \frac{\mu^\top \Psi}{\mu^\top \psi},$$

Combining the linear forms of the numerator and denominator reveal that $\nabla \log d_h^\pi \in \mathcal{G}_h$. Lastly, the pseudo-dimension of $\mathcal{G}_h$ follows directly from applying Lemma 24 of [HCJ23], which bounds the pseudo-dimension of linear-over-linear function classes with $\mathsf{p} = 1$, in all $\mathsf{p}$ dimensions. $\qquad \square$

---

**Algorithm 3** Online Occupancy-based PG for General Functionals

---

**Input:** Functional $F = \{F_h\}$; Samples $n$; iterations $T$; policy class $\Pi_\Theta$; function class $\mathcal{G}$; learning rate $\eta$; function class $\mathcal{F}$;

1: **for** $t = 0, \ldots, T-1$ **do**
2:     For all $h \in [H]$, deploy $\pi^{(t)}$ for $3n$ trajectories. Set $\mathcal{D}_h^{\mathrm{reg}} = \{(s_h, a_h, s_{h+1})\}_{i=1}^n$, and similarly for $\mathcal{D}_h^{\mathrm{grad}}$ and $\mathcal{D}_h^{\mathrm{mle}}$
3:     **for** $h = 1, \ldots, H-1$ **do**
4:         Define $\mathcal{L}_{h-1}^{(t)}(g_h, g_{h-1}) := \frac{1}{n} \sum_{(s,a,s') \in \mathcal{D}_{h-1}^{\mathrm{reg}}} \left\| g_h(s') - \left( \nabla \log \pi^{(t)}(a|s) + g_{h-1}(s) \right) \right\|^2$,
         and set
$$\widehat{g}_h^{(t)} = \mathrm{argmin}_{g_h \in \mathcal{G}_h} \mathcal{L}_{h-1}^{(t)}(g_h, \widehat{g}_{h-1}^{(t)}).$$
5:     **end for**
6:     Estimate $\widehat{d_h^\pi} \leftarrow \mathrm{MLE}(\mathcal{D}^{\mathrm{mle}}, \mathcal{F})$.                       // Alg. 4
7:     Estimate $\widehat{\nabla} J_F(\pi) = \frac{1}{n} \sum_h \sum_{s \in \mathcal{D}_h^{\mathrm{est}}} \widehat{g}_h^\pi(s) \left. \frac{\partial F_h(d)}{\partial d(s)} \right|_{d = \widehat{d}_h^\pi}$
8:     Update $\theta^{(t+1)} = \mathrm{Proj}_\Theta \left( \theta^{(t)} + \eta \widehat{\nabla} J(\pi^{(t)}) \right)$.
9: **end for**

---

## B.5   Policy optimization of general functionals

Alg. 3 displays the full algorithm for optimization of general functions (described in Sec. 3.3). It shares its occupancy gradient estimation module with OCCUPG. Compared to Alg. 1, the only change is the objective gradient calculation in Line 7, which uses a plug-in estimate of the occupancy (Line 6) to evaluate the partial derivative.

Since the algorithmic change is small, the analysis for Alg. 3 requires only a few adaptations from the analysis of OCCUPG. For smooth and differentiable functionals, we provide the gradient estimation guarantee below. The smoothness ensures that using plug-in occupancy estimates to evaluate the partial derivative leads to consistent gradient estimates, and is in line with the spirit of standard objective smoothness requirements (Asm. 3.2).

**Assumption B.1.** Suppose that for all $h$, $F_h$ has a smooth gradient, i.e., for any $f, f' \in \Delta(\mathcal{S})$ that

$$\sum_s \left| \left. \frac{\partial F_h(d)}{\partial d(s)} \right|_{d=f} - \left. \frac{\partial F_h(d)}{\partial d(s)} \right|_{d=f'} \right| \le L_F \|f - f'\|_1,$$

and has bounded range $\|\partial F_h(d)\|_\infty \le C_F$.

**Theorem B.1.** *Suppose that Asm. 2.1 and Asm. B.1 hold. Fix $\pi \in \Pi_\Theta$. With probability at least $1 - \delta$,*

$$\|\nabla J_F(\pi) - \widehat{\nabla} J_F(\pi)\| \lesssim H^2 G L_F \sqrt{\frac{\mathsf{p} \log(2 \mathsf{p} H |\mathcal{F}|/\delta)}{n}} + C_F \sqrt{\frac{\mathsf{pd}_{\mathcal{G}} H^6 G^2 \log(2H \mathsf{p}/\delta)}{n}}.$$

When Asm. 3.2 holds, this result directly leads to a stationary convergence guarantee similar to Cor. 3.1, by union bounding Thm. B.1 over all $T$ then plugging it into Lem. G.5 (see proof of Cor. 3.1). We expect that the global convergence in Cor. 3.2 can also be extended with little overhead when $\{F_h\}$ are convex, but leave a full investigation to future work.

**Proof of Thm. B.1**    The analysis follows largely the same lines as the proof of Thm. 3.1. However, we must additional account for the error of approximating $\left. \frac{\partial F_h(d)}{\partial d(s)} \right|_{d=\widehat{d}_h^\pi}$ with the plug-in occupancy estimate. This was unnecessary for the expected return in Sec. 3.2 since $\frac{\partial F_h(d)}{\partial d(s)} = R_h(s)$ is independent of the occupancy.

First, for all $h \in [H]$, with probability at least $1 - \delta$ we have occupancy estimates from Alg. 4 such that

$$\|d_h^\pi - \widehat{d}_h^\pi\|_1 \le \sqrt{\frac{2 \log(H|\mathcal{F}|/\delta)}{n}} := \varepsilon^{\mathrm{mle}}, \; \forall h \in [H].$$

This follows directly from Lem. D.1 with a union bound over $H$.

Next, we isolate the occupancy estimation-related term from the error we would like to bound. Define $\nabla \widehat{J}_F(\pi) := \sum_h \mathbb{E}_{s \sim d_h^\pi} \left[ \frac{\partial F_h(d)}{\partial d(s)} \big|_{d=\widehat{d}_h^\pi} \nabla \log d_h^\pi(s) \right]$, and decompose

$$\|\nabla J_F(\pi) - \widehat{\nabla} J_F(\pi)\| \leq \|\nabla J_F(\pi) - \nabla \widehat{J}_F(\pi)\| + \|\nabla \widehat{J}_F(\pi) - \widehat{\nabla} J_F(\pi)\|$$

For the first term,

$$
\begin{aligned}
\|\nabla J_F(\pi) - \nabla \widehat{J}_F(\pi)\| &\leq \sum_h \left\| \mathbb{E}_{s \sim d_h^\pi} \left[ \frac{\partial F_h(d)}{\partial d(s)} \big|_{d=d_h^\pi} \nabla \log d_h^\pi(s) - \frac{\partial F_h(d)}{\partial d(s)} \big|_{d=\widehat{d}_h^\pi} \nabla \log d_h^\pi(s) \right] \right\| \\
&\leq HG \sum_h \left\| \mathbb{E}_{s \sim d_h^\pi} \left[ \frac{\partial F_h(d)}{\partial d(s)} \big|_{d=d_h^\pi} - \frac{\partial F_h(d)}{\partial d(s)} \big|_{d=\widehat{d}_h^\pi} \right] \right\| \\
&\leq HG \sum_{h,s} \left| \frac{\partial F_h(d)}{\partial d(s)} \big|_{d=d_h^\pi} - \frac{\partial F_h(d)}{\partial d(s)} \big|_{d=\widehat{d}_h^\pi} \right| \\
&\leq H^2 G L_F \max_h \|d_h^\pi - \widehat{d}_h^\pi\|_1, \\
&\leq H^2 G L_F \varepsilon^{\mathrm{mle}}.
\end{aligned}
$$

using Asm. B.1 in the second to last inequality. This takes care of the aforementioned occupancy estimation error.

Conditioned on such $\{\widehat{d}_h^\pi\}$, the second pair of terms $\|\nabla \widehat{J}_F(\pi) - \widehat{\nabla} J_F(\pi)\|$ is analogous to the error bounded in Thm. 3.1, and the proof follows identically thereon, but with dependence on the range $C_F$ of the functionals.

## C  Additional results and proofs for Sec. 4

### C.1  Proofs for Sec. 4.1

**Proof of Lem. 4.1**  By passing the gradient through the clipped Bellman flow equation in Def. 4.2, we have

$$
\begin{aligned}
&\nabla \bar{d}_h^\pi(s') \\
&= \sum_{s,a} P(s'|s,a) \left( \nabla \bar{\pi}_{h-1}(a|s) \bar{d}_{h-1}^\pi(s) + \pi(a|s) \nabla \left( \bar{d}_{h-1}^\pi(s) \wedge C_{h-1}^{\mathbf{s}} d_{h-1}^D(s) \right) \right) \\
&= \sum_{s,a} P(s'|s,a) \bar{\pi}_{h-1}(a|s) \left( \bar{d}_{h-1}^\pi(s) \wedge C_{h-1}^{\mathbf{s}} d_{h-1}^D(s) \right) \Big( \nabla \log \bar{\pi}_{h-1}(a|s) \\
&\qquad\qquad\qquad\qquad\qquad\qquad + \nabla \log \left( \bar{d}_{h-1}^\pi(s) \wedge C_{h-1}^{\mathbf{s}} d_{h-1}^D(s) \right) \Big)
\end{aligned}
$$

Next, dropping the $h-1$ subscript for a moment, observe that

$$
\nabla \log \left( \bar{d}^\pi(s) \wedge C^{\mathbf{s}} d^D(s) \right) = \begin{cases} \nabla \log \bar{d}^\pi(s), & \text{if } \bar{d}^\pi(s) < C^{\mathbf{s}} d^D(s), \\ 0, & \text{if } \bar{d}^\pi(s) > C^{\mathbf{s}} d^D(s), \end{cases}
$$

with a discontinuity at $\bar{d}^\pi(s) = C^{\mathbf{s}} d^D(s)$. For simplicity, we set $\nabla \log \left( \bar{d}^\pi(s) \wedge C^{\mathbf{s}} d^D(s) \right) = \nabla \log \bar{d}^\pi(s) \odot \mathbf{1}[\bar{d}^\pi(s) \leq C^{\mathbf{s}} d^D(s)]$. Similarly, we have $\nabla \log \bar{\pi}(a|s) = \nabla \log \pi(a|s) \odot \mathbf{1}[\pi(a|s) \leq C^{\mathbf{a}} \pi^D(a|s)]$.

Finally, $\nabla \log \bar{d}_h^\pi(s') = \nabla \bar{d}_h^\pi(s')/\bar{d}_h^\pi(s')$, where $\bar{d}_h^\pi(s') = \sum_{s,a} P(s'|s,a) \pi_{h-1}^D(a|s) d_{h-1}^D(s) \bar{\rho}_{h-1}^\pi$. The lemma statement follows from the definition of $\mathbf{E}_{h-1}^{D,\bar{\rho}}$, and the gradient magnitude bound results from invoking Lem. C.2 with $\sigma(x,c) = (x \wedge c)$.

**Proof of Prop. 4.1**  This result follows from applying Lem. C.7, which is a more general version of the proposition statement that holds for any (smooth-)clipping function, to $\sigma(x,c) = (x \wedge c)$.

**Proof of Prop. 4.2**  The MDP we will describe corresponds to a multi-armed bandit with 2 actions. Consider an MDP with $H = 2$, and $\mathcal{S}^0 = \{s_0\}$, $\mathcal{S}^1 = \{s_L, s_R\}$, $\mathcal{S}^2 = \{s_-, s_+\}$ which are terminal. In any state there are two actions, $\mathcal{A} = \{L, R\}$, with deterministic transitions. For the first level, we have $s_0 \overset{L}{\to} s_L$ and $s_0 \overset{R}{\to} s_R$. For the second level, we have $s_L \to s_-$ and $s_R \to s_+$, regardless of action taken. For the reward function, $R(s_+) = 1$ and is 0 otherwise.

The policy is parameterized by a single parameter $\theta$ such that $\pi(L) = 1 - \theta$, and $\pi(R) = \theta$, such that $d_1^{\pi_\theta}(s_R) = d_2^{\pi_\theta}(s_+) = \theta$. Further, both the offline data and behavior policy are uniform in each level. Consequently, $\pi_0^D(L) = \pi_0^D(R) = \frac{1}{2}$ and $d_1^D = \mathrm{unif}(\mathcal{S}^1)$. We set $C_1^{\mathbf{s}} = C_2^{\mathbf{s}} = 2$, and $C_2^{\mathbf{a}} = 2$ so that $\bar{\pi}_h = \pi_h$ for all $h$.

Fix $\theta$ and estimated occupancies $\hat{d}^{\pi_\theta}$ and $\hat{d}^D$. For any $s' \in \mathcal{S}^2$ we have

$$\left\| \nabla \log d_2^{\pi_\theta}(s') - \widehat{\nabla} \log \bar{d}_2^{\pi_\theta}(s') \right\| = \left\| \nabla \bar{d}_1^{\pi_\theta}(s') \left( \mathbf{1}[\hat{d}_1^{\pi_\theta}(s') \le \hat{d}_1^D(s')] - \mathbf{1}[d_1^{\pi_\theta}(s') \le d_1^D(s')] \right) \right\|$$

Next, we instantiate $\hat{d}^{\pi_\theta}, \hat{d}^D$ for any $\pi_\theta$. The preconditions of the proposition are satisfied by an estimated occupancy with $\hat{d}_1^{\pi_\theta}(s_L) = \theta + \epsilon/2$ and $\hat{d}_1^{\pi_\theta}(s_R) = \theta - \epsilon/2$. In addition, we have an estimate $\hat{d}^D$ with $\hat{d}_1^D(s_L) = 1/2 - \epsilon/2$ and $\hat{d}_1^D(s_R) = 1/2 + \epsilon/2$.

We will consider $\theta = 1/2$, so that $d_1^{\pi_\theta} \le C_1^{\mathbf{s}} d_1^D$. However, $\hat{d}_1^{\pi_\theta}(s_L) > \hat{d}_1^D(s_L)$. As a result,

$$\left\| \nabla \log d_2^{\pi_\theta}(s') - \widehat{\nabla} \log \bar{d}_2^{\pi_\theta}(s') \right\| = \left\| \nabla \bar{d}_1^{\pi_\theta}(s') \right\| = O(1)$$

## C.2 Proofs for Sec. 4.2

First, we formally state and prove the claim that Lem. 4.2 can be reduced to minimizing a squared-loss regression problem recursively over timesteps, i.e.,

$$\nabla \log \widetilde{d}_{h+1}^\pi \tag{11}$$
$$= \operatorname*{argmin}_{g:\mathcal{S}\to\mathbb{R}^{\mathsf{p}}} \mathbb{E}_{\mathcal{D}_h} \left[ \left\| g(s') - \left( \nabla \log \pi \odot \tilde{\mathbf{1}} \left( \pi, C_h^{\mathbf{a}} d_h^D \right) + \nabla \log \widetilde{d}_{h-1}^\pi \odot \tilde{\mathbf{1}} \left( \widetilde{d}_h^\pi, C_h^{\mathbf{s}} d_h^D \right) \right) \right\|^2 \right].$$

This is a reweighted offline analog of Eq. (2) from the online setting, and a more general version is presented below with proof.

**Lemma C.1.** *For $g : \mathcal{S} \to \mathbb{R}^{\mathsf{p}}$ and $f : \mathcal{S} \times \mathcal{A} \to \mathbb{R}^{\mathsf{p}}$ and reweighting function $\rho : \mathcal{S} \times \mathcal{A} \to \mathbb{R}_+$, define the offline reweighted squared loss regression objective*

$$\widetilde{\mathcal{L}}_h(g; f, \rho) = \mathbb{E}_{\mathcal{D}_h}[\rho(s_h, a_h)\|g(s_{h+1}) - f(s_h, a_h)\|^2].$$

*Then for any such $f$,*

$$\mathbf{E}_h^{D,\rho}(f) = \operatorname*{argmin}_{g:\mathcal{S}\to\mathbb{R}^{\mathsf{p}}} \widetilde{\mathcal{L}}_h(g; f, \rho).$$

*Further, for the smooth-clipped density ratio $\widetilde{\rho}_h^\pi = \frac{\sigma\left(\widetilde{d}_h^\pi(s), C_h^{\mathbf{s}} d_h^D(s)\right)}{d_h^D(s)} \frac{\widetilde{\pi}_h(a|s)}{\pi_h^D(a|s)}$ and smooth-clipped target function $y_h^\pi := \nabla \log \pi \odot \tilde{\mathbf{1}} \left( \pi, C_h^{\mathbf{a}} d_h^D \right) + \nabla \log \widetilde{d}_h^\pi \odot \tilde{\mathbf{1}} \left( \widetilde{d}_h^\pi, C_h^{\mathbf{s}} d_h^D \right)$ from Lem. 4.2, we have*

$$\nabla \log \widetilde{d}_{h+1}^\pi = \operatorname*{argmin}_{g:\mathcal{S}\to\mathbb{R}^{\mathsf{p}}} \widetilde{\mathcal{L}}_h \left( g; y_h^\pi, \widetilde{\rho}_h^\pi \right).$$

*Proof of Lem. B.2.*  Since the objective is convex, can solve for the minimizer in closed form by taking the derivative and setting it to 0 in an element-wise manner. For each $s'$,

$$0 = g(s') \left( \sum_{s,a} P(s'|s,a)\pi_h^D(a|s)d_h^D(s)\rho(s,a) \right) - \sum_{s,a} P(s'|s,a)\pi_h^D(a|s)d_h^D(s)\rho(s,a)f(s,a).$$

Rearranging and using the definition of $\mathbf{E}_h^{D,\rho}$ (Eq. (5)) gives the result. The second statement follows from Lem. 4.2. $\qquad\square$

**Proof of Prop. 4.3** Part 1 follows from the gradient formula

$$\partial_x \sigma(x, c) = x^{-\beta-1}\left(x^{-\beta} + c^{-\beta}\right)^{-1/\beta-1} = \left(1 + x^\beta c^{-\beta}\right)^{-1/\beta-1}.$$

It can be seen that $\partial_x \sigma(x,c) \in [0,1]$ and is non-increasing in its inputs, thus $\sigma$ is monotonic. Additionally, $|\sigma(x,c) - \sigma(x',c)| \leq |x - x'|$. Since $\sigma$ is symmetric in its arguments, we also have $|\sigma^{\mathbf{s}}(x,c) - \sigma^{\mathbf{s}}(x,c')| \leq |c - c'|$.

Next, we prove Part 2. Let $z = (x \wedge c)$, and observe that $z - \sigma(x,c) \leq z - \sigma(z,z)$ since $\sigma$ is monotonic. Further,

$$\frac{z - \sigma(z,z)}{z} = \frac{z - (2z^{-\beta})^{-1/\beta}}{z} = 1 - 2^{-1/\beta} \leq 1 - e^{-1/\beta} \leq \frac{1}{\beta}.$$

Rearranging and plugging in the expression for $z$ gives the result.

Part 3 can be derived algebraically (but not easily), and is best intuited from the plot of the maximum slope $\sup_{x,x',c,c' \in [0,1]} |\tilde{\mathbf{1}}(x,c) - \tilde{\mathbf{1}}(x',c)|/|x - x'|$ in Figure C.2, which corresponds to $L_\sigma/c$ in the RHS of the bound. The left plot shows that the maximum slope increases linearly in $\beta$, and the right plot shows it increases inversely with $c$. The dashed red line is a "guess" for the exact constant $L_\sigma/c = 0.3\beta/c$, that upper-bounds the maximum slope. Clearly, $L_\sigma = O(\beta)$.

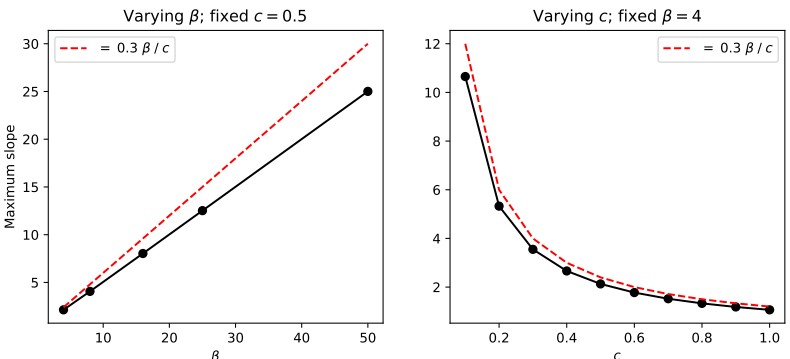

Figure 2: The y-axis plots the maximum slope $\sup_{x,x',c,c' \in [0,1]} \frac{|\tilde{\mathbf{1}}(x,c) - \tilde{\mathbf{1}}(x',c)|}{|x - x'|} = L_\sigma/c$.

**Proof of Lem. 4.2** Using the chain rule,

$$\nabla \tilde{d}_h^\pi(s')$$

$$= \sum_{s,a} P(s'|s,a)\left(\nabla \tilde{\pi}_{h-1}(a|s)\tilde{d}_{h-1}^\pi(s) + \pi(a|s)\nabla \sigma\left(\tilde{d}_{h-1}^\pi(s), C_{h-1}^{\mathbf{s}} d_{h-1}^D(s)\right)\right)$$

$$= \sum_{s,a} P(s'|s,a)\tilde{\pi}_{h-1}(a|s)\sigma\left(\tilde{d}_{h-1}^\pi(s), C_{h-1}^{\mathbf{s}} d_{h-1}^D(s)\right)\left(\nabla \log \tilde{\pi}_{h-1}(a|s)\right.$$

$$\left. + \nabla \log \sigma\left(\tilde{d}_{h-1}^\pi(s), C_{h-1}^{\mathbf{s}} d_{h-1}^D(s)\right)\right)$$

$$= \sum_{s,a} P(s'|s,a)\pi_{h-1}^D(a|s)d_{h-1}^D(s)\tilde{\rho}_{h-1}^\pi(s,a)\left(\nabla \log \tilde{\pi}_{h-1}(a|s)\right.$$

$$\left. + \nabla \log \sigma\left(\tilde{d}_{h-1}^\pi(s), C_{h-1}^{\mathbf{s}} d_{h-1}^D(s)\right)\right),$$

where in the last line we use the definition of $\tilde{\rho}_{h-1}^\pi$ from Lem. 4.2 to make a change-of-measure. Further,

$$\nabla \log \sigma\left(\tilde{d}_{h-1}^\pi(s), C_{h-1}^{\mathbf{s}} d_{h-1}^D(s)\right) = \nabla \tilde{d}_{h-1}^\pi(s) \odot \partial_x \sigma\left(\tilde{d}_{h-1}^\pi(s), C_{h-1}^{\mathbf{s}} d_{h-1}^D(s)\right)$$

$$= \nabla \log \tilde{d}_{h-1}^\pi(s) \odot \left(\tilde{d}_{h-1}^\pi(s) \cdot \partial_x \sigma\left(\tilde{d}_{h-1}^\pi(s), C_{h-1}^{\mathbf{s}} d_{h-1}^D(s)\right)\right)$$

$$= \nabla \log \widetilde{d}^\pi_{h-1}(s) \odot \widetilde{\mathbf{1}} \left( \widetilde{d}^\pi_{h-1}(s), C^{\mathbf{s}}_{h-1} d^D_{h-1}(s) \right),$$

by definition. We can make the analogous statement for $\nabla \log \widetilde{\pi}_{h-1}$. Next, using the same change of measure in Def. 4.3, we have

$$\widetilde{d}^\pi_h(s') = \sum_{s,a} P(s'|s,a) \pi^D_{h-1}(a|s) d^D_{h-1}(s) \widetilde{\rho}^\pi_{h-1}(s,a).$$

The lemma statement follows from $\nabla \log \widetilde{d}^\pi_h(s') = \nabla \widetilde{d}^\pi_h(s') / \widetilde{d}^\pi_h(s')$, and the definition of $\mathbf{E}^{D,\widetilde{\rho}}_{h-1}$ in Eq. (5). The gradient magnitude bound is proved in Lem. C.2.

**Lemma C.2** (Bounded gradient magnitude)**.** *Suppose $\sigma$ is differentiable almost everywhere. Under Part 1 of Asm. 4.1 and Asm. 2.1,*

$$\max_{h,s} \left\| \nabla \log \widetilde{d}^\pi_h(s) \right\|_\infty \leq hG.$$

*Proof of Lem. C.2.* As a consequence of Asm. 4.1, which states that the gradient of $\sigma$ is nonincreasing in the first argument, for any $x, c \geq 0$ we have

$$\sigma(x,c) = \int_0^x \partial_x \sigma^{\mathbf{s}}(z,c)\, dz \geq \int_0^x \partial_x \sigma^{\mathbf{s}}(x,c)\, dz = x\, \partial_x \sigma(x,c)$$

Then substituting $x \leftarrow \widetilde{d}^\pi_{h-1}(s)$ and $c \leftarrow d^D_{h-1}(s)$, the above shows that $\nabla \log \sigma\left( \widetilde{d}^\pi_{h-1}(s), d^D_{h-1}(s) \right) \leq \nabla \log \widetilde{d}^\pi_{h-1}$ pointwise. Since Lem. 4.2 involves a valid (conditional) expectation, for any $s \in \mathcal{S}$ we have

$$\left\| \nabla \log \widetilde{d}^\pi_h(s') \right\|_\infty$$
$$\leq \max_{s,a} \left\{ \left\| \nabla \log \sigma\left( \pi(a|s), C^{\mathbf{a}}_{h-1} \pi^D_{h-1}(a|s) \right) \right\|_\infty + \left\| \nabla \log \sigma\left( \widetilde{d}^\pi_{h-1}(s), C^{\mathbf{s}}_{h-1} d^D_{h-1}(s) \right) \right\|_\infty \right\}$$
$$\leq G + \max_s \left\| \nabla \log \sigma\left( \widetilde{d}^\pi_{h-1}(s), C^{\mathbf{s}}_{h-1} d^D_{h-1}(s) \right) \right\|_\infty$$
$$\leq hG$$

where we use Asm. 2.1 in the second line, and unroll the same inequalities through levels in the last line. $\qquad\square$

**Proof of Prop. 4.4** First, we bound the difference between the soft-clipped and clipped density functions:

$$\left\| \widetilde{d}^\pi_h - \bar{d}^\pi_h \right\|_1 \leq \left\| \sigma\left( \widetilde{d}^\pi_{h-1}, C^{\mathbf{s}}_{h-1} d^D_{h-1} \right) - \left( \bar{d}^\pi_{h-1} \wedge C^{\mathbf{s}}_{h-1} d^D_{h-1} \right) \right\|_1 + \max_s \left\| \widetilde{\pi}_{h-1}(\cdot|s) - \bar{\pi}_{h-1}(\cdot|s) \right\|_1$$

For the second term and any $s$,

$$\left\| \widetilde{\pi}_{h-1}(\cdot|s) - \bar{\pi}_{h-1}(\cdot|s) \right\|_1 = \left\| \sigma\left( \pi_{h-1}(\cdot|s), C^{\mathbf{a}}_{h-1} \pi^D_{h-1}(\cdot|s) \right) - \left( \pi_{h-1}(\cdot|s) \wedge C^{\mathbf{a}}_{h-1} \pi^D_{h-1}(\cdot|s) \right) \right\|_1$$
$$\leq D_\sigma \left\| \left( \pi_{h-1}(\cdot|s) \wedge C^{\mathbf{a}}_{h-1} \pi^D_{h-1}(\cdot|s) \right) \right\|_1 \leq D_\sigma$$

For the first term,

$$\left\| \sigma\left( \widetilde{d}^\pi_{h-1}, C^{\mathbf{s}}_{h-1} d^D_{h-1} \right) - \left( \bar{d}^\pi_{h-1} \wedge C^{\mathbf{s}}_{h-1} d^D_{h-1} \right) \right\|_1$$
$$\leq \left\| \sigma\left( \widetilde{d}^\pi_{h-1}, C^{\mathbf{s}}_{h-1} d^D_{h-1} \right) - \left( \widetilde{d}^\pi_{h-1} \wedge C^{\mathbf{s}}_{h-1} d^D_{h-1} \right) \right\|_1 + \left\| \left( \widetilde{d}^\pi_{h-1} \wedge C^{\mathbf{s}}_{h-1} d^D_{h-1} \right) - \left( \bar{d}^\pi_{h-1} \wedge C^{\mathbf{s}}_{h-1} d^D_{h-1} \right) \right\|_1$$
$$\leq D_\sigma \left\| \left( \widetilde{d}^\pi_{h-1} \wedge C^{\mathbf{s}}_{h-1} d^D_{h-1} \right) \right\|_1 + \left\| \widetilde{d}^\pi_{h-1} - \bar{d}^\pi_{h-1} \right\|_1$$

where in the last line we use Asm. 4.1 to upper bound the first term, and the properties of the pointwise minimum $\wedge$ to upper bound the second term. Since $\left( \widetilde{d}^\pi_{h-1} \wedge C^{\mathbf{s}}_{h-1} d^D_{h-1} \right) \leq \widetilde{d}^\pi_{h-1} \leq d^\pi_{h-1}$, we have

$$\left\| \widetilde{d}^\pi_h - \bar{d}^\pi_h \right\|_1 \leq 2D_\sigma + \left\| \widetilde{d}^\pi_{h-1} - \bar{d}^\pi_{h-1} \right\|_1 \leq h(D_\sigma + D_\sigma)$$

after rolling out the induction. Then for any $\pi$,

$$\bar{J}(\pi) - \widetilde{J}(\pi) \leq \sum_{h=1}^{H} \left\| \widetilde{d}_h^\pi - \bar{d}_h^\pi \right\|_1 \leq 2H^2 D_\sigma$$

Lastly, let $\widetilde{\pi}^* = \mathrm{argmax}_{\pi \in \Pi_\Theta} \widetilde{J}(\pi)$, and define $\bar{\pi}^*$ similarly. Then

$$\bar{J}(\bar{\pi}^*) - \widetilde{J}(\widetilde{\pi}^*) \leq \bar{J}(\bar{\pi}^*) - \widetilde{J}(\bar{\pi}^*) \leq 2H^2 D_\sigma.$$

## C.3   Proofs for Sec. 4.3

First, we give an example of $\mathcal{G}$ that satisfies Asm. 4.2 in low-rank MDPs.

**Proposition C.1.** *Suppose $\mathcal{M}$ is a low-rank MDP (Def. B.1), and suppose $\mu$ is known. For each layer $h$, define the function class*

$$\mathcal{G}_h = \left\{ g_h = \frac{\mu^\top \Psi}{\mu^\top \psi} : \Psi \in \mathbb{R}^{k \times \mathsf{p}}, \psi \in \mathbb{R}^k, \|g_h\|_\infty \leq hG, \ \forall h \in [H] \right\}.$$

*Then $\{\mathcal{G}_h\}$ satisfies Asm. 4.2 and has pseudodimension (Def. F.1) $\mathsf{d}_{\mathcal{G}_h} = O(k\mathsf{p})$.*

*Proof of Prop. C.1.* It suffices to show that, for any $f : \mathcal{S} \times \mathcal{A} \to \mathbb{R}^\mathsf{p}$, reweighting function $\rho : \mathcal{S} \times \mathcal{A} \to \mathbb{R}_+$, and $h \in [H]$, the gradient update in Lem. 4.2 has $[\mathbf{E}_h^{D,\rho} f] \in \mathcal{G}_{h+1}$.

Fix $\rho$, $f$, and $h$. From the definition of $\mathbf{E}_f^\rho$, we have

$$[\mathbf{E}_h^\rho f](s') = \frac{\sum_{s,a} P(s'|s,a) \pi_h^D(a|s) d_h^D(s,a) \rho(s,a) f(s,a)}{\sum_{s,a} P(s'|s,a) \pi_h^D(a|s) d_h^D(s,a) \rho(s,a)}$$

Then since $P(s'|s,a) = \langle \phi(s,a), \mu(s') \rangle$, we can apply the same steps as the proof of Prop. B.1 to show that there exists $\Psi \in \mathbb{R}^{k \times \mathsf{p}}$ and $\psi \in \mathbb{R}^k$ such that

$$[\mathbf{E}_h^\rho f](s') = \frac{\mu(s')^\top \Psi}{\mu(s')^\top \psi}, \ \forall s' \in \mathcal{S}.$$

Specifically, $\psi = \sum_{s,a} \phi(s,a) \pi_h^D(a|s) d_h^D(s,a) \rho(s,a)$, and the $p$-th column of $\Psi$ is $\Psi^p = \sum_{s,a} \phi(s,a) \pi_h^D(a|s) d_h^D(s,a) \rho(s,a) f^p(s,a)$. $\qquad\square$

**Proof of Thm. 4.1**   For the remainder of this section, we define the constants $\varepsilon^{\mathrm{w}}$ and $\varepsilon^{\mathrm{mle}}$ to be the estimation errors of $\widetilde{w}^\pi$ and $d^D$, respectively, such that for a given $\pi$ and any $h \in [H]$ we have

$$\|\widehat{w}_h^\pi - \widetilde{w}_h^\pi\|_{1,d_{h-1}^{D,\dagger}} \leq \varepsilon^{\mathrm{w}}$$

$$\left\| \widehat{d}_h^D - d_h^D \right\|_1 \leq \varepsilon^{\mathrm{mle}} \quad \text{and} \quad \left\| \widehat{d}_h^{D,\dagger} - d_h^{D,\dagger} \right\|_1 \leq \varepsilon^{\mathrm{mle}}.$$

We can obtain such estimates using Alg. 4 and Alg. 5, and a direct application of Lem. D.1 with union bound gives $\varepsilon^{\mathrm{mle}} = O\left( \sqrt{\frac{\log(H|\mathcal{F}|/\delta)}{n}} \right)$, and similarly Thm. E.1 states that $\varepsilon^{\mathrm{wreg}} = O\left( \sqrt{\frac{\log(H|\mathcal{W}|/\delta)}{n}} \right)$.

Next, recall that

$$\widehat{\nabla} \widetilde{J}(\pi) = \frac{1}{|\mathcal{D}_h^{\mathrm{grad}}|} \sum_{h=0}^{H-1} \sum_{(s,a,s') \in \mathcal{D}_h^{\mathrm{grad}}} \widehat{w}_h^\pi(s') R_h(s') \widehat{g}_h^\pi(s').$$

The expected value over draws of $\mathcal{D}^{\mathrm{grad}}$ is

$$\mathbb{E}_{\mathcal{D}_h^{\mathrm{grad}}} \left[ \widehat{\nabla} \widetilde{J}(\pi) \right] = \sum_h \mathbb{E}_{s' \sim d_{h-1}^{D,\dagger}} \left[ \widehat{w}_h^\pi(s') R_h(s') \widehat{g}_h^\pi(s') \right].$$

First we bound the statistical error from using samples to approximate $\nabla \widetilde{J}(\pi)$, given the gradient estimate. Fix the other datasets, then

$$\left\| \widehat{\nabla} \widetilde{J}(\pi) - \mathbb{E}_{\mathcal{D}^{\text{grad}}} \left[ \widehat{\nabla} \widetilde{J}(\pi) \right] \right\|$$

$$\leq \sqrt{\mathsf{p}} \max_{p \in [\mathsf{p}]} \sum_h \left| \widehat{\mathbb{E}}_{s' \sim d_{h-1}^{D,\dagger}} [\widehat{w}_h^\pi(s') R_h(s') \widehat{g}_h^\pi(s')] - \mathbb{E}_{s' \sim d_{h-1}^{D,\dagger}} [\widehat{w}_h^\pi(s') R_h(s') \widehat{g}_h^\pi(s')] \right|$$

$$\leq \sqrt{\mathsf{p}} \left( \sum_h C_h^{\mathbf{s}} C_h^{\mathbf{a}} \right) \max_{p \in [\mathsf{p}], h \in [H]} \left| \widehat{\mathbb{E}}_{s' \sim d_{h-1}^{D,\dagger}} [\widehat{g}_h^\pi(s')] - \mathbb{E}_{s' \sim d_{h-1}^{D,\dagger}} [\widehat{g}_h^\pi(s')] \right|$$

$$\leq \sqrt{\mathsf{p}} \left( \sum_h C_h^{\mathbf{s}} C_h^{\mathbf{a}} \right) \varepsilon^{\text{stat}} \tag{12}$$

where $\varepsilon^{\text{stat}} = HG\sqrt{\frac{\log(8\mathsf{p}H/\delta)}{2n}}$ is obtained by using Hoeffding's with $\delta' = \delta/4$, since the randomness in $\widehat{w}$ and $\widehat{g}$ are fixed. Then for any $p \in [\mathsf{p}]$,

$$\left\| \mathbb{E}_{\mathcal{D}_h^{\text{grad}}} \left[ \widehat{\nabla} \widetilde{J}(\pi) \right] - \nabla \widetilde{J}(\pi) \right\|$$

$$\leq \sum_h \left\| \sum_{s'} d_{h-1}^{D,\dagger}(s') R_h(s') \left( \widehat{w}_h^\pi(s') \cdot \widehat{g}_h^\pi(s') - w_h^\pi(s') \cdot \nabla \log \widetilde{d}_h^\pi(s') \right) \right\|$$

$$\leq \sum_h \left\| \sum_{s'} d_{h-1}^{D,\dagger}(s') R_h(s') w_h^\pi(s') \left( \widehat{g}_h^\pi(s') - \nabla \log \widetilde{d}_h^\pi(s') \right) \right\| + \left\| \sum_{s'} d_{h-1}^{D,\dagger}(s') R_h(s') \widehat{y}_h^\pi(s') \left( \widehat{w}_h^\pi(s') - w_h^\pi(s') \right) \right\|$$

$$\leq \sqrt{\mathsf{p}} \sum_h \left( hG \left\| \widehat{w}_h^\pi - w_h^\pi \right\|_{1, d_{h-1}^{D,\dagger}} + \max_{p \in [\mathsf{p}]} \left\| [\widehat{g}_h^\pi]^p - \nabla \log \widetilde{d}_h^\pi \right\|_{1, \widetilde{d}_h^\pi} \right) \tag{13}$$

The first term is bounded by [Thm. E.1](). For the second term, we use the following decomposition, which is proved at the end of this section.

**Lemma C.3** (Gradient estimation error decomposition). *Let $\varepsilon^{\text{mle}}$ and $\varepsilon^{\text{w}}$ be such that for all $h \in [h]$ and $\pi \in \Pi_\Theta$, we have*

$$\|\widehat{d}_h^D - d_h^D\|_1, \|\widehat{d}_h^{D,\dagger} - d_h^{D,\dagger}\|_1 \leq \varepsilon_h^{\text{mle}} \quad \text{and} \quad \|\widehat{w}_h^\pi - \widetilde{w}_h^\pi\|_{1, d_{h-1}^{D,\dagger}} \leq \varepsilon_h^{\text{w}}.$$

*Then under [Asm. 2.1]() and [Asm. 4.1](), for any $p \in [\mathsf{p}]$, $\widehat{g}_h^\pi$ from [Alg. 2]() satisfies*

$$\left\| \widehat{g}_h^{\pi,p} - \nabla^p \log \widetilde{d}_h^\pi \right\|_{1, \widetilde{d}_h^\pi} \leq 6h C_{h-1}^{\mathbf{s}} C_{h-1}^{\mathbf{a}} L_\sigma G \, \varepsilon_{h-1}^{\text{mle}} \qquad \text{(data distribution estimation error)}$$

$$+ 3h L_\sigma G \, \varepsilon_{h-1}^{\text{w}} \qquad \text{(occupancy estimation error)}$$

$$+ \left\| \widehat{g}_h^{\pi,p} - [\mathbf{E}_{h-1}^{D,\widehat{\rho}}(\nabla \log \widetilde{\pi}_{h-1} + \widehat{g}_{h-1})]^p \right\|_{1, \widetilde{d}_h^\pi} \qquad \text{(statistical regression error)}$$

$$+ \|\widehat{g}_{h-1}^{\pi,p} - \nabla^p \log \widetilde{d}_{h-1}^\pi\|_{1, \widetilde{d}_{h-1}^\pi} \qquad \text{(recursive term)}$$

From [Lem. C.4](), we have

$$\left\| [\mathbf{E}_{h-1}^{D,\widehat{\rho}} \widehat{g}_{h-1}]^p - \widehat{g}_h^{\pi,p} \right\|_{1, \widetilde{d}_h^\pi} \leq \sqrt{2 \left( 1 + C_{h-1}^{\mathbf{s}} \varepsilon_{h-1}^{\text{mle}} \right) } \varepsilon_h^{\text{reg}} + 2hG \left( 2C_{h-1}^{\mathbf{s}} \varepsilon_{h-1}^{\text{mle}} + \varepsilon_{h-1}^{\text{w}} \right)$$

where $\varepsilon_h^{\text{reg}} = O(\sqrt{\frac{\mathsf{d}_{\mathcal{G}} C_{h-1}^{\mathbf{s}} C_{h-1}^{\mathbf{a}} h^2 G^2 \log(n\mathsf{p}H/\delta)}{n}})$. Then plugging the above into the decomposition in [Lem. C.3](), we have

$$\left\| \widehat{g}_h^p - \nabla^p \log \widetilde{d}_h^\pi \right\|_{1, \widetilde{d}_h^\pi} \leq 10 C_{h-1}^{\mathbf{s}} C_{h-1}^{\mathbf{a}} hG L_\sigma \, \varepsilon_{h-1}^{\text{mle}} + 5h G L_\sigma \, \varepsilon_{h-1}^{\text{w}}$$

$$+ \sqrt{2 \left( 1 + C_{h-1}^{\mathbf{s}} \varepsilon_{h-1}^{\text{mle}} \right)} \varepsilon_h^{\text{reg}} + \|\widehat{g}_{h-1}^p - \nabla^p \log \widetilde{d}_{h-1}^\pi\|_{1, \widetilde{d}_{h-1}^\pi}$$

Unrolling through timesteps, we have

$$\left\|\widehat{g}_h^p - \nabla^p \log \widetilde{d}_h^\pi\right\|_{1,\widetilde{d}_h^\pi} \le 10h^2 GL_\sigma \sum_{g<h} C_g^{\mathbf{s}} C_g^{\mathbf{a}} \varepsilon_g^{\mathrm{mle}} + 5h^2 GL_\sigma \sum_{g<h} \varepsilon_g^{\mathrm{w}} + \sum_{g\le h} \sqrt{2(1+C_g^{\mathbf{s}}\varepsilon_g^{\mathrm{mle}})}\,\varepsilon_g^{\mathrm{reg}}$$

$$\le 10H^2 GL_\sigma \left(\sum_h C_h^{\mathbf{s}} C_h^{\mathbf{a}}\right)\varepsilon^{\mathrm{mle}} + 5H^3 GL_\sigma \varepsilon_H^{\mathrm{w}} + \sum_h \varepsilon_h^{\mathrm{reg}}\left(\sqrt{2}+C_h^{\mathbf{s}}\varepsilon^{\mathrm{mle}}\right)$$

Since $\varepsilon_H^{\mathrm{w}} \le \left(\sum_h C_h^{\mathbf{s}} C_h^{\mathbf{a}} + 2\sum_h C_h^{\mathbf{s}}\right)\varepsilon^{\mathrm{mle}} + \sqrt{2}\left(\sum_h C_h^{\mathbf{s}} C_h^{\mathbf{a}}\right)\varepsilon^{\mathrm{wreg}}$,

$$\left\|\widehat{g}_h^p - \nabla^p \log \widetilde{d}_h^\pi\right\|_{1,\widetilde{d}_h^\pi}$$

$$\le 25H^3 GL_\sigma \left(\sum_h C_h^{\mathbf{s}} C_h^{\mathbf{a}}\right)\varepsilon^{\mathrm{mle}} + 5\sqrt{2}H^3 GL_\sigma \left(\sum_h C_h^{\mathbf{s}} C_h^{\mathbf{a}}\right)\varepsilon^{\mathrm{wreg}} + \sum_h \varepsilon_h^{\mathrm{reg}}\left(\sqrt{2}+C_h^{\mathbf{s}}\varepsilon^{\mathrm{mle}}\right)$$

Finally, combining with Eq. (12) and upper bounding $\varepsilon^{\mathrm{reg}}$ further, we have

$$\left\|\nabla \widetilde{J}(\pi) - \widehat{\nabla}\widetilde{J}(\pi)\right\|$$

$$\le \sqrt{\mathsf{p}}\left(\sum_h C_h^{\mathbf{s}} C_h^{\mathbf{a}}\right)\left(\varepsilon^{\mathrm{stat}} + 25H^3 GL_\sigma \varepsilon^{\mathrm{mle}} + 5\sqrt{2}H^3 GL_\sigma \varepsilon^{\mathrm{wreg}}\right) + \sqrt{\mathsf{p}}\left(H\sqrt{2} + \varepsilon^{\mathrm{mle}}\sum_h C_h^{\mathbf{s}}\right)\left(\max_h \varepsilon_h^{\mathrm{reg}}\right),$$

Combining inequalities and plugging in the expression for each $\varepsilon$, we have

$$\left\|\nabla \widetilde{J}(\pi) - \widehat{\nabla}\widetilde{J}(\pi)\right\| \lesssim c\sqrt{\frac{\mathsf{d}_{\mathcal{G}}\,\mathsf{p} H^6 G^2 \left(\sum_h C_h^{\mathbf{s}} C_h^{\mathbf{a}}\right)^2 L_\sigma^2 \log(|\mathcal{W}||\mathcal{F}|/\delta)}{n}}.$$

**Additional results** Lastly, we state and prove the helper lemmas used above.

*Proof of Lem. C.3.* First from Lem. C.1, the population minimizer of Eq. (10) given $\widehat{\rho}^\pi, \widehat{y}_{h-1}^\pi$ is $g_h^\pi = \zeta_h^\pi / f_h^\pi$, where

$$\zeta_h^\pi := \mathbf{E}_{h-1}^{\widehat{\rho}}\left(\nabla \log \widetilde{\pi} + \widehat{y}_{h-1}^\pi\right)$$
$$f_h^\pi := \mathbf{E}_{h-1}^{\widehat{\rho}}\left(\mathbf{1}\right)$$

Below, we use superscript $p$ to select the $p$-th parameter of a gradient object. We first separate out the statistical regression error by decomposing

$$\left\|\nabla^p \log \widetilde{d}_h^\pi - \widehat{g}_h^{\pi,p}\right\|_{1,\widetilde{d}_h^\pi} \le \|g_h^{\pi,p} - \widehat{g}_h^{\pi,p}\|_{1,\widetilde{d}_h^\pi} + \left\|\nabla^p \log \widetilde{d}_h^\pi - g_h^{\pi,p}\right\|_{1,\widetilde{d}_h^\pi}$$

The first term appears as the regression error in Lem. C.3. Since $\nabla \log \widetilde{d}_h^\pi = \nabla \widetilde{d}_h^\pi / \widetilde{d}_h^\pi$, for any $p \in [\mathsf{p}]$ we have

$$\begin{aligned}\left\|\nabla^p \log \widetilde{d}_h^\pi - g_h^{\pi,p}\right\|_{1,\widetilde{d}_h^\pi} &= \left\|\widetilde{d}_h^\pi \frac{\zeta_h^{\pi,p}}{f_h^\pi} - \widetilde{d}_h^\pi \frac{\nabla^p \widetilde{d}_h^\pi}{\widetilde{d}_h^\pi}\right\|_1 \\ &\le \left\|\left(\widetilde{d}_h^\pi - f_h^\pi\right)\frac{\zeta_h^{\pi,p}}{f_h^\pi}\right\|_1 + \left\|\zeta_h^{\pi,p} - \nabla^p \widetilde{d}_h^\pi\right\|_1 \\ &\le \|\mathcal{G}_h\|_\infty \|\widetilde{d}_h^\pi - f_h^\pi\|_1 + \|\zeta_h^{\pi,p} - \nabla^p \widetilde{d}_h^\pi\|_1 \\ &\le 2C_{h-1}^{\mathbf{s}}\|\mathcal{G}_h\|_\infty \left\|\widehat{d}_{h-1}^D - d_{h-1}^D\right\|_1 + \|\mathcal{G}_h\|_\infty \left\|\widehat{d}_{h-1}^\pi - \widetilde{d}_{h-1}^\pi\right\|_1 \\ &\quad + \|\zeta_h^{\pi,p} - \nabla^p \widetilde{d}_h^\pi\|_1 \end{aligned} \tag{14}$$

The error $\|\widetilde{d}_h^\pi - f_h^\pi\|_1$ bounded by Lem. C.5, and $\|g_h^{\pi,p}\|_\infty \le \|\mathcal{G}_h\|_\infty$ is bounded by Lem. C.2.

We will bound the second term above. Letting $y^\pi_{h-1} := \nabla \log \sigma \left( \widetilde{d}^\pi_{h-1}, C^{\mathbf{s}}_{h-1} d^D_{h-1} \right)$ be the (true) regression target and using the gradient Bellman equation for $\nabla \log \widetilde{d}^\pi_h$ in Lem. 4.2, we have

$$\|\zeta^{\pi,p}_h - \nabla^p \widetilde{d}^\pi_h\|_1$$

$$= \left\| \mathbf{E}^{\widehat{\rho}}_{h-1} \left( \nabla^p \log \widetilde{\pi} + \widehat{y}^{\pi,p}_{h-1} \right) - \mathbf{E}^{\widetilde{\rho}^\pi}_{h-1} \left( \nabla^p \log \widetilde{\pi} + y^{\pi,p}_{h-1} \right) \right\|_1$$

$$\leq \left\| \frac{\sigma\left(\widehat{d}^\pi_{h-1}, C^{\mathbf{s}}_{h-1}\widehat{d}^D_{h-1}\right)}{\widehat{d}^D_{h-1}} \frac{\widetilde{\pi}_{h-1}}{\pi^D_{h-1}} d^D_{h-1} \pi^D_{h-1} \left( \nabla^p \log \widetilde{\pi} + \widehat{y}^{\pi,p}_{h-1} \right) - \sigma\left(\widetilde{d}^\pi_{h-1}, C^{\mathbf{s}}_{h-1} d^D_{h-1}\right) \widetilde{\pi}_{h-1} \left( \nabla^p \log \widetilde{\pi} + y^{\pi,p}_{h-1} \right) \right\|_1$$

$$\leq \left\| \sigma\left(\widehat{d}^\pi_{h-1}, C^{\mathbf{s}}_{h-1}\widehat{d}^D_{h-1}\right) \left( \nabla^p \log \widetilde{\pi} + \widehat{y}^{\pi,p}_{h-1} \right) - \sigma\left(\widetilde{d}^\pi_{h-1}, C^{\mathbf{s}}_{h-1} d^D_{h-1}\right) \left( \nabla^p \log \widetilde{\pi} + y^{\pi,p}_{h-1} \right) \right\|_1$$

$$\qquad + C^{\mathbf{s}}_{h-1} \left( G + \left\| \widehat{y}^{\pi,p}_{h-1} \right\|_\infty \right) \| d^D_{h-1} - \widehat{d}^D_{h-1} \|_1$$

$$\leq \left\| \sigma\left(\widehat{d}^\pi_{h-1}, C^{\mathbf{s}}_{h-1}\widehat{d}^D_{h-1}\right) \widehat{y}^{\pi,p}_{h-1} - \sigma\left(\widetilde{d}^\pi_{h-1}, C^{\mathbf{s}}_{h-1} d^D_{h-1}\right) y^{\pi,p}_{h-1} \right\|_1$$

$$\qquad + G \left\| \sigma\left(\widehat{d}^\pi_{h-1}, C^{\mathbf{s}}_{h-1}\widehat{d}^D_{h-1}\right) - \sigma\left(\widetilde{d}^\pi_{h-1}, C^{\mathbf{s}}_{h-1} d^D_{h-1}\right) \right\|_1 + C^{\mathbf{s}}_{h-1} \left( G + \|\mathcal{G}_{h-1}\|_\infty \right) \| d^D_{h-1} - \widehat{d}^D_{h-1} \|_1$$

$$\leq \left\| \sigma\left(\widehat{d}^\pi_{h-1}, C^{\mathbf{s}}_{h-1}\widehat{d}^D_{h-1}\right) \widehat{y}^{\pi,p}_{h-1} - \sigma\left(\widetilde{d}^\pi_{h-1}, C^{\mathbf{s}}_{h-1} d^D_{h-1}\right) y^{\pi,p}_{h-1} \right\|_1$$

$$\qquad + G\|\widehat{d}^\pi_{h-1} - \widetilde{d}^\pi_{h-1}\|_1 + C^{\mathbf{s}}_{h-1} \left( 2G + \|\mathcal{G}_{h-1}\|_\infty \right) \| d^D_{h-1} - \widehat{d}^D_{h-1} \|_1 \qquad (15)$$

Now consider the first term above, where

$$\widehat{y}^\pi_{h-1} = \widehat{g}^\pi_{h-1} \odot \widetilde{\mathbf{1}}\left(\widehat{d}^\pi_{h-1}, C^{\mathbf{s}}_{h-1}\widehat{d}^D_{h-1}\right) \quad \text{and} \quad y^\pi_{h-1} = \nabla \log d^\pi_{h-1} \odot \widetilde{\mathbf{1}}\left(\widetilde{d}^\pi_{h-1}, C^{\mathbf{s}}_{h-1} d^D_{h-1}\right).$$

Then plugging this into the first line from the previous block, we have

$$\left\| \sigma\left(\widehat{d}^\pi_{h-1}, C^{\mathbf{s}}_{h-1}\widehat{d}^D_{h-1}\right) \widehat{y}^{\pi,p}_{h-1} - \sigma\left(\widetilde{d}^\pi_{h-1}, C^{\mathbf{s}}_{h-1} d^D_{h-1}\right) y^{\pi,p}_{h-1} \right\|_1$$

$$= \left\| \sigma\left(\widehat{d}^\pi_{h-1}, C^{\mathbf{s}}_{h-1}\widehat{d}^D_{h-1}\right) \widetilde{\mathbf{1}}\left(\widehat{d}^\pi_{h-1}, C^{\mathbf{s}}_{h-1}\widehat{d}^D_{h-1}\right) \widehat{g}^{\pi,p}_{h-1} - \sigma\left(\widetilde{d}^\pi_{h-1}, C^{\mathbf{s}}_{h-1} d^D_{h-1}\right) \widetilde{\mathbf{1}}\left(\widetilde{d}^\pi_{h-1}, C^{\mathbf{s}}_{h-1} d^D_{h-1}\right) \nabla^p \log d^\pi_{h-1} \right\|_1$$

$$\leq \left\| \widehat{g}^{\pi,p}_{h-1} - \nabla^p \log d^\pi_{h-1} \right\|_{1,\widetilde{d}^\pi_{h-1}}$$

$$\qquad + \|\mathcal{G}_{h-1}\|_\infty \left\| \sigma\left(\widehat{d}^\pi_{h-1}, C^{\mathbf{s}}_{h-1}\widehat{d}^D_{h-1}\right) \widetilde{\mathbf{1}}\left(\widehat{d}^\pi_{h-1}, C^{\mathbf{s}}_{h-1}\widehat{d}^D_{h-1}\right) - \sigma\left(\widetilde{d}^\pi_{h-1}, C^{\mathbf{s}}_{h-1} d^D_{h-1}\right) \widetilde{\mathbf{1}}\left(\widetilde{d}^\pi_{h-1}, C^{\mathbf{s}}_{h-1} d^D_{h-1}\right) \right\|_1,$$

$$(16)$$

where we add and subtract $\sigma\left(\widetilde{d}^\pi_{h-1}, C^{\mathbf{s}}_{h-1} d^D_{h-1}\right) \widetilde{\mathbf{1}}\left(\widetilde{d}^\pi_{h-1}, C^{\mathbf{s}}_{h-1} d^D_{h-1}\right) \widehat{g}^{\pi,p}_{h-1}$ to obtain the inequality. The first error is the recursive term, so it remains to bound the second, for which we will use the smoothness properties of $\widetilde{\mathbf{1}}(x,c)$ from Asm. 4.1.

$$\left\| \sigma\left(\widehat{d}^\pi_{h-1}, C^{\mathbf{s}}_{h-1}\widehat{d}^D_{h-1}\right) \widetilde{\mathbf{1}}\left(\widehat{d}^\pi_{h-1}, C^{\mathbf{s}}_{h-1}\widehat{d}^D_{h-1}\right) - \sigma\left(\widetilde{d}^\pi_{h-1}, C^{\mathbf{s}}_{h-1} d^D_{h-1}\right) \widetilde{\mathbf{1}}\left(\widetilde{d}^\pi_{h-1}, C^{\mathbf{s}}_{h-1} d^D_{h-1}\right) \right\|_1$$

$$\leq \left\| \sigma\left(\widetilde{d}^\pi_{h-1}, C^{\mathbf{s}}_{h-1} d^D_{h-1}\right) \left( \widetilde{\mathbf{1}}\left(\widehat{d}^\pi_{h-1}, C^{\mathbf{s}}_{h-1}\widehat{d}^D_{h-1}\right) - \widetilde{\mathbf{1}}\left(\widetilde{d}^\pi_{h-1}, C^{\mathbf{s}}_{h-1} d^D_{h-1}\right) \right) \right\|_1$$

$$\qquad + \left\| \sigma\left(\widehat{d}^\pi_{h-1}, C^{\mathbf{s}}_{h-1}\widehat{d}^D_{h-1}\right) \left( \widetilde{\mathbf{1}}\left(\widehat{d}^\pi_{h-1}, C^{\mathbf{s}}_{h-1}\widehat{d}^D_{h-1}\right) - \widetilde{\mathbf{1}}\left(\widehat{d}^\pi_{h-1}, C^{\mathbf{s}}_{h-1} d^D_{h-1}\right) \right) \right\|_1$$

$$\qquad + \left\| \left( \sigma\left(\widetilde{d}^\pi_{h-1}, C^{\mathbf{s}}_{h-1} d^D_{h-1}\right) - \sigma\left(\widehat{d}^\pi_{h-1}, C^{\mathbf{s}}_{h-1}\widehat{d}^D_{h-1}\right) \right) \widetilde{\mathbf{1}}\left(\widehat{d}^\pi_{h-1}, C^{\mathbf{s}}_{h-1} d^D_{h-1}\right) \right\|_1$$

$$\leq L_\sigma \left\| \frac{\sigma\left(\widetilde{d}^\pi_{h-1}, C^{\mathbf{s}}_{h-1} d^D_{h-1}\right)}{C^{\mathbf{s}}_{h-1} d^D_{h-1}} \left( \widehat{d}^\pi_{h-1} - \widetilde{d}^\pi_{h-1} \right) \right\|_1 + L_\sigma \left\| \frac{\sigma\left(\widehat{d}^\pi_{h-1}, C^{\mathbf{s}}_{h-1}\widehat{d}^D_{h-1}\right)}{\widehat{d}^\pi_{h-1}} \left( \widehat{d}^D_{h-1} - d^D_{h-1} \right) \right\|_1$$

$$\qquad + \left\| \left( \sigma\left(\widetilde{d}^\pi_{h-1}, C^{\mathbf{s}}_{h-1} d^D_{h-1}\right) - \sigma\left(\widehat{d}^\pi_{h-1}, C^{\mathbf{s}}_{h-1}\widehat{d}^D_{h-1}\right) \right) \right\|_1$$

$$\leq (L_\sigma + 1) \left\| \widehat{d}^\pi_{h-1} - \widetilde{d}^\pi_{h-1} \right\|_1 + \left( L_\sigma + C^{\mathbf{s}}_{h-1} \right) \left\| \widehat{d}^D_{h-1} - d^D_{h-1} \right\|_1 \qquad (17)$$

Lastly,

$$\left\| \widehat{d}^\pi_h - \widetilde{d}^\pi_h \right\|_1 \leq C^{\mathbf{s}} C^{\mathbf{a}} \left\| \widehat{d}^{D,\dagger}_{h-1} - d^{D,\dagger}_{h-1} \right\|_1 + \|\widehat{w}^\pi_h - \widetilde{w}^\pi_h\|_{1,d^{D,\dagger}_{h-1}} \qquad (18)$$

Finally, after combining Eq. (14), Eq. (15), Eq. (16), Eq. (17), and Eq. (18), we have

$$
\begin{aligned}
\left\| \nabla^p \log \widetilde{d}_h^\pi - \widehat{g}_h^{\pi,p} \right\|_{1,\widetilde{d}_h^\pi} \leq\ & \left\| \widehat{g}_{h-1}^{\pi,p} - \nabla^p \log d_{h-1}^\pi \right\|_{1,\widetilde{d}_{h-1}^\pi} \\
& + \left\| \zeta_h^{\pi,p} - \widehat{g}_h^{\pi,p} \right\|_{1,\widetilde{d}_h^\pi} \\
& + \left( G + (L_\sigma + 1)\|\mathcal{G}_{h-1}\|_\infty + \|\mathcal{G}_h\|_\infty \right)\ \left\| \widehat{w}_{h-1}^\pi - \widetilde{w}_{h-1}^\pi \right\|_{1,d_{h-1}^{D,\dagger}} \\
& + C_{h-1}^{\mathbf{s}} C_{h-1}^{\mathbf{a}} \left( G + (L_\sigma + 1)\|\mathcal{G}_{h-1}\|_\infty + \|\mathcal{G}_h\|_\infty \right) \left\| \widehat{d}_{h-1}^{D,\dagger} - d_{h-1}^{D,\dagger} \right\|_1 \\
& + \left( 2C_{h-1}^{\mathbf{s}} G + (L_\sigma + C_{h-1}^{\mathbf{s}})\|\mathcal{G}_{h-1}\|_\infty + \|\mathcal{G}_h\|_\infty \right)\ \|\widehat{d}_{h-1}^D - d_{h-1}^D\|_1
\end{aligned}
$$

Plugging in $\|\mathcal{G}_h\|_\infty \leq hG$ and consolidating terms, gives the result. $\qquad\square$

**Lemma C.4.** *With probability $\geq 1 - \delta$, for all $h \in [H]$ and a fixed $\pi$ we have*

$$
\left\| \widehat{g}_h^\pi - \mathbf{E}_{h-1}^{D,\widehat{\rho}^\pi} (\nabla \log \widetilde{\pi}_{h-1} + \widehat{g}_{h-1}^\pi) \right\|_{1,\widetilde{d}_h^\pi} \leq \sqrt{2 \left( 1 + C_{h-1}^{\mathbf{s}} \varepsilon_{h-1}^{\mathrm{mle}} \right) \varepsilon_h^{\mathrm{reg}}} + 2hG \left( 2C_{h-1}^{\mathbf{s}} \varepsilon_{h-1}^{\mathrm{mle}} + \varepsilon_{h-1}^{\mathrm{w}} \right)
\tag{19}
$$

*Proof.* Let $f_h^\pi(s') = \sum_{s,a} P(s'|s,a)\widehat{\rho}^\pi(s,a)d_{h-1}^D(s)\pi^D(a|s)$ be the data distribution reweighted by $\widehat{\rho}^\pi$. For short, we use $y_h^{\pi,p} = [\mathbf{E}_{h-1}^{D,\widehat{\rho}^\pi} (\nabla \log \widetilde{\pi}_{h-1} + \widehat{g}_{h-1}^\pi)]^p$. For any $p \in [\mathsf{p}]$,

$$
\begin{aligned}
\|\widehat{g}_h^{\pi,p} - y_h^{\pi,p}\|_{1,\widetilde{d}_h^\pi} &\leq \|\widehat{g}_h^{\pi,p} - y_h^{\pi,p}\|_{1,f_h^\pi} + \|\widehat{g}_h^{\pi,p} - y_h^{\pi,p}\|_\infty \cdot \left\| f_h^\pi - \widetilde{d}_h^\pi \right\|_1 \\
&\leq \left\| \sqrt{f_h^\pi} \right\|_2 \cdot \|\widehat{g}_h^{\pi,p} - y_h^{\pi,p}\|_{2,f_h^\pi} + 2hG \left\| f_h^\pi - \widetilde{d}_h^\pi \right\|_1,
\end{aligned}
$$

where in the second line we use Cauchy-Schwarz on the first term and Lem. C.2 on the second term. Consider the first term. One can loosely bound

$$
\begin{aligned}
\left\| \sqrt{f_h^\pi} \right\|_2^2 &= \sum_{s,a,s'} P(s'|s,a)\widehat{\rho}_{h-1}^\pi(s,a)\pi_{h-1}^D(a|s)d_{h-1}^D(s) \\
&\leq C_{h-1}^{\mathbf{s}} \sum_{s,a,s'} P(s'|s,a)\widetilde{\pi}_{h-1}(a|s)d_{h-1}^D(s) \leq C_{h-1}^{\mathbf{s}},
\end{aligned}
$$

or a get a tighter result with

$$
\begin{aligned}
\left\| \sqrt{f_h^\pi} \right\|_2^2 &= \sum_{s,a,s'} P(s'|s,a)\widehat{\rho}_{h-1}^\pi(s,a)\pi_{h-1}^D(a|s)d_{h-1}^D(s) \\
&= \sum_{s,a,s'} P(s'|s,a)\frac{\sigma\left(\widetilde{d}_{h-1}^\pi(s), C_{h-1}^{\mathbf{s}} d_{h-1}^D(s)\right)}{\widehat{d}_{h-1}^D(s)}\widetilde{\pi}_{h-1}(a|s)\left(d_{h-1}^D(s) - \widehat{d}_{h-1}^D(s)\right) \\
&\quad + \sum_{s,a,s'} P(s'|s,a)\widetilde{\pi}_{h-1}(a|s)\sigma\left(\widetilde{d}_{h-1}^\pi(s), C_{h-1}^{\mathbf{s}} d_{h-1}^D(s)\right) \\
&\leq C_{h-1}^{\mathbf{s}} \left\| d_{h-1}^D - \widehat{d}_{h-1}^D \right\|_1 + 1
\end{aligned}
$$

Next we bound $\|\widehat{g}_h^{\pi,p} - y_h^{\pi,p}\|_{2,f_h^\pi}$. Define

$$
\mathcal{L}_{h-1}^p(g; y, \rho) = \widehat{\mathbb{E}}_{(s,a,s') \sim \mathcal{D}_{h-1}} \left[ \rho(s,a) \left( g^p(s') - (\nabla \log \widetilde{\pi}_{h-1}(a|s) + y^p(s)) \right)^2 \right]
$$

to be the $p$-th parameter version of Eq. (10). Recall the regression target $\widehat{y}_{h-1}^\pi$, then we have

$$
\begin{aligned}
\|\widehat{g}_h^{\pi,p} - y_h^{\pi,p}\|_{2,f_h^\pi}^2 &= \mathbb{E}\left[ \mathcal{L}_{h-1}^p(\widehat{g}_h^\pi; \widehat{y}_{h-1}^\pi, \widehat{\rho}_{h-1}^\pi) \right] - \mathbb{E}\left[ \mathcal{L}_{h-1}^p(y_h^\pi; \widehat{y}_{h-1}^\pi, \widehat{\rho}_{h-1}^\pi) \right] \\
&\leq 2 \left( \mathcal{L}_{h-1}^p(\widehat{g}_h^\pi; \widehat{y}_{h-1}^\pi, \widehat{\rho}_{h-1}^\pi) - \mathcal{L}_{h-1}^p(y_h^\pi; \widehat{y}_{h-1}^\pi, \widehat{\rho}_{h-1}^\pi) \right) + 2\varepsilon_h^{\mathrm{reg}} \qquad \text{(Lem. F.2)}
\end{aligned}
$$

$$\leq 2\varepsilon_h^{\text{reg}} \qquad\qquad\qquad (y_h^\pi \in \mathcal{G}_h, \text{Asm. 4.2})$$

Then

$$\|\hat{y}_h^p - y_h^p\|_{1,\widetilde{d}_h^\pi} \leq \sqrt{2\left(1 + C_{h-1}^{\mathbf{s}}\left\|d_{h-1}^D - \hat{d}_{h-1}^D\right\|_1\right)\varepsilon_h^{\text{reg}}} + 2hG\left\|f_h^\pi - \widetilde{d}_h^\pi\right\|_1$$

Plugging in the bound from Lem. C.5 for $\|f_h^\pi - \widetilde{d}_h^\pi\|_1$ gives the result. $\qquad\square$

**Lemma C.5.** *For any $\pi$ and estimates $\{\hat{d}_h^\pi\}, \{\hat{d}_h^D\}$, let $\hat{\rho}^\pi$ be defined as in Alg. 2, and for any $h \in [H]$ and $s' \in \mathcal{S}$ define*

$$f_h^\pi(s') := \sum_{s,a} P(s'|s,a)\hat{\rho}^\pi(s,a)\pi_{h-1}^D(a|s)d_{h-1}^D(s),$$

*to be the next-state marginal distribution induced by reweighting $d^D$ with $\hat{\rho}$. We have*

$$\left\|f_h^\pi - \widetilde{d}^\pi\right\|_1 \leq 2C_{h-1}^{\mathbf{s}}\left\|\hat{d}_{h-1}^D - d_{h-1}^D\right\|_1 + \left\|\hat{d}_{h-1}^\pi - \widetilde{d}_{h-1}^\pi\right\|_1$$

*Proof.* Using the definition of $\hat{\rho}^\pi$, we first rewrite $f_h^\pi(s') = \sum_{s,a} P(s'|s,a)\widetilde{\pi}_{h-1}(a|s)\frac{\sigma(\hat{d}_{h-1}^\pi(s), C_{h-1}^{\mathbf{s}}\hat{d}_{h-1}^D(s))}{\hat{d}_{h-1}^D(s)}d_{h-1}^D(s)$. Then

$$\left\|f_h^\pi - \widetilde{d}_h^\pi\right\|_1$$
$$\leq \left\|\frac{\sigma\left(\hat{d}_{h-1}^\pi, C_{h-1}^{\mathbf{s}}\hat{d}_{h-1}^D\right)}{\hat{d}_{h-1}^D}d_{h-1}^D - \sigma\left(\widetilde{d}_{h-1}^\pi, C_{h-1}^{\mathbf{s}}d_{h-1}^D\right)\right\|_1$$
$$\leq \left\|\frac{\sigma\left(\hat{d}_{h-1}^\pi, C_{h-1}^{\mathbf{s}}\hat{d}_{h-1}^D\right)}{\hat{d}_{h-1}^D}\left(\hat{d}_{h-1}^D - d_{h-1}^D\right)\right\|_1 + \left\|\sigma\left(\hat{d}_{h-1}^\pi, C_{h-1}^{\mathbf{s}}\hat{d}_{h-1}^D\right) - \sigma\left(\widetilde{d}_{h-1}^\pi, C_{h-1}^{\mathbf{s}}d_{h-1}^D\right)\right\|_1$$
$$\leq 2C_{h-1}^{\mathbf{s}}\left\|\hat{d}_{h-1}^D - d_{h-1}^D\right\|_1 + \left\|\hat{d}_{h-1}^\pi - \widetilde{d}_{h-1}^\pi\right\|_1$$

where in the last inequality we use Asm. 4.1 to bound the second term. $\qquad\square$

### C.4 Local convergence of OFF-OCCUPG

We demonstrate that OFF-OCCUPG can converge to a $\varepsilon$-stationary point. In order to establish this result, we will need the guarantee in Thm. 4.1 to hold for all possible policies, i.e., $\|\widehat{\nabla}\widetilde{J}(\pi) - \nabla\widetilde{J}(\pi)\| \leq \varepsilon$ for all $\pi \in \Pi_\Theta$. This is because the fixed offline data is reused throughout the algorithm, which introduces additional correlations between iterations. In the online setting it was sufficient to simply union bound over iterations, and not functions in our function classes, because we drew fresh trajectories for each policy iterate.

Since $\mathcal{G}$ and $\Pi_\Theta$ are continuous function classes, we will start our result in terms of their covering numbers, defined below. We handle this in the simplest manner by using $\ell_\infty$ coverings, and leave a more refined analysis to future work. Def. C.1 is a covering on the clipped policy ratio over $\pi^D$. For example, the direct policy parameterization with $\pi_\theta = \theta$ has $\mathcal{N}_\infty^D(\gamma, \Pi_\Theta) \leq (\max_h C_h^{\mathbf{a}}/\gamma)^{HSA}$ (Lem. C.10). In the below definition, we overload the definition of $\overline{\pi}$ (the clipped policy in Lem. 4.1) temporarily.

**Definition C.1** (Policy ratio $\gamma$-cover)**.** Let $\overline{\Pi}_\Theta$ be an $\ell_\infty$ covering of $\Pi_\Theta$ such that for any $\pi \in \Pi_\Theta$ there exists $\overline{\pi} \in \overline{\Pi}_\Theta$ with $\|\frac{\sigma(\pi, C_h^{\mathbf{a}}\pi_h^D)}{\pi_h^D} - \frac{\sigma(\overline{\pi}_h, C_h^{\mathbf{a}}\pi_h^D)}{\pi_h^D}\|_\infty \leq \gamma$. Let $\mathcal{N}_\infty^D(\gamma, \Pi_\Theta)$ denote its minimum cardinality.

**Definition C.2** (Gradient function class $\gamma$-cover)**.** Denote $\mathcal{N}_\infty(\gamma, \mathcal{G})$ to be the $\ell_\infty$ covering number of $\{\mathcal{G}_h\}$ with resolution $\gamma$.

Next, we state the stationary convergence guarantee in terms of these function class complexities, the offline coverage coefficient determined by input clipping constants $\{C_h^{\mathbf{s}}, C_h^{\mathbf{a}}\}$, and $L_\sigma$ that represents the second-order smoothness of $\sigma$.

**Corollary C.1.** *Suppose Asm. 3.2 holds. Then under the preconditions of Thm. 4.1,*

$$\frac{1}{T} \sum_t \mathbb{E}\left[\|G^\eta(\pi^{(t)}, \nabla \widetilde{J}(\pi^{(t)}))\|^2\right] \leq \varepsilon$$

*when*

$$T = \widetilde{O}\left(\frac{\beta H}{\varepsilon}\right)$$

$$n = \widetilde{O}\left(\frac{\mathsf{p} H^6 G^2 \left(\sum_h C_h^{\mathbf{s}} C_h^{\mathbf{a}}\right)^2 L_\sigma^2 \log(\mathcal{N}_\infty(\varepsilon, \mathcal{G}) \mathcal{N}_\infty^D(\varepsilon, \Pi_\Theta) |\mathcal{W}| |\mathcal{F}|)}{\varepsilon}\right).$$

**Proof of Cor. C.1** First, we invoke a union-bounded version of the offline regression estimation guarantee in Lem. F.2. For any $\pi$, $g : \mathcal{S} \to \mathbb{R}^{\mathsf{p}}$, reweighting function $\rho : \mathcal{S} \times \mathcal{A} \to \mathbb{R}_+$, and target function $y : \mathcal{S} \to \mathbb{R}^{\mathsf{p}}$, define the $p$-th parameter squared loss for a fixed policy to be

$$\mathcal{L}_h^{\pi,p}(g; y, \rho) = \widehat{\mathbb{E}}_{(s,a,s') \in \mathcal{D}_h}\left[\left(g^p(s') - (\nabla^p \log \widetilde{\pi}_h(a|s) + y^p(s)))^2\right)\right]$$

Then from Lem. F.3, With probability at least $1 - \delta'$, for all $h \in [H], p \in [\mathsf{p}], g \in \mathcal{G}_{h+1}$, and $\rho, y$ induced by $\mathcal{F}, \mathcal{W}$ (see preconditions of Lem. F.3 for more exact statement), we have

$$\left|\mathbb{E}[\mathcal{L}_h^{\pi,p}(g; y, \rho) - \mathcal{L}_h^{\pi,p}(g_{h+1}^*; y, \rho)] - \mathcal{L}_h^{\pi,p}(g; y, \rho) - \mathcal{L}_h^{\pi,p}(g_{h+1}^*; y, \rho)\right|$$

$$\leq \frac{1}{2}\mathbb{E}[\mathcal{L}_h^{\pi,p}(g; y, \rho) - \mathcal{L}_h^{\pi,p}(g_{h+1}^*; y, \rho)] + (\varepsilon_{h+1}^{\mathrm{reg}})^2,$$

where $g_{h+1}^* = \mathbf{E}_h^{D,\rho}[\nabla \log \widetilde{\pi}_h + y_h]$ and $\varepsilon_h^{\mathrm{reg}} = O\left(\sqrt{\frac{C_{h-1}^{\mathbf{s}} C_{h-1}^{\mathbf{a}} h^2 G^2 \log(\mathcal{N}_\infty(n^{-1}, \mathcal{G}) \mathsf{p} H |\mathcal{W}_h| |\mathcal{F}_h| / \delta')}{n}}\right)$.

To complete the regression part of the analysis we need to take a union bound over the result in Lem. F.3 for all $\pi \in \Pi_\Theta$. For any $\pi \in \Pi_\Theta$, let $\overline{\pi} \in \overline{\Pi}_\Theta$ of Def. C.1 be its $\ell_\infty$ cover. We need to bound the covering approximation error $\mathcal{L}_h^{\pi,p}(g; y, \rho) - \mathcal{L}_h^{\overline{\pi},p}(g; y, \rho)$. Consider a fixed $(h, s, a, s')$ and fix the inputs $(g, \rho, y, \pi)$, for which

$$\mathcal{L}_h^{\pi,p}(g; y, \rho)[s, a, s'] - \mathcal{L}_h^{\overline{\pi},p}(g; y, \rho)[s, a, s']$$
$$= \rho(s) \frac{\sigma\left(\pi(a|s), C_h^{\mathbf{a}} \pi_h^D(a|s)\right)}{\pi_h^D(a|s)} \left(g^p(s') - 2(\nabla^p \log \widetilde{\pi}(a|s) + y^p(s)) + g_{h+1}^{*,p}(s')\right) \left(g^p(s') - g_{h+1}^{*,p}(s')\right)$$

Then

$$\left|\mathbb{E}[\mathcal{L}_h^{\pi,p}(g; y, \rho) - \mathcal{L}_h^{\overline{\pi},p}(g; y, \rho)] - (\mathcal{L}_h^{\pi,p}(g; y, \rho) - \mathcal{L}_h^{\overline{\pi},p}(g; y, \rho))\right|$$

$$\leq 8 C^{\mathbf{s}} h^2 G^2 \left\|\frac{\sigma\left(\pi, C_h^{\mathbf{a}} \pi_h^D\right)}{\pi_h^D} - \frac{\sigma\left(\overline{\pi}, C_h^{\mathbf{a}} \pi_h^D\right)}{\pi_h^D}\right\|_\infty + 8 C^{\mathbf{s}} C^{\mathbf{a}} h G \max_{s,a} \|\nabla \log \widetilde{\pi}(a|s) - \nabla \log \widetilde{[\overline{\pi}]}(a|s)\|_\infty$$

$$\leq 8(C^{\mathbf{s}} C^{\mathbf{a}} h^2 G^2 + C^{\mathbf{s}} C^{\mathbf{a}} h G L_\sigma \beta)\varepsilon$$

where we get smoothness of the gradient portion using Asm. 4.1 and Asm. 3.2. Then by setting $\varepsilon = (8(C^{\mathbf{s}} C^{\mathbf{a}} h^2 G^2 + C^{\mathbf{s}} C^{\mathbf{a}} h G L_\sigma \beta))n^{-1}$ and combining the above errors with Lem. F.3, we have that with probability at least $1 - \delta$ for all $\pi \in \Pi_\Theta$ that

$$\left|\mathbb{E}[\mathcal{L}_{\mathcal{D}_h}^p(\rho_h, g_{h+1}, y_h, \pi) - \mathcal{L}_{\mathcal{D}_h}^p(\rho_h, g_{h+1}^*, y_h, \pi)] - \mathcal{L}_{\mathcal{D}_h}^p(\rho_h, g_{h+1}, y_h, \pi) - \mathcal{L}_{\mathcal{D}_h}^p(\rho_h, g_{h+1}^*, y_h, \pi)\right|$$

$$\leq \frac{1}{2}\mathbb{E}[\mathcal{L}_{\mathcal{D}_h}^p(\rho_h, g_{h+1}, y_h, \pi) - \mathcal{L}_{\mathcal{D}_h}^p(\rho_h, g_{h+1}^*, y_h, \pi)]$$

$$+ c\sqrt{\frac{C_{h-1}^{\mathbf{s}} C_{h-1}^{\mathbf{a}} h^2 G^2 \log(\mathcal{N}_\infty^D(n^{-1}, \Pi_\Theta) \mathcal{N}_\infty(n^{-1}, \mathcal{G}) \mathsf{p} H |\mathcal{W}_h| |\mathcal{F}_h| / \delta)}{n}} := \varepsilon_h^{\mathrm{reg}},$$

for some absolute constant $c$. The remainder of the proof is identical to the proof of Thm. 4.1 using the above $\varepsilon_h^{\mathrm{reg}}$, which is then combined with Lem. G.1 to give the result.

## C.5 Global convergence of OFF-OCCUPG

We now turn our attention to analyzing gradient domination of the offline objective $\widetilde{J}(\pi)$. The preconditions of our result are written in terms of the smooth-clipped analog of the pessimistic value function $\bar{Q}^\pi$ (Prop. 4.1), induced by the smooth-clipped occupancy gradient $\nabla \widetilde{d}^\pi$. For each $(h, s, a)$, define

$$\widetilde{Q}_h^\pi(s,a) := \partial_x \sigma\left(\pi(a|s), C_h^{\mathbf{a}} \pi_h^D(a|s)\right) \sum_{s'} P(s'|s,a)\Big(R(s')$$
$$+ \partial_x \sigma\left(\widetilde{d}_{h+1}^\pi(s'), C_{h+1}^{\mathbf{s}} d_{h+1}^D(s')\right) \widetilde{Q}_{h+1}^\pi(s', \widetilde{\pi}_{h+1})\Big).$$

Lem. C.6 shows that the optimality gap of a policy for the smooth-clipped objective $\widetilde{J}(\pi)$ is bounded by a measure of its gradient magnitude, as well as a coverage coefficient. This is because our trick with exploratory $\mu$ in Lem. 3.2 isn't applicable, as it is not covered by the data in $\mathcal{D}_0$ supported on $\mathcal{S}^0$. Without this, our offline gradient domination guarantee in Lem. C.6 has a coverage coefficient that resembles the first inequality of Lem. B.3 when $\mu = d_0$, the original initial state distribution.

**Lemma C.6.** *For any $\pi$ and $\pi'$, define $\widetilde{B}^\pi(\pi') := \sum_{h,s,a} \widetilde{d}_h^\pi(s) \widetilde{\pi}'(a|s) \widetilde{Q}_h^\pi(s,a)$. Suppose that $\forall \pi \in \Pi_\Theta$,*

1. *(Policy completeness) There exists $\pi^+ \in \Pi_\Theta$ such that $\pi^+ \in \arg\max_{\pi'} \widetilde{B}^\pi(\pi')$.*
2. *(Gradient domination) $\max_{\pi' \in \Pi_\Theta} \widetilde{B}^\pi(\pi') - \widetilde{B}^\pi(\pi) \le m \max_{\theta' \in \Theta} \left\langle \nabla \widetilde{B}^\pi(\pi), \theta' - \theta \right\rangle$.*

*Then for any comparator policy $\pi^E$ and $\pi_\theta \in \Pi_\Theta$, we have*

$$\widetilde{J}(\pi^E) - \widetilde{J}(\pi_\theta) \le m \left( \max_h \left\| \frac{\sigma\left(\widetilde{d}_h^{\pi^E}, C_h^{\mathbf{s}} d_h^D\right)}{\sigma\left(\widetilde{d}_h^{\pi_\theta}, C_h^{\mathbf{s}} d_h^D\right)} \right\|_\infty \right) \max_{\theta' \in \Theta} \langle \nabla \widetilde{J}(\pi_\theta), \theta' - \theta \rangle.$$

Compared to Lem. 3.2, the first precondition of Lem. C.6 may be stronger because $\pi^+$ can be a stochastic policy, whereas deterministic policies suffice in the online setting. The second precondition is of similar strength. More importantly—as we have previously discussed— coverage coefficients of the form in Lem. C.6 are not ideal because they involve $\pi_\theta$ in the denominator, which are variable over the learning process.

**Offline data as an exploratory initialization**  Since all occupancies are clipped to some constant of the offline data, however, we might wonder if the offline data distribution itself might serve as an exploratory initial distribution to use with OFF-OCCUPG (in some sense, this, or some reweighted version of it, is the only thing available to us in the offline setting). Prop. C.2 shows that this is indeed possible when the offline data is exploratory enough, and we use clipping for simplicity. Notably, the coverage coefficient present in the gradient domination bound is the input clipping constant $\sum_h C_h^{\mathbf{s}}$. This can be seen as an offline analog of $\mathcal{C}^{\pi^*}$ in Lem. 3.2, since all occupancies are clipped to have this ratio over the offline data.

**Proposition C.2.** *Given $\{d_h^D\}$, define a new data distribution where $d_h^{D'} = \frac{1}{H} \sum_{g=0}^{H-1} d_g^D$, $\forall h \in [H]$. Then for any $\pi$, use $\{d_h^{D'}\}$, $\sigma = \wedge$, and clipping constants $\{C_h^{\mathbf{s}'}, C_h^{\mathbf{a}'}\}$ to define $[\widetilde{d}_h^\pi]'$ according to Def. 4.3. Let $\widetilde{J}'(\pi) = \sum_h \langle [\widetilde{d}_h^\pi]', R \rangle$.*

*For any $\pi$ and $\pi'$, recall $\widetilde{B}^\pi(\pi') := \sum_{h,s,a} [\widetilde{d}_h^\pi]'(s) \widetilde{\pi}'(a|s) [\widetilde{Q}^\pi]'_h(s,a)$. Suppose that $\forall \pi \in \Pi_\Theta$,*

1. *(Policy completeness) There exists $\pi^+ \in \Pi_\Theta$ such that $\pi^+ \in \arg\max_{\pi'} \widetilde{B}^\pi(\pi')$.*
2. *(Gradient domination) $\max_{\pi' \in \Pi_\Theta} \widetilde{B}^\pi(\pi') - \widetilde{B}^\pi(\pi) \le m \max_{\theta' \in \Theta} \left\langle \nabla \widetilde{B}^\pi(\pi), \theta' - \theta \right\rangle$.*

*Then if $\{C_h^{\mathbf{s}'}, C_h^{\mathbf{a}'}\}$ are such that $[\widetilde{d}_h^{\pi_\theta}]' \le C_h^{\mathbf{s}'} d_h^{D'}$, $\forall h$, for any $\pi_\theta \in \Pi_\Theta$, we have*

$$\max_\pi \widetilde{J}(\pi) - \widetilde{J}(\pi_\theta) \le m H \left( \sum_h C_h^{\mathbf{s}} \right) \max_{\theta' \in \Theta} \langle \nabla \widetilde{J}'(\pi_\theta), \theta' - \theta \rangle.$$

In practice, we can easily generate a new dataset $\mathcal{D}'$ satisfying Prop. C.2 by first splitting each $\mathcal{D}_h$ into $H$ equal parts $\{\mathcal{D}_h^i\}_{i=1}^H$, then setting $\mathcal{D}'_h = \cup_{g=1}^H \mathcal{D}_g^h$. The sample complexity of running Alg. 2 on $\mathcal{D}'$ will then scale with $\sum_h C_h^{\mathbf{s}}$, which are input parameters, instead of the coefficient in Lem. C.6, which is $\theta$-dependent and cannot be controlled. In exchange, it requires all-policy coverage w.r.t the new $[\widetilde{d^\pi}]'$, which, while strong, was insufficient for optimality in Lem. C.6. One justification (formalized in the hardness result of Prop. C.3) is that the exploratory initialization can cause policies to exceed coverage thresholds on reward-generating states, despite being covered on (the original) $d_0$. Clipping causes gradient signals to vanish, so a stationary policy might be far off-support, instead of optimal. While everything works out conceptually if $\{C_h^{\mathbf{s}}, C_h^{\mathbf{a}}\}$ are set to be high enough, it's unclear whether doing this will require exponentially large coefficients in the worst case.

Lastly, we combine the above gradient domination claims with the stationary convergence guarantee in Cor. C.1 to state the following global convergence result. Cor. C.2 is stated for $\bar{J}(\pi)$, our original offline optimization objective, and therefore takes into the account of approximating the clipping function with its smooth-clipped version.

**Corollary C.2.** *Suppose* $\widetilde{J}(\pi)$ *satisfies* Asm. 3.2. *If* Alg. 2 *with* $\mathcal{D}'$ *as defined in* Prop. C.2 *satisfies the preconditions of* Prop. C.2, *then set* $\mathrm{CC} = H\sum_h C_h^{\mathbf{s}}$. *Otherwise, define* $\mathrm{CC} = \max_{\pi \in \Pi_\Theta} \max_h \left\| \sigma\left(\widetilde{d}_h^{\pi^*}, C_h^{\mathbf{s}} d_h^D\right) / \sigma\left(\widetilde{d}_h^\pi, C_h^{\mathbf{s}} d_h^D\right) \right\|_\infty$ *and assume the preconditions of* Lem. C.6. *Then under* Def. C.1 *and the preconditions of* Thm. 4.1, Alg. 2 *satisfies*

$$\mathbb{E}\left[\frac{1}{T}\sum_t \left\{\max_\pi \bar{J}(\pi) - \bar{J}(\pi^{(t)})\right\}\right] \leq \varepsilon + 2H^2 D_\sigma$$

*when* $T = \widetilde{O}\left(\frac{B^2 m^2 (\mathrm{CC})^2 \beta H}{\varepsilon^2}\right)$ *and*

$$n = \widetilde{O}\left(\frac{B^2 m^2 (\mathrm{CC})^2 \mathsf{p} H^6 G^2 \left(\sum_h C_h^{\mathbf{s}} C_h^{\mathbf{a}}\right)^2 L_\sigma^2 \log(\mathcal{N}_\infty(\varepsilon, \mathcal{G}) \, \mathcal{N}_\infty^D(\varepsilon, \Pi_\Theta)|\mathcal{F}||\mathcal{W}|)}{\varepsilon^2}\right).$$

Though we optimize $\widetilde{J}(\pi)$, the guarantee in Cor. C.2 is with respect to our target offline objective $\bar{J}(\pi)$, which implies that the learned policy competes with the best policy fully covered by offline data. Generally $L_\sigma$ and $D_\sigma$ trade-off between ease of convergence (smoothness) and approximation error, respectively. For example, instantiating the bound with $\sigma$ from Prop. 4.3 with $b \propto \varepsilon$ results in a final $\varepsilon^{-1/4}$ rate.

## C.6 Proofs for App. C.5

**Proof of Lem. C.6** We will use superscript $h$ to refer to $s^h \in \mathcal{S}^h$, the set of states visitable at timestep $h$, and drop $C_h^{\mathbf{s}}$ below to reduce clutter. By the performance difference upper bound for the smooth-clipped objective in Lem. C.8, we have

$$\widetilde{J}(\pi^*) - \widetilde{J}(\pi_\theta) \leq \sum_{h=0}^{H-1} \sum_{s,a} \sigma\left(\widetilde{d}_h^{\pi^*}(s), d_h^D(s)\right) \left(\widetilde{\pi}^*(a|s) - \widetilde{\pi}_\theta(a|s)\right) \widetilde{Q}_h^{\pi_\theta}(s,a)$$

$$= \sum_{h=0}^{H-1} \sum_{s^h,a^h} \sigma\left(\widetilde{d}_h^{\pi^*}(s^h), d_h^D(s^h)\right) \left(\widetilde{\pi}^*(a^h|s^h) - \widetilde{\pi}_\theta(a^h|s^h)\right) \widetilde{Q}_h^{\pi_\theta}(s^h,a^h)$$

since $d_h^D(s) > 0$ only if $s \in \mathcal{S}^h$. Now define $\pi^+$ such that for any $s$, $\widetilde{\pi}^+(\cdot|s) = \mathrm{argmax}_{\pi \in \Delta(\mathcal{A})} \left\langle \widetilde{\pi}, \widetilde{Q}_h^{\pi_\theta}(s,\cdot)\right\rangle$. Then

$$\widetilde{J}(\pi^E) - \widetilde{J}(\pi_\theta)$$
$$\leq \sum_{h=0}^{H-1} \sum_{s^h,a^h} \sigma\left(\widetilde{d}_h^{\pi^E}(s^h), d_h^D(s^h)\right) \left(\widetilde{\pi}^E(a^h|s^h) - \widetilde{\pi}_\theta(a^h|s^h)\right) \widetilde{Q}_h^{\pi_\theta}(s^h,a^h)$$
$$\leq \sum_{h=0}^{H-1} \sum_{s^h,a^h} \sigma\left(\widetilde{d}_h^{\pi^E}(s^h), d_h^D(s^h)\right) \left(\widetilde{\pi}^+(a^h|s^h) - \widetilde{\pi}_\theta(a^h|s^h)\right) \widetilde{Q}_h^{\pi_\theta}(s^h,a^h) \tag{20}$$

$$\leq \max_h \left\| \frac{\sigma\left(\widetilde{d}_h^{\pi^E}, C_h^{\mathbf{s}} d_h^D\right)}{\sigma\left(\widetilde{d}_h^{\pi_\theta}, C_h^{\mathbf{s}} d_h^D\right)} \right\|_\infty \sum_{h=0}^{H-1} \sum_{s^h, a^h} \sigma\left(\widetilde{d}_h^{\pi_\theta}(s^h), d_h^D(s^h)\right)\left(\pi^+(a^h|s^h) - \pi_\theta(a^h|s^h)\right)$$
$$\cdot \widetilde{\mathbf{1}}\left(\pi_\theta(a^h|s^h), C_h^{\mathbf{a}}\pi^D(a^h|s^h)\right) \widetilde{Q}_h^{\pi_\theta}(s^h, a^h)$$

$$\leq \max_h \left\| \frac{\sigma\left(\widetilde{d}_h^{\pi^E}, C_h^{\mathbf{s}} d_h^D\right)}{\sigma\left(\widetilde{d}_h^{\pi_\theta}, C_h^{\mathbf{s}} d_h^D\right)} \right\|_\infty \sum_{h=0}^{H-1} \sum_{s^h, a^h} \sigma\left(\widetilde{d}_h^{\pi_\theta}(s^h), d_h^D(s^h)\right)\left(\widetilde{\pi}^+(a^h|s^h) - \widetilde{\pi}_\theta(a^h|s^h)\right) \widetilde{Q}_h^{\pi_\theta}(s^h, a^h)$$

$$= \max_h \left\| \frac{\sigma\left(\widetilde{d}_h^{\pi^E}, C_h^{\mathbf{s}} d_h^D\right)}{\sigma\left(\widetilde{d}_h^{\pi_\theta}, C_h^{\mathbf{s}} d_h^D\right)} \right\|_\infty \max_{\pi^+}\left(\widetilde{B}^{\pi^+}(\pi_\theta) - \widetilde{B}^{\pi_\theta}(\pi_\theta)\right)$$

$$= \max_h \left\| \frac{\sigma\left(\widetilde{d}_h^{\pi^E}, C_h^{\mathbf{s}} d_h^D\right)}{\sigma\left(\widetilde{d}_h^{\pi_\theta}, C_h^{\mathbf{s}} d_h^D\right)} \right\|_\infty \max_{\pi_{\theta'} \in \Pi_\Theta}\left(\widetilde{B}^{\pi_{\theta'}}(\pi_\theta) - \widetilde{B}^{\pi_\theta}(\pi_\theta)\right)$$

**Proof of Prop. C.2**  Under all-policy coverage, we can apply the result in Lem. 3.2, noting that $d'_0 = \frac{1}{H} d_h^D$, and $d_h^{\pi^*} \leq C_h^{\mathbf{s}} d_h^D$.

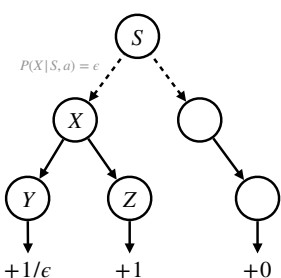

Figure 3: Example in Prop. C.3

**Proposition C.3** (Vanishing gradient from clipping with exploratory data). *Consider the MDP in App. C.6, and the data distribution where $d^D(X) = 1/2$ and $d^D(Y) = \epsilon$ and $d^D(Z) = (1-\epsilon)/2$ for some $\epsilon \in [0, 1]$. For any C, we have all-policy coverage, i.e., $d_h^\pi \leq C_h d_h^D$ for all h and all policies $\pi$. Let $\pi$ be the stationary (and in this case, optimal) policy of running Alg. 2 with $\mathcal{D}'$ described in Prop. C.2. Then*
$$J(\pi^*) - J(\pi) = (1-\epsilon)(1 - 2C_Y\epsilon).$$
*If $\epsilon$ is exponentially small, $J(\pi^*) - J(\pi) = O(1)$ unless $C_Y$ is exponentially large.*

*Proof.* The example boils down to a simple bandit problem of choosing either $L$ or $R$ in state $X$. $\pi(L|X) = \frac{C_Y d^D(Y)}{d^D(X)} = 2C_Y\epsilon$, and $\pi(R|X) = 1 - \pi(L|X)$. In comparison, $\pi^*(L|X) = 1$. Then $\widetilde{J}(\pi) = J(\pi) = \frac{C_Y d^D(Y)}{d^D(X)} + \epsilon(1 - \pi(L|X))$. In comparison, $J(\pi^*) = 1$, so

$$J(\pi^*) - J(\pi) = (1-\epsilon)(1 - 2C_Y\epsilon)$$

For reasonable choices of $C_Z$ (say, 2 or 3), $C_Y$ must be proportional to $\epsilon^{-1}$ for the suboptimality gap to shrink, and in particular if $\epsilon$ is exponentially small then $C_Y$ must be exponentially large, which blows up the RHS of the bound. □

**Proof of Cor. C.2**  The first step follows the proof of Cor. 3.2. Combining Thm. 4.1 with Lem. G.5 and plugging in above, we have

$$\mathbb{E}\left[\frac{1}{T}\sum_t \widetilde{J}(\pi^*) - \widetilde{J}(\pi^{(t)})\right]$$

$$\lesssim Bm\mathrm{CC}\left(\sqrt{\frac{\beta H}{T}} + \sqrt{\frac{\mathsf{p}H^6 G^2\left(\sum_h C_h^{\mathbf{s}}C_h^{\mathbf{a}}\right)^2 L_\sigma^2 \log(\mathcal{N}_\infty(\varepsilon,\mathcal{G})\,\mathcal{N}_\infty^D(\varepsilon,\Pi_\Theta)|\mathcal{F}||\mathcal{W}|)}{n}}\right)$$

Then we also have

$$\mathbb{E}\left[\frac{1}{T}\sum_t \bar{J}(\pi^*) - \bar{J}(\pi^{(t)})\right]$$

$$\leq \mathbb{E}\left[\frac{1}{T}\sum_t \widetilde{J}(\pi^*) - \widetilde{J}(\pi^{(t)})\right] + 2H^2 D_\sigma \qquad\qquad\text{(Prop. 4.4)}$$

$$\lesssim 2H^2 D_\sigma + Bm\mathrm{CC}\left(\sqrt{\frac{\beta H}{T}} + \sqrt{\frac{\mathsf{p}H^6 G^2\left(\sum_h C_h^{\mathbf{s}}C_h^{\mathbf{a}}\right)^2 L_\sigma^2 \log(\mathcal{N}_\infty(\varepsilon,\mathcal{G})\,\mathcal{N}_\infty^D(\varepsilon,\Pi_\Theta)|\mathcal{F}||\mathcal{W}|)}{n}}\right)$$

**Additional results**  Helper lemmas are stated and proved below.

**Lemma C.7.** *Suppose $\sigma$ satisfies Parts 1 and 2 of Asm. 4.1. Then for $\widetilde{J}(\pi) = \sum_h \sum_s \widetilde{d}_h^\pi(s) R_h(s)$,*

$$\nabla \widetilde{J}(\pi) = \sum_{h=0}^{H-1}\sum_{s,a}\sigma\left(\widetilde{d}_h^\pi(s), C_h^{\mathbf{s}}d_h^D(s)\right)\nabla\widetilde{\pi}_h(a|s)\widetilde{Q}_h^\pi(s,a),$$

*where*

$$\widetilde{Q}_h^\pi(s,a) = \sum_{s'}P_h(s'|s,a)\left(R_{h+1}(s') + \sum_{a'}\widetilde{\pi}_{h+1}(a'|s')\,\partial_x\sigma\left(\widetilde{d}_{h+1}^\pi(s'), C_{h+1}^{\mathbf{s}}d_{h+1}^D(s')\right)\widetilde{Q}_{h+1}^\pi(s',a')\right).$$

*Proof of Lem. C.7.* For notational clarity we omit $C_h^{\mathbf{s}}$ below. Expanding $\nabla\widetilde{d}_h^\pi$, we have

$$\nabla\widetilde{d}_h^\pi(s_h) = \sum_{s_{h-1},a_{h-1}}P(s_h|s_{h-1},a_{h-1})\Big(\nabla\widetilde{\pi}_{h-1}(a_{h-1}|s_{h-1})\sigma\left(\widetilde{d}_{h-1}^\pi(s_{h-1}), d_{h-1}^D(s_{h-1})\right)$$

$$+ \widetilde{\pi}_{h-1}(a_{h-1}|s_{h-1})\,\partial_x\sigma\left(\widetilde{d}_{h-1}^\pi(s_{h-1}), d_{h-1}^D(s_{h-1})\right)\nabla\widetilde{d}_{h-1}^\pi(s_{h-1})\Big)$$

$$= \sum_{g<h}\sum_{(s_{h-1},a_{h-1},\ldots,s_g,a_g)}\left[\prod_{t=g+1}^{h-1}P(s_{t+1}|s_t,a_t)\widetilde{\pi}_t(a_t|s_t)\,\partial_x\sigma\left(\widetilde{d}_t^\pi(s_t), d_t^D(s_t)\right)\right]$$

$$\cdot P(s_{g+1}|s_g,a_g)\sigma\left(\widetilde{d}_g^\pi(s_g), d_g^D(s_g)\right)\nabla\widetilde{\pi}_g(a_g|s_g)$$

For short, define

$$\widetilde{\mathbf{P}}^{\widetilde{\pi}}(s_h|s_g,a_g) := \sum_{(s_{h-1},a_{h-1},\ldots,s_{g+1},a_{g+1})}\left[\prod_{t=g+1}^{h-1}P(s_{t+1}|s_t,a_t)\widetilde{\pi}_t(a_t|s_t)\,\partial_x\sigma\left(\widetilde{d}_t^\pi(s_t), d_t^D(s_t)\right)\right]P(s_{g+1}|s_g,a_g)$$

observing if $\widetilde{\pi}_h = \pi_h$ and $\partial_x\sigma\left(\widetilde{d}_h^\pi, d_h^D\right) = 1$ for all $h$, we have $\widetilde{\mathbf{P}}^{\widetilde{\pi}}(s_h|s_g,a_g) = \mathbf{P}^\pi(s_h|s_g,a_g)$, the standard transition kernel from $(s_g,a_g) \to s_h$. This occurs, for example, when $\sigma^{\mathbf{s}}$ is hard clipping and $\pi$ is fully covered by data. Then using the above definition, we have

$$\nabla\widetilde{d}_h^\pi(s_h) = \sum_{g<h}\sum_{s_g,a_g}\sigma\left(\widetilde{d}_g^\pi(s_g), d_g^D(s_g)\right)\nabla\widetilde{\pi}_g(a_g|s_g)\widetilde{\mathbf{P}}^{\widetilde{\pi}}(s_h|s_g,a_g). \qquad(21)$$

Plugging this expression into $\nabla J(\pi)$, we obtain

$$\nabla J(\pi) = \sum_h \sum_{s_h} \nabla \widetilde{d}_h^\pi(s_h) R(s_h)$$

$$= \sum_h \sum_{s_h} \left( \sum_{g=0}^{h-1} \sum_{s_g,a_g} \sigma\left(\widetilde{d}_g^\pi(s_g), d_g^D(s_g)\right) \nabla \widetilde{\pi}_g(a_g|s_g) \widetilde{\mathbf{P}}^{\widetilde{\pi}}(s_h|s_g,a_g) \right) R(s_h)$$

$$= \sum_{g=0}^{H-1} \sum_{s_g,a_g} \sigma\left(\widetilde{d}_g^\pi(s_g), d_g^D(s_g)\right) \nabla \widetilde{\pi}_g(a_g|s_g) \left( \sum_{h=g+1}^{H} \sum_{s_h} \widetilde{\mathbf{P}}^{\widetilde{\pi}}(s_h|s_g,a_g) R(s_h) \right)$$

$$= \sum_{g=0}^{H-1} \sum_{s_g,a_g} \sigma\left(\widetilde{d}_g^\pi(s_g), d_g^D(s_g)\right) \nabla \widetilde{\pi}_g(a_g|s_g) \widetilde{Q}^\pi(s_g,a_g)$$

where we have defined

$$\widetilde{Q}_g^\pi(s_g,a_g) := \sum_{h=g+1}^{H} \sum_{s_h} \widetilde{\mathbf{P}}^{\widetilde{\pi}}(s_h|s_g,a_g) R(s_h)$$

$$= \sum_{s_{g+1}} P(s_{g+1}|s_g,a_g) \left( R(s_{g+1}) + \sum_{a_{g+1}} \widetilde{\pi}_{g+1}(a_{g+1}|s_{g+1}) \, \partial_x \sigma\left(\widetilde{d}_{g+1}^\pi(s_{g+1}), d_{g+1}^D(s_{g+1})\right) \widetilde{Q}_{g+1}^{\widetilde{\pi}}(s_{g+1},a_{g+1}) \right)$$

$\square$

**Lemma C.8.** *If $\sigma$ is concave in its first argument, for any $\pi'$ and $\pi$ we have*

$$\widetilde{J}(\pi') - \widetilde{J}(\pi) \leq \sum_{h=0}^{H-1} \sum_{s,a} \sigma\left(\widetilde{d}_h^{\pi'}(s), d_h^D(s)\right) \left(\widetilde{\pi}_h'(a|s) - \widetilde{\pi}_h(a|s)\right) \widetilde{Q}_h^\pi(s,a),$$

*where $\widetilde{Q}_h^\pi$ is defined in Lem. C.7.*

*Proof.* This statement follows straightforwardly from plugging in Lem. C.9 and rearranging, similar to the proof of Lem. C.7. $\square$

**Lemma C.9.** *If $\sigma$ is concave in its their first arguments, then for any $h$ and $\pi, \pi'$*

$$\widetilde{d}_h^{\pi'}(s') - \widetilde{d}_h^\pi(s')$$
$$\leq \sum_{g<h} \sum_{s,a} \sigma\left(\widetilde{d}_g^{\pi'}(s), d_g^D(s)\right) \left(\sigma\left(\pi_g'(a|s), \pi_g^D(a|s)\right) - \sigma\left(\pi_g(a|s), \pi_g^D(a|s)\right)\right) \widetilde{\mathbf{P}}^\pi(s_h = s'|s_g = s, a_g = a),$$

*where*

$$\widetilde{\mathbf{P}}^\pi(s_h|s_g,a_g) := \sum_{s_{h-1:g+1},a_{h-1:g+1}} \left[ \prod_{t=g+1}^{h-1} P(s_{t+1}|s_t,a_t) \widetilde{\pi}_t(a_t|s_t) \, \partial_x \sigma\left(\widetilde{d}_t^\pi(s_t), d_t^D(s_t)\right) \right] P(s_{g+1}|s_g,a_g).$$

*Proof of Lem. C.9.* Define $\pi^g = \{\pi_1', \ldots, \pi_{g-1}', \pi_g, \ldots, \pi_{H-1}\}$, i.e., a policy that starts playing $\pi$ at timestep $g$, and plays $\pi'$ for the timesteps before that.

$$\widetilde{d}_h^{\pi'}(s') - \widetilde{d}_h^\pi(s') = \widetilde{d}_h^{\pi'}(s') - \widetilde{d}_h^{\pi^{h-1}}(s') + \widetilde{d}_h^{\pi^{h-1}}(s') - \widetilde{d}_h^\pi(s')$$

For the first pair of terms, $\pi'$ and $\pi^{h-1}$ only differ the policy used to take the action $a_{h-1}$ (and both play $\pi'$ before that), thus $d_{h-1}^{\pi'} = d_{h-1}^{\pi^{h-1}}$ and

$$\widetilde{d}_h^{\pi'}(s') - \widetilde{d}_h^{\pi^{h-1}}(s')$$
$$= \sum_{s,a} P(s'|s,a) \left(\sigma\left(\pi_{h-1}'(a|s), \pi_{h-1}^D(a|s)\right) - \sigma\left(\pi_{h-1}(a|s), \pi_{h-1}^D(a|s)\right)\right) \sigma\left(\widetilde{d}_{h-1}^{\pi'}(s), d_{h-1}^D(s)\right)$$

For the second pair of terms, $\pi^{h-1}$ and $\pi$ both play $\pi$ at time $h-1$, but the former uses $\pi'$ for timesteps $1, \ldots, h-2$:

$$\widetilde{d}_h^{\pi^{h-1}}(s') - \widetilde{d}_h^{\pi}(s')$$

$$= \sum_{s,a} P(s'|s,a)\sigma\left(\pi_{h-1}(a|s), \pi_{h-1}^D(a|s)\right)\left(\sigma\left(\widetilde{d}_{h-1}^{\pi^{h-1}}(s), d_{h-1}^D(s)\right) - \sigma\left(\widetilde{d}_{h-1}^{\pi}(s), d_{h-1}^D(s)\right)\right)$$

$$= \sum_{s,a} P(s'|s,a)\sigma\left(\pi_{h-1}(a|s), \pi_{h-1}^D(a|s)\right)\left(\sigma\left(\widetilde{d}_{h-1}^{\pi'}(s), d_{h-1}^D(s)\right) - \sigma\left(\widetilde{d}_{h-1}^{\pi}(s), d_{h-1}^D(s)\right)\right)$$

$$\leq \sum_{s,a} P(s'|s,a)\sigma\left(\pi_{h-1}(a|s), \pi_{h-1}^D(a|s)\right) \partial_x\sigma\left(\widetilde{d}_{h-1}^{\pi}(s), d_{h-1}^D(s)\right)\left(\widetilde{d}_{h-1}^{\pi'}(s) - \widetilde{d}_{h-1}^{\pi}(s)\right)$$

where the last inequality above uses the concavity of $\sigma^{\mathbf{s}}$ in the first argument (recall concave functions $f$ satisfy $f(y) \leq f(x) + f'(x)(y-x)$). Combining the above two inequalities, we have the recursive relationship

$$\widetilde{d}_h^{\pi'}(s') - \widetilde{d}_h^{\pi}(s')$$

$$\leq \sum_{s,a} P(s'|s,a)\Bigg( \left(\sigma\left(\pi_{h-1}'(a|s), \pi_{h-1}^D(a|s)\right) - \sigma\left(\pi_{h-1}(a|s), \pi_{h-1}^D(a|s)\right)\right)\sigma\left(\widetilde{d}_{h-1}^{\pi'}(s), d_{h-1}^D(s)\right)$$

$$+ \sigma\left(\pi_{h-1}(a|s), \pi_{h-1}^D(a|s)\right) \partial_x\sigma\left(\widetilde{d}_{h-1}^{\pi}(s), d_{h-1}^D(s)\right)\left(\widetilde{d}_{h-1}^{\pi'}(s) - \widetilde{d}_{h-1}^{\pi}(s)\right)\Bigg)$$

Unrolling through timesteps gives the lemma statement. $\qquad \square$

**Lemma C.10.** *Let $C^{\mathbf{a}} = \max_h C_h^{\mathbf{a}}$. Suppose $\Pi_\Theta$ is the direct policy parameterization, i.e., $\pi_\theta(a|s) = \theta_{s,a}$, and $\sigma$ is such that $D_\sigma \leq C^{\mathbf{a}}$ for all $h$. Then for any $\gamma \in (0,1]$, in [Def. C.1](#) we have $\mathcal{N}_\infty^D(\gamma, \Pi_\Theta) \leq (C^{\mathbf{a}}/\gamma)^{SAH}$.*

*Proof of [Lem. C.10](#).* Typical gridding-style arguments discretize the range of $\pi(a|s)$ for each $(s,a)$. Since we are concerned with creating a cover for the policy ratio, however, a naive argument will incur $1/\min_{s,a} \pi^D(a|s)$ in the grid's cardinality. Our solution is to grid $\Pi_\Theta$ adaptively according to the magnitude of $\pi^D(a|s)$. Intuitively, we only need to grid up to the threshold

For each $(h,s,a)$, define the adaptive gridding scale to be $\gamma'_{hsa} = \gamma\pi_h^D(a|s)$. For any $\pi \in \Pi_\Theta$, set its cover $\overline{\pi}$ as follows.

$$\overline{\pi}_h(a|s) = \begin{cases} \left\lfloor \frac{\pi(a|s)}{\gamma'_{hsa}} \right\rfloor, & \text{if } \pi(a|s) \leq C_h^{\mathbf{a}}\pi_h^D(a|s), \\ C_h^{\mathbf{a}}\pi_h^D(a|s), & \text{otherwise.} \end{cases}$$

Let $\overline{\Pi}_\Theta = \{\overline{\pi} : \pi \in \Pi_\Theta\}$. Then $|\overline{\Pi}_\Theta| \leq (\max_h C_h^{\mathbf{a}}/\gamma)^{HSA}$. Further,

$$\left|\left(\pi(a|s) \wedge C_h^{\mathbf{a}}\pi_h^D(a|s)\right) - \left(\overline{\pi}_h(a|s) \wedge C_h^{\mathbf{a}}\pi_h^D(a|s)\right)\right| \leq \gamma\pi_h^D(a|s),$$

thus $\|\frac{(\pi \wedge C_h^{\mathbf{a}}\pi_h^D) - (\overline{\pi}_h \wedge C_h^{\mathbf{a}}\pi_h^D)}{\pi_h^D}\|_\infty \leq \gamma$, and applying [Lem. C.11](#) gives the result. $\qquad \square$

**Lemma C.11.** *Suppose $\overline{\Pi}_\Theta$ satisfies [Def. C.1](#) with $\sigma xc = (x \wedge c)$. Then for any $\pi \in \Pi_\Theta$, let $\overline{\pi} \in \overline{\Pi}_\Theta$ be its cover. Under [Asm. 4.1](#), we have*

$$\left\|\frac{\sigma(\pi, C\pi^D)}{\pi^D} - \frac{\sigma(\overline{\pi}, C\pi^D)}{\pi^D}\right\|_\infty \leq C(\gamma + D_\sigma).$$

*Proof of [Lem. C.11](#).* If $\pi(a|x) \leq C\pi^D(a|x)$, using the 1-Lipschitzness of $\sigma$ we have

$$|\sigma\left(\pi(a|s), C\pi^D(a|s)\right) - \sigma\left(\overline{\pi}(a|s), C\pi^D(a|s)\right)|$$

$$= |\sigma\left(\left(\pi(a|s) \wedge C\pi^D(a|s)\right), C\pi^D(a|s)\right) - \sigma\left(\overline{\pi}(a|s), C\pi^D(a|s)\right)|$$

$$\leq |\left(\pi(a|s) \wedge C\pi^D(a|s)\right) - \overline{\pi}(a|s)|$$

---

**Algorithm 4** Maximum Likelihood Estimation

---

**Input:** datasets $\{\mathcal{D}_h\}$, function class $\mathcal{F}$
1: **for** $h = 1, \ldots, H$ **do**
2:   Estimate marginal data distributions $\widehat{d}_{h-1}^D$ and $\widehat{d}_{h-1}^{D,\dagger}$ by MLE on dataset $\mathcal{D}_{h-1}$

$$\widehat{d}_{h-1}^D = \operatorname*{argmax}_{d_{h-1} \in \mathcal{F}_{h-1}} \frac{1}{|\mathcal{D}_{h-1}|} \sum_{(s,\cdot,\cdot) \in \mathcal{D}_{h-1}} \log\left(d_{h-1}(s)\right) \qquad (22)$$

$$\widehat{d}_{h-1}^{D,\dagger} = \operatorname*{argmax}_{d_h \in \mathcal{F}_h} \frac{1}{|\mathcal{D}_{h-1}|} \sum_{(\cdot,\cdot,s') \in \mathcal{D}_{h-1}} \log\left(d_h(s')\right).$$

3: **end for**
**Output:** estimated data distributions $\{\widehat{d}_h^D\}_{h \in [H]}$ and $\{\widehat{d}_h^{D,\dagger}\}_{h \in [H]}$

---

$$\leq C\gamma\pi^D(a|s).$$

If $\pi(a|x) > C\pi^D(a|x)$,

$$|\sigma\left(\pi(a|s), C\pi^D(a|s)\right) - \sigma\left(\bar{\pi}(a|s), C\pi^D(a|s)\right)| \leq C\pi^D(a|s) - \sigma\left(\bar{\pi}(a|s), C\pi^D(a|s)\right)$$
$$\leq C\pi^D(a|s) - (1 - D_\sigma)\left(\bar{\pi}(a|s) \wedge C\pi^D(a|s)\right)$$
$$\leq C(D_\sigma + \gamma)\pi^D,$$

using Asm. 4.1 in the second inequality. As a result,

$$\left| \frac{\sigma\left(\pi(a|s), C\pi^D(a|s)\right)}{\pi^D(a|s)} - \frac{\sigma\left(\bar{\pi}(a|s), C\pi^D(a|s)\right)}{\pi^D(a|s)} \right| \leq C(\gamma + D_\sigma)$$

$\square$

# D   Maximum Likelihood Estimation

Algorithm 4 displays the data distribution estimation procedure used in offline gradient estimation (Algorithm 2), which is a direct application of MLE. The general formulation of the MLE problem utilized in this paper is to estimate a probability distribution over the instance space $\mathcal{S}$. Given an i.i.d. sampled dataset $\mathcal{D} = \{s^{(i)}\}_{i=1}^n$ and a function class $\mathcal{F}$, we optimize the MLE objective of the form

$$\widehat{f} = \operatorname*{argmin}_{f \in \mathcal{F}} \frac{1}{|\mathcal{D}|} \sum_{s \in \mathcal{D}} \log\left(f(s)\right). \qquad (23)$$

We assume $\mathcal{F}$ is finite, and refer readers to [LNSJ23; HCJ23] for techniques for handling infinite function classes. The general MLE guarantee is stated below, and is a well-established result (for example, a proof can be found in Appendix E of [AKKS20]).

**Lemma D.1** (MLE guarantee). *Let $\mathcal{D} = \{s^{(i)}\}_{i=1}^n$ be a dataset, where $s^{(i)}$ are drawn i.i.d. from some fixed probability distribution $f^*$ over $\mathcal{S}$. Consider a function class $\mathcal{F}$ that satisfies: (i) $f^* \in \mathcal{F}$, and (ii) each function $f \in \mathcal{F}$ is a valid probability distribution over $\mathcal{S}$ (i.e., $f \in \Delta(\mathcal{S})$) Then with probability at least $1 - \delta$, $\widehat{f}$ from Eq. (23) has $\ell_1$ error guarantee*

$$\|\widehat{f} - f^*\|_1 \leq \sqrt{\frac{2\log(|\mathcal{F}|/\delta)}{n}}.$$

The formal guarantee of Algorithm 4 is stated below, which is a straightforward application of Lemma D.1 with union bound (over all functions in $\mathcal{F}$, and over all timesteps).

**Assumption D.1** (MLE Realizability). Suppose that $\forall h \in [h]$, we have $d_h^D, d_{h-1}^{D,\dagger} \in \mathcal{F}_h$ for $\mathcal{D}$ defined in Def. 4.1. Additionally, $f \in \Delta(\mathcal{S})$ is a valid distribution for all $f \in \mathcal{F}_h$.

**Algorithm 5** **F**itted **O**ccupancy **I**teration with Smooth Clipping

---

**Input:** policy $\pi$, datasets $\{\mathcal{D}_h\}$, function class $\mathcal{W}$, clipping thresholds $\{C_h^{\mathbf{s}}, C_h^{\mathbf{a}}\}$, data estimates $\{\widehat{d}_h^D\}$ and $\{\widehat{d}_h^{D,\dagger}\}$.
1: Initialize $\widehat{d}_0^\pi = \widehat{d}_0^D$.
2: **for** $h = 1, \ldots, H$ **do**
3:   Define $\mathcal{L}_{\mathcal{D}_{h-1}}(w_h, w_{h-1}, \widetilde{\pi}_{h-1}) := \frac{1}{|\mathcal{D}_{h-1}|} \sum_{(s,a,s') \in \mathcal{D}_{h-1}} \left( w_h(s') - w_{h-1}(s) \frac{\widetilde{\pi}_{h-1}(a|s)}{\pi_{h-1}^D(a|s)} \right)^2$,
   and estimate
$$\widehat{w}_h^\pi = \underset{w_h \in \mathcal{W}_h}{\operatorname{argmin}} \, \mathcal{L}_{\mathcal{D}_{h-1}} \left( w_h, \frac{\sigma\left(\widehat{d}_{h-1}^\pi, C_{h-1}^{\mathbf{s}} \widehat{d}_{h-1}^D\right)}{\widehat{d}_{h-1}^D}, \sigma\left(\pi_{h-1}, C_{h-1}^{\mathbf{a}} \pi_{h-1}^D\right) \right), \qquad (24)$$
4:   Set the estimate $\widehat{d}_h^\pi = \widehat{w}_h^\pi \, \widehat{d}_{h-1}^{D,\dagger}$.
5: **end for**
**Output:** estimated state occupancies $\{\widehat{w}_h^\pi\}_{h \in [H]}$.

---

**Lemma D.2.** *Suppose* $\{\mathcal{F}_h\}$ *satisfies* [Asm. D.1]. *Then with probability at least* $1 - \delta$, *for all* $h \in [H]$ *the outputs of* [Algorithm 4] *satisfy* $\left\| \widehat{d}_h^D - d_h^D \right\|_1 \leq \varepsilon^{\mathrm{mle}}$ *and* $\left\| \widehat{d}_h^{D,\dagger} - d_h^{D,\dagger} \right\|_1 \leq \varepsilon^{\mathrm{mle}}$, *where*

$$\varepsilon^{\mathrm{mle}} := \sqrt{\frac{2 \log(2H|\mathcal{F}|/\delta)}{n}}.$$

## E  Offline Density Estimation

The algorithm for offline density estimation is displayed in [Alg. 5], and is directly copied from Algorithm 1 of [HCJ23], but with two minor modifications. The first is that the densities are clipped using a function $\sigma$, that can take clipping as a special case. The second is that it outputs the learned weights instead of the learned densities. The weight function class completeness assumption is shown [Asm. E.1], and is satisfied in low-rank MDPs using linear-over-linear function classes that have pseudo-dimension bounded by MDP rank. It can be seen as a 1-dimensional version of [Asm. 4.2] where $\rho = 1$ and in that sense strictly weaker.

**Assumption E.1** (Weight function completeness). *For any* $\pi \in \Pi_\Theta$ *and* $h \in [H]$, *we have*

$$\mathbf{E}_{h-1}^{D,1} \left( \frac{\sigma\left(w \cdot f', C_{h-1}^{\mathbf{s}} f\right)}{f} \frac{\widetilde{\pi}_{h-1}}{\pi_{h-1}^D} \right) \in \mathcal{W}_h, \; \forall w \in \mathcal{W}_{h-1}, \; \forall f, f' \in \mathcal{F}_{h-1},$$

**Theorem E.1.** *Suppose* $\sigma$ *satisfies* [Asm. 4.1] *and* $\mathcal{W}$ *satisfies* [Asm. E.1]. *Let* $\{\widehat{d}_h^D\} g$ *and* $\{\widehat{d}_h^{D,\dagger}\}$ *be such that* $\forall h \in [H]$,

$$\left\| \widehat{d}_h^D - d_h^D \right\|_1 \leq \varepsilon^{\mathrm{mle}} \quad \text{and} \quad \left\| \widehat{d}_h^{D,\dagger} - d_h^{D,\dagger} \right\|_1 \leq \varepsilon^{\mathrm{mle}}.$$

*Then with probability at least* $1 - \delta$, *the outputs* $\{\widehat{w}_h^\pi\}$ *of* [Algorithm 5] *satisfy for all* $h \in [H]$

$$\|\widehat{w}_h^\pi - \widetilde{w}_h^\pi\|_{1, d_{h-1}^{D,\dagger}} \leq \left( \sum_{g < h-1} C_g^{\mathbf{s}} C_g^{\mathbf{a}} + 2 \sum_{g < h} C_g^{\mathbf{s}} \right) \varepsilon^{\mathrm{mle}} + \sqrt{2} \left( \sum_{g < h} C_g^{\mathbf{s}} C_g^{\mathbf{a}} \right) \varepsilon^{\mathrm{wreg}}, \qquad (25)$$

*where* $\varepsilon^{\mathrm{wreg}} := \sqrt{\frac{c \log(H|\mathcal{W}|/\delta)}{n_{\mathrm{reg}}}}$ *for some absolute constant c.*

**Proof of [Theorem E.1]**  We begin by stating the following decomposition on the error of $\widehat{w}_h^\pi$, which is proved at the end of this section.

**Lemma E.1.** *Suppose* $\sigma$ *satisfies* [Assumption 4.1]. *Then for any* $h \in [H]$, *the error between* $\widehat{w}_h^\pi$ *and the target* $\widetilde{w}_h^\pi = \widetilde{d}_h^\pi / d_{h-1}^{D,\dagger}$ *can be recursively decomposed as*

$$\|\widehat{w}_h^\pi - \widetilde{w}_h^\pi\|_{1, d_{h-1}^{D,\dagger}} \leq \left\| \widehat{w}_{h-1}^\pi - \widetilde{w}_{h-1}^\pi \right\|_{1, d_{h-2}^{D,\dagger}}$$

$$+ 2C_{h-1}^{\mathbf{s}} \left\| \widehat{d}_{h-1}^D - d_{h-1}^D \right\|_1 + C_{h-2}^{\mathbf{s}} C_{h-2}^{\mathbf{a}} \left\| \widehat{d}_{h-2}^{D,\dagger} - d_{h-2}^{D,\dagger} \right\|_1$$
$$+ \left\| \widehat{w}_h^\pi - \mathbf{E}_{h-1}^{\bar{\pi}} \left( d_{h-1}^D \, \omega_{h-1}^\pi \right) \right\|_{2, d_{h-1}^{D,\dagger}},$$

*where* $\omega^\pi := \frac{\sigma\left(\widehat{d}_{h-1}^\pi, C_{h-1}^{\mathbf{s}} \widehat{d}_{h-1}^D\right)}{\widehat{d}_{h-1}^D}$.

Applying Lem. E.2 with union bound over all $h$, we have

$$\left\| \widehat{w}_h^\pi - \mathbf{E}_{h-1}^{\bar{\pi}} \left( d_{h-1}^D \, \omega_{h-1}^\pi \right) \right\|_{2, d_{h-1}^{D,\dagger}}^2$$
$$= \mathbb{E}\left[ \mathcal{L}_{\mathcal{D}_{h-1}^{\mathrm{reg}}} \left( \widehat{w}_h^\pi, \, \omega_{h-1}^\pi, \overline{\pi} \right) \right] - \mathbb{E}\left[ \mathcal{L}_{\mathcal{D}_{h-1}^{\mathrm{reg}}} \left( \mathbf{E}_{h-1}^{\bar{\pi}} \left( d_{h-1}^D \, \omega_{h-1}^\pi \right), \, \omega_{h-1}^\pi, \overline{\pi} \right) \right]$$
$$\leq 2 \left( \mathcal{L}_{\mathcal{D}_{h-1}^{\mathrm{reg}}} \left( \widehat{w}_h^\pi, \, \omega_{h-1}^\pi, \overline{\pi} \right) - \mathcal{L}_{\mathcal{D}_{h-1}^{\mathrm{reg}}} \left( \mathbf{E}_{h-1}^{\bar{\pi}} \left( d_{h-1}^D \, \omega_{h-1}^\pi \right), \, \omega_{h-1}^\pi, \overline{\pi} \right) \right) + 2 \left( C_{h-1}^{\mathbf{s}} C_{h-1}^{\mathbf{a}} \right)^2 (\varepsilon^{\mathrm{wreg}})^2,$$

Then unrolling Lemma E.1, for any $h$, we have for $\varepsilon^{\mathrm{wreg}} = \frac{c \log(H|\mathcal{W}|n/\delta)}{n_{\mathrm{reg}}}$,

$$\left\| \widehat{w}_h^\pi - \widetilde{w}_h^\pi \right\|_{1, d_{h-1}^{D,\dagger}} \leq \left( \sum_{g<h-1} C_g^{\mathbf{s}} C_g^{\mathbf{a}} + 2 \sum_{g<h} C_g^{\mathbf{s}} \right) \varepsilon^{\mathrm{mle}} + \sqrt{2} \left( \sum_{g<h} C_g^{\mathbf{s}} C_g^{\mathbf{a}} \right) \varepsilon^{\mathrm{wreg}}$$

Lastly, we state and prove the intermediate results below.

**Lemma E.2** (Deviation bound for regression with squared loss from [HCJ23])**.** *If* $\{\mathcal{W}_h\}$ *satisfies Asm. E.1, then with probability* $\geq 1 - \delta$, *for any* $h \in [H]$, *there exists a universal constant* $c$ *such that*

$$\left| \mathbb{E}\left[ \mathcal{L}_{\mathcal{D}_h^{\mathrm{reg}}} (w_{h+1}, w_h, \pi) - \mathcal{L}_{\mathcal{D}_h^{\mathrm{reg}}}(\mathbf{E}^{\bar{\pi}} w_h, w_h, \pi) \right] - \left( \mathcal{L}_{\mathcal{D}_h^{\mathrm{reg}}} (w_{h+1}, w_h, \pi) - \mathcal{L}_{\mathcal{D}_h^{\mathrm{reg}}}(\mathbf{E}^{\bar{\pi}} w_h, w_h, \pi) \right) \right|$$
$$\leq \frac{1}{2} \mathbb{E}\left[ \mathcal{L}_{\mathcal{D}_h^{\mathrm{reg}}} (w_{h+1}, w_h, \pi) - \mathcal{L}_{\mathcal{D}_h^{\mathrm{reg}}}(\mathbf{E}_h^{\bar{\pi}} w_h, w_h, \pi) \right] + \frac{c(C_h^{\mathbf{s}} C_h^{\mathbf{a}})^2 \log\left( H|\mathcal{W}|/\delta \right)}{n_{\mathrm{reg}}}.$$

*Proof of Lemma E.1.* Decompose

$$\left\| \widehat{w}_h^\pi - \widetilde{w}_h^\pi \right\|_{1, d_{h-1}^{D,\dagger}}$$
$$\leq \left\| \widehat{w}_h^\pi - \mathbf{E}_{h-1}^{\bar{\pi}} \left( d_{h-1}^D \frac{\sigma\left(\widehat{d}_{h-1}^\pi, C_{h-1}^{\mathbf{s}} \widehat{d}_{h-1}^D\right)}{\widehat{d}_{h-1}^D} \right) \right\|_{2, d_{h-1}^{D,\dagger}} + \left\| \widetilde{w}_h^\pi - \mathbf{E}_{h-1}^{\bar{\pi}} \left( d_{h-1}^D \frac{\sigma\left(\widehat{d}_{h-1}^\pi, C_{h-1}^{\mathbf{s}} \widehat{d}_{h-1}^D\right)}{\widehat{d}_{h-1}^D} \right) \right\|_{1, d_{h-1}^{D,\dagger}}$$

The first term is the statistical error of regression. The second term reflects the bias between the population regression solution (involving plug-in estimates for the regression target) and our target weight function. Since $\widetilde{d}_h^\pi = \mathbf{P}_{h-1} \left( \sigma\left(\widetilde{d}_{h-1}^\pi, C_{h-1}^{\mathbf{s}} d_{h-1}^D\right) \right)$, then $\widetilde{w}_h^\pi = \mathbf{E}_{h-1} \left( \sigma\left(\widetilde{d}_{h-1}^\pi, C_{h-1}^{\mathbf{s}} d_{h-1}^D\right) \right)$, for the second term we have

$$\left\| \widetilde{w}_h^\pi - \mathbf{E}_{h-1}^{\bar{\pi}} \left( d_{h-1}^D \frac{\sigma\left(\widehat{d}_{h-1}^\pi, C_{h-1}^{\mathbf{s}} \widehat{d}_{h-1}^D\right)}{\widehat{d}_{h-1}^D} \right) \right\|_{1, d_{h-1}^{D,\dagger}}$$
$$= \left\| \mathbf{E}_{h-1} \left( \sigma\left(\widetilde{d}_{h-1}^\pi, C_{h-1}^{\mathbf{s}} d_{h-1}^D\right) \right) - \mathbf{E}_{h-1}^{\bar{\pi}} \left( d_{h-1}^D \frac{\sigma\left(\widehat{d}_{h-1}^\pi, C_{h-1}^{\mathbf{s}} \widehat{d}_{h-1}^D\right)}{\widehat{d}_{h-1}^D} \right) \right\|_{1, d_{h-1}^{D,\dagger}}$$
$$= \left\| \mathbf{P}_{h-1} \left( \sigma\left(\widetilde{d}_{h-1}^\pi, C_{h-1}^{\mathbf{s}} d_{h-1}^D\right) \right) - \mathbf{P}_{h-1}^{\bar{\pi}} \left( d_{h-1}^D \frac{\sigma\left(\widehat{d}_{h-1}^\pi, C_{h-1}^{\mathbf{s}} \widehat{d}_{h-1}^D\right)}{\widehat{d}_{h-1}^D} \right) \right\|_1$$
$$\leq \left\| \sigma\left(\widetilde{d}_{h-1}^\pi, C_{h-1}^{\mathbf{s}} d_{h-1}^D\right) - d_{h-1}^D \frac{\sigma\left(\widehat{d}_{h-1}^\pi, C_{h-1}^{\mathbf{s}} \widehat{d}_{h-1}^D\right)}{\widehat{d}_{h-1}^D} \right\|_1$$

$$\leq C_{h-1}^{\mathbf{s}} \left\| \widehat{d}_{h-1}^D - d_{h-1}^D \right\|_1 + \left\| \sigma\left( \widetilde{d}_{h-1}^\pi, C_{h-1}^{\mathbf{s}} d_{h-1}^D \right) - \sigma\left( \widehat{d}_{h-1}^\pi, C_{h-1}^{\mathbf{s}} \widehat{d}_{h-1}^D \right) \right\|_1$$

$$\leq 2 C_{h-1}^{\mathbf{s}} \left\| \widehat{d}_{h-1}^D - d_{h-1}^D \right\|_1 + \left\| \widetilde{d}_{h-1}^\pi - \widehat{d}_{h-1}^\pi \right\|_1 \qquad \text{(Assumption 4.1)}$$

Finally, since $\widehat{d}_{h-1}^\pi = \widehat{w}_{h-1}^\pi \widehat{d}_{h-2}^{D,\dagger}$,

$$\left\| \widetilde{d}_{h-1}^\pi - \widehat{d}_{h-1}^\pi \right\|_1 = \left\| \widetilde{w}_{h-1}^\pi d_{h-2}^{D,\dagger} - \widehat{w}_{h-1}^\pi \widehat{d}_{h-2}^{D,\dagger} \right\|_1$$

$$\leq C_{h-2}^{\mathbf{s}} C_{h-2}^{\mathbf{a}} \left\| d_{h-2}^{D,\dagger} - \widehat{d}_{h-2}^{D,\dagger} \right\|_1 + \left\| \widetilde{w}_{h-1}^\pi - \widehat{w}_{h-1}^\pi \right\|_{1, d_{h-2}^{D,\dagger}}$$

Combining the inequalities completes the proof. $\qquad \square$

## F  Probabilistic Tools

**Definition F.1** (Pseudodimension). Suppose a function class $\mathcal{F} \subseteq \mathbb{R}^{\mathcal{X}}$, and $x_1^n = \{x_i\}_{i=1}^n \in \mathcal{X}^n$. We say $x_1^n$ is pseudo-shattered by $\mathcal{F}$ if there exists $v \in \mathbb{R}^n$ such that for all $y \in \{-1, +1\}^n$, there exists $f \in \mathcal{F}$ such that $\text{sign}(f(x_1^n - c)) = y$. The pseudo-dimension of $\mathcal{F}$ is defined as

$$\mathsf{d}_\mathcal{F} = \max\{ n \in \mathbb{N} : \exists x_1^n \in \mathcal{X}^n \text{ s.t. } x_1^n \text{ is pseudo-shattered by } \mathcal{F} \},$$

i.e., the cardinality of the largest set of points in $\mathcal{X}$ that $\mathcal{F}$ pseudo-shatters.

**Lemma F.1** (Lemma 26 from [HCJ23]). *For $b \geq 1$, let $\mathcal{H} \subseteq (\mathcal{Z} \to [-b, b])$ be a hypothesis class and $Z^n = (z_1, \ldots, z_n) \in \mathcal{Z}^n$, where $z_i$ are iid samples drawn from a distribution supported on $\mathcal{Z}$. Then for any $h \in \mathcal{H}$, we have*

$$\mathbb{P}\left( \left| \mathbb{E}[h(z)] - \frac{1}{n} \sum_i h(z_i) \right| > \varepsilon \right) \leq 36 \mathcal{N}_1\left( \frac{\varepsilon^3}{640b}, \mathcal{H}, \frac{40nb^2}{\varepsilon^2} \right) \exp\left( -\frac{n\varepsilon^2}{128 \mathbb{V}[h(z)] + 512 \varepsilon b} \right)$$

**Lemma F.2.** *Fix $\pi$. For any $h \in [H]$, consider functions $y_h : \mathcal{S} \times \mathcal{A} \to [-hG, hG]^{\mathsf{p}}$ and $\rho_h : \mathcal{S} \times \mathcal{A} \to [0, C_h^{\mathbf{s}} C_h^{\mathbf{a}}]$ that depend only on the datasets $\mathcal{D}_{<h}^{\text{mle}}$ and $\mathcal{D}_{<h}^{\text{FORC}}$ and $\mathcal{D}_{<h}^{\text{grad}}$. Let $\mathcal{G} = \{\mathcal{G}_h\}$ be function classes and with pseudo-dimension $\mathsf{d}_\mathcal{G}$ (Def. F.1). For any $g_{h+1} \in \mathcal{G}_{h+1}$ and $p \in [\mathsf{p}]$, define the loss function*

$$\mathcal{L}_h^{\pi,p}(g_{h+1}; y_h, \rho_h) = \frac{1}{n} \sum_{(s,a,s') \in \mathcal{D}_h^{\text{reg}}} \rho_h(s,a) \left( g_{h+1}^p(s') - (\nabla \log \widetilde{\pi}_h(a|s) + y_h^p(s,a)) \right)^2.$$

*Then with probability at least $1 - \delta$, for all $g_{h+1} \in \mathcal{G}_{h+1}$ and $p \in [\mathsf{p}]$ and $h \in [H]$, we have*

$$\left| \mathbb{E}[\mathcal{L}_h^{\pi,p}(g_{h+1}; y_h, \rho_h) - \mathcal{L}_h^{\pi,p}(g_{h+1}^*; y_h, \rho_h)] - \mathcal{L}_h^{\pi,p}(g_{h+1}; y_h, \rho_h) - \mathcal{L}_h^{\pi,p}(g_{h+1}^*; y_h, \rho_h) \right|$$

$$\leq \frac{1}{2} \mathbb{E}[\mathcal{L}_h^{\pi,p}(g_{h+1}; y_h, \rho_h) - \mathcal{L}_h^{\pi,p}(g_{h+1}^*; y_h, \rho_h)] + (\varepsilon_{h+1}^{\text{reg}})^2,$$

*where $g_{h+1}^* = \mathbf{E}_h^{D,\rho}(\nabla \log \widetilde{\pi}_h + y_h)$ and for some absolute constant $c$,*

$$\varepsilon_h^{\text{reg}} = c \sqrt{ \frac{ \mathsf{d}_\mathcal{G} C_{h-1}^{\mathbf{s}} C_{h-1}^{\mathbf{a}} h^2 G^2 \log(n\mathsf{p}H/\delta) }{n} }. \qquad (26)$$

*Proof.* Fix $\mathcal{D}_{<h}^{\text{mle}}$ and $\mathcal{D}_{<h}^{\text{FORC}}$ and $\mathcal{D}_{<h}^{\text{grad}}$. We first prove the stated bound conditioned on these datasets, which means $g_h^\pi$ and $\rho_h^\pi$ are fixed, and the randomness below comes from random draws of $\mathcal{D}_h^{\text{grad}}$. Consider the following hypothesis class induced by $\mathcal{G}_{h+1}$:

$$\mathcal{Z}(\rho_h, g_{h+1}, y_h) = \left\{ \rho_h(s,a) \left( (g_{h+1}(s') - y_h(s,a))^2 - (g_{h+1}^*(s') - y_h(s,a))^2 \right) : g_{h+1} \in \mathcal{G}_{h+1} \right\}.$$

and for any $Z \in \mathcal{Z}$, we have $|Z| \leq 2 C_h^{\mathbf{s}} C_h^{\mathbf{a}} \left( \|g_{h+1}\|_\infty^2 + \|y_h\|_\infty^2 \right) \leq 4 C_h^{\mathbf{s}} C_h^{\mathbf{a}} h^2 G^2$. We also have $\mathbb{E}[Z^p(\rho, g_{h+1}, y_h)] = \left\| y_{h+1}^p - g_{h+1}^{*,p} \right\|_{2, f_h^\pi}^2$. Further,

$$\mathbb{V}\left[ Z^p(\rho_h, g_{h+1}, y_h) \right] \leq \mathbb{E}[Z^p(\rho_h, g_{h+1}, y_h)^2]$$

$$= \mathbb{E}\left[\rho_h(s,a)^2 \left(\left(g_{h+1}^p(s') - y_h(s,a)\right)^2 - \left(g_{h+1}^{*,p}(s') - y_h(s,a)\right)^2\right)^2\right]$$

$$= \mathbb{E}\left[\rho_h(s,a)^2 \left(g_{h+1}^p(s') - 2y_h(s,a) + g_{h+1}^{*,p}(s')\right)^2 \left(g_{h+1}^p(s') - g_{h+1}^{*,p}(s')\right)^2\right]$$

$$\leq 16 C_h^{\mathbf{s}} C_h^{\mathbf{a}} h^2 G^2 \mathbb{E}\left[\rho_h(s,a)\left(g_{h+1}^p(s') - g_{h+1}^{*,p}(s')\right)^2\right]$$

$$= 16 C_h^{\mathbf{s}} C_h^{\mathbf{a}} h^2 G^2 \mathbb{E}[Z^p(\rho_h, g_{h+1}, y_h)]$$

Next, we show that the uniform covering number of $\mathcal{Z}$ can be bounded by the uniform covering number of $\mathcal{G}$, since for any $g_{h+1}, g'_{h+1} \in \mathcal{G}_{h+1}$ we have

$$\left|Z^p(\rho_h, g_{h+1}, y_h) - Z^p(\rho_h, g'_{h+1}, y_h)\right| = \rho_h(s,a)\left|(g_{h+1}(s') - y_h(s,a))^2 - (g'_{h+1}(s') - y_h(s,a))^2\right|$$
$$\leq 16 C_h^{\mathbf{s}} C_h^{\mathbf{a}} (h+1)^2 G^2 |g_{h+1}(s') - g'_{h+1}(s')|$$

In other words, any $\gamma/16 C_{h-1}^{\mathbf{s}} C_{h-1}^{\mathbf{a}} h^2 G^2$ covering of $\mathcal{G}_h$ is a covering of $\mathcal{Z}_h$. Then combining the above with [Lem. F.1](), we have

$$\mathbb{P}\left(\left|\mathbb{E}[Z^p(\rho_{h-1}, g_h, y_{h-1})] - \frac{1}{n}\sum_i Z_i^p(\rho_{h-1}, g_h, y_{h-1})\right| > \varepsilon\right)$$

$$\leq 36 \mathcal{N}_1\left(\frac{\varepsilon^3}{10240 C_h^{\mathbf{s}} C_h^{\mathbf{a}} h^2 G^2}, \mathcal{Z}(\mathcal{G}_h, \rho_{h-1}, y_{h-1}), \frac{640n(C_{h-1}^{\mathbf{s}} C_{h-1}^{\mathbf{a}})^2 h^4 G^4}{\varepsilon^2}\right)$$

$$\cdot \exp\left(-\frac{n\varepsilon^2}{2048 C_{h-1}^{\mathbf{s}} C_{h-1}^{\mathbf{a}} h^2 G^2 \mathbb{E}[Z^p(\rho_{h-1}, g_h, y_{h-1}, \pi)] + 2048\varepsilon C_{h-1}^{\mathbf{s}} C_{h-1}^{\mathbf{a}} h^2 G^2}\right)$$

$$\leq 36 \mathcal{N}_1\left(\frac{\varepsilon^3}{163840 (C_h^{\mathbf{s}} C_h^{\mathbf{a}})^2 h^4 G^4}, \mathcal{G}_h, \frac{640n(C_{h-1}^{\mathbf{s}} C_{h-1}^{\mathbf{a}})^2 h^4 G^4}{\varepsilon^2}\right)$$

$$\cdot \exp\left(-\frac{n\varepsilon^2}{2048 C_{h-1}^{\mathbf{s}} C_{h-1}^{\mathbf{a}} h^2 G^2 \mathbb{E}[Z^p(\rho_{h-1}, g_h, y_{h-1})] + 2048\varepsilon C_{h-1}^{\mathbf{s}} C_{h-1}^{\mathbf{a}} h^2 G^2}\right)$$

Define $N := \mathcal{N}_1\left(\frac{\varepsilon^3}{163840 (C_h^{\mathbf{s}} C_h^{\mathbf{a}})^2 h^4 G^4}, \mathcal{G}_h, \frac{640n(C_{h-1}^{\mathbf{s}} C_{h-1}^{\mathbf{a}})^2 h^4 G^4}{\varepsilon^2}\right)$. Then setting the RHS equal to $\delta'$, this implies that

$$n = \frac{2048 C_{h-1}^{\mathbf{s}} C_{h-1}^{\mathbf{a}} h^2 G^2 \left(\mathbb{E}[Z^p(\rho_{h-1}, g_h, y_{h-1})] + \varepsilon\right) \log\left(36N/\delta'\right)}{\varepsilon^2}$$

and

$$\varepsilon \leq \sqrt{\frac{2048 C_{h-1}^{\mathbf{s}} C_{h-1}^{\mathbf{a}} h^2 G^2 \mathbb{E}[Z^p(\rho_{h-1}, g_h, y_{h-1})] \log(36N/\delta')}{n}} + \frac{2048 C_{h-1}^{\mathbf{s}} C_{h-1}^{\mathbf{a}} h^2 G^2 \log(36N/\delta')}{n}.$$

Since $n \geq \frac{2048 C_{h-1}^{\mathbf{s}} C_{h-1}^{\mathbf{a}} h^2 G^2}{\varepsilon}$, there exists an absolute constant $c$ such that $\log(36N/\delta') \leq c \mathrm{d}_{\mathcal{G}} \log(n/\delta')$. Then with probability at least $1 - \delta'$,

$$\left|\mathbb{E}[Z^p(\rho_{h-1}, g_h, y_{h-1})] - \frac{1}{n}\sum_i Z_i^p(\rho_{h-1}, g_h, y_{h-1})\right|$$

$$\leq \sqrt{\frac{2048 c\mathrm{d}_{\mathcal{G}_h} C_{h-1}^{\mathbf{s}} C_{h-1}^{\mathbf{a}} h^2 G^2 \mathbb{E}[Z^p(\rho_{h-1}, g_h, y_{h-1})] \log(n/\delta')}{n}} + \frac{2048 c\mathrm{d}_{\mathcal{G}} C_{h-1}^{\mathbf{s}} C_{h-1}^{\mathbf{a}} h^2 G^2 \log(n/\delta')}{n}$$

$$\leq \frac{1}{2}\mathbb{E}[Z^p(\rho_{h-1}, g_h, y_{h-1})] + \frac{3072 c\mathrm{d}_{\mathcal{G}} C_{h-1}^{\mathbf{s}} C_{h-1}^{\mathbf{a}} h^2 G^2 \log(n/\delta')}{n}$$

Since the above bound holds for a fixed datasets, it also holds for the expectation over the datasets. Applying the above bound for all $p \in [\mathsf{p}]$ and $h \in [H]$ with $\delta' = \delta/H\mathsf{p}$ and taking the union bound, then plugging in the definition of $Z^p$, gives the result. $\qquad\square$

**Lemma F.3.** *Fix $h$ and denote the product class composed from $\mathcal{G}, \mathcal{W}, \mathcal{F}$ to be*

$$\mathcal{Y}_h \times \mathcal{P}_h = \left\{(y, \rho) : y = g \odot \tilde{\mathbf{1}}\left(wf', C_h^{\mathbf{s}} f\right), \rho = \frac{\sigma\left(wf', C_h^{\mathbf{s}} f\right)}{f}, g_h \in \mathcal{G}_h, w \in \mathcal{W}_h, f \in \mathcal{F}_{h+1}, f' \in \mathcal{F}_h\right\}.$$

*Fix $\pi$ and $p \in [\mathsf{p}]$, and define the loss function*

$$\mathcal{L}_h^{\pi,p}(g;y,\rho) = \frac{1}{n} \sum_{(s,a,s')\in\mathcal{D}_h} \rho(s) \frac{\sigma\left(\pi(a|s), C_h^{\mathbf{a}}\pi_h^D(a|s)\right)}{\pi_h^D(a|s)} \left(g^p(s') - (\nabla^p \log \widetilde{\pi}(a|s) + y^p(s))\right)^2.$$

*Then with probability at least $1 - \delta$, for all $h \in [H], p \in [\mathsf{p}]$ and $g \in \mathcal{G}_{h+1}, (y,\rho) \in \mathcal{Y}_h \times \mathcal{P}_h$, we have*

$$\left| \mathbb{E}[\mathcal{L}_h^{\pi,p}(g;y,\rho) - \mathcal{L}_h^{\pi,p}(g_{h+1}^*;y,\rho)] - \mathcal{L}_h^{\pi,p}(g;y,\rho) - \mathcal{L}_h^{\pi,p}(g_{h+1}^*;y,\rho)\right|$$

$$\leq \frac{1}{2}\mathbb{E}[\mathcal{L}_h^{\pi,p}(g;y,\rho) - \mathcal{L}_h^{\pi,p}(g_{h+1}^*;y,\rho)] + (\varepsilon_{h+1}^{\mathrm{reg}})^2,$$

*where $g_{h+1}^* = \mathbf{E}_h^{D,\rho}(\nabla \log \widetilde{\pi}_h + y_h)$ and $\varepsilon_h^{\mathrm{reg}} = O\left(\sqrt{\frac{C_{h-1}^{\mathbf{s}} C_{h-1}^{\mathbf{a}} h^2 G^2 \log(\mathcal{N}_\infty(n^{-1},\mathcal{G})\mathsf{p}H|\mathcal{W}||\mathcal{F}|/\delta)}{n}}\right).$*

*Proof of [Lem. F.3](). Use the same notation for $Z^p$ as in the proof of [Lem. F.2](). Using Bernstein's inequality with a union bound over all $\mathcal{W}_h$ and $\mathcal{F}_h$ and the $\ell_\infty$ covers of $\mathcal{G}_h, \mathcal{G}_{h+1}$, with probability at least $1 - \delta$ we have*

$$\left| \mathbb{E}[Z^p(\rho_h, g_{h+1}, y_h)] - \frac{1}{n}\sum_{i=1}^n Z_i^p(\rho_h, g_{h+1}, y_h) \right|$$

$$\leq \sqrt{\frac{2\mathbb{V}\left[Z^p(\rho_h, y_{h+1}, g_h)\right]\log(|\mathcal{W}||\mathcal{F}|\mathcal{N}_\infty(\varepsilon,\mathcal{G})/\delta)}{n}} + \frac{4C_h^{\mathbf{s}} C_h^{\mathbf{a}} h^2 G^2 \log(|\mathcal{W}||\mathcal{F}|\mathcal{N}_\infty(\varepsilon,\mathcal{G})/\delta)}{3n}$$

$$\leq \sqrt{\frac{32C_h^{\mathbf{s}} C_h^{\mathbf{a}} h^2 G^2 \mathbb{E}[Z^p(\rho_h, y_{h+1}, g_h)]\log(|\mathcal{W}||\mathcal{F}|\mathcal{N}_\infty(\varepsilon,\mathcal{G})/\delta)}{n}} + \frac{4C_h^{\mathbf{s}} C_h^{\mathbf{a}} h^2 G^2 \log(\mathcal{N}_\infty(\varepsilon,\mathcal{G})|\mathcal{W}||\mathcal{F}|/\delta)}{3n}$$

By accounting for the $\ell_\infty$ covering error, we then have

$$\left| \mathbb{E}[Z^p(\rho_h, g_{h+1}, y_h)] - \frac{1}{n}\sum_{i=1}^n Z_i^p(\rho_h, g_{h+1}, y_h) \right|$$

$$\leq \sqrt{\frac{32C_h^{\mathbf{s}} C_h^{\mathbf{a}} h^2 G^2 \mathbb{E}[Z^p(\rho_h, g_{h+1}, y_h)]\log(|\mathcal{W}||\mathcal{F}|\mathcal{N}_\infty(\varepsilon,\mathcal{G})/\delta)}{n}}$$

$$+ \frac{4C_h^{\mathbf{s}} C_h^{\mathbf{a}} h^2 G^2 \log(\mathcal{N}_\infty(\varepsilon,\mathcal{G})|\mathcal{W}||\mathcal{F}|/\delta)}{3n} + 16C_h^{\mathbf{s}} C_h^{\mathbf{a}} h^2 G^2 \varepsilon$$

Using the AM-GM inequality with $\varepsilon = O(1/C_h^{\mathbf{s}} C_h^{\mathbf{a}} h^2 G^2 n)$ gives the result. $\qquad\square$

# G  Optimization Tools

**Definition G.1** (Gradient mapping)**.**

$$G^\eta(x, g) := \frac{1}{\eta}\left(x - \mathrm{Proj}_\mathcal{X}(x + \eta g)\right) \qquad (27)$$

**Lemma G.1** (Stationary convergence of PGD)**.** *Suppose $f : \mathcal{X} \to \mathbb{R}$ is $\beta$-smooth over $\mathcal{X}$, a nonempty closed and convex set, and that we have access to a gradient oracle such that $\mathbb{E}[g(x)|x] = \nabla f(x)$ and $\mathbb{E}[\|g(x) - \nabla f(x)\|^2|x] \leq \varepsilon^2$. Then if $\eta = 1/\beta$, we have*

$$\frac{1}{T}\sum_t \mathbb{E}\left[\|G^\eta(x^{(t)}, \nabla f(x^{(t)}))\|^2\right] \leq \frac{4\beta(f_0 - f^*)}{T} + 6\varepsilon^2$$

*Proof.* For any $x$, define $x^+ = \mathrm{Proj}_\mathcal{X}(x - \eta g(x))$. Since $x^+ = \mathrm{prox}_{\cdot, I_\mathcal{X}}(x - \eta g(x))$, where $I_\mathcal{X}$ is the indicator function for the set $\mathcal{X}$, from [Lem. G.2]() we have

$$\langle x - \eta g(x) - x^+, x - x^+ \rangle \leq 0.$$

Rearranging, this implies

$$\langle g(x), x^+ - x \rangle + \frac{1}{\eta}\|x - x^+\|^2 \le 0.$$

Next, since $f$ is $\beta$-smooth, for any $x$ and $x^+$ we have

$$f(x^+) \le f(x) + \langle \nabla f(x), x^+ - x \rangle + \frac{\beta}{2}\left\|x - x^+\right\|^2$$

$$= f(x) + \langle g(x), x^+ - x \rangle + \langle \nabla f(x) - g(x), x^+ - x \rangle + \frac{\beta}{2}\left\|x - x^+\right\|^2$$

$$\le f(x) + \eta \langle g(x) - \nabla f(x), G^\eta(x, g(x)) \rangle + \left(\frac{\eta^2 \beta}{2} - \eta\right)\|G^\eta(x, g(x))\|^2$$

$$= f(x) + \eta \langle g(x) - \nabla f(x), G^\eta(x, \nabla f(x)) \rangle + \eta \langle g(x) - \nabla f(x), G^\eta(x, g(x)) - G^\eta(x, \nabla f(x)) \rangle$$

$$\quad + \left(\frac{\eta^2 \beta}{2} - \eta\right)\|G^\eta(x, g(x))\|^2$$

where we substitute the definition of $G^\eta(x, g(x)) = \frac{1}{\eta}(x - x^+)$ in the the second to last line. Notice that

$$\langle g(x) - \nabla f(x), G^\eta(x, g(x)) - G^\eta(x, \nabla f(x)) \rangle \le \|g(x) - \nabla f(x)\|\|G^\eta(x, g(x)) - G^\eta(x, \nabla f(x))\|$$

$$\le \|g(x) - \nabla f(x)\|^2$$

from the non-expansion of the projection operator. Then we have

$$f(x^+) \le f(x) + \eta \langle g(x) - \nabla f(x), G^\eta(x, \nabla f(x)) \rangle + \eta\|g(x) - \nabla f(x)\|^2 + \left(\frac{\eta^2 \beta}{2} - \eta\right)\|G^\eta(x, g(x))\|^2$$

Next, we take the expectation of both sides conditioned on $x$.

$$\mathbb{E}[f(x^+)|x] \le f(x) + \eta \langle \mathbb{E}[g(x)|x] - \nabla f(x), G^\eta(x, \nabla f(x)) \rangle$$

$$\quad + \eta\mathbb{E}[\|g(x) - \nabla f(x)\|^2|x] + \left(\frac{\eta^2 \beta}{2} - \eta\right)\mathbb{E}[\|G^\eta(x, g(x))\|^2|x]$$

$$\le f(x) + \eta\varepsilon^2 + \left(\frac{\eta^2 \beta}{2} - \eta\right)\mathbb{E}[\|G^\eta(x, g(x))\|^2|x]$$

Then unrolling the recursion through iterations and substituting $\eta = 1/\beta$, we have

$$\frac{1}{T}\sum_t \mathbb{E}[\|G^\eta(x^{(t)}, g(x^{(t)}))\|^2] \le \frac{2\beta(f(x^{(0)}) - f(x^{(T)}))}{T} + 2\varepsilon^2 \le \frac{2\beta(f(x^{(0)}) - f(x^*))}{T} + 2\varepsilon^2$$

if $f$ is nonnegative. Lastly,

$$\frac{1}{T}\sum_t \mathbb{E}[\|G^\eta(x^{(t)}, \nabla f(x^{(t)}))\|^2] = \frac{1}{T}\sum_t \mathbb{E}[\|G^\eta(x^{(t)}, g(x^{(t)})) - G^\eta(x^{(t)}, g(x^{(t)})) + G^\eta(x^{(t)}, \nabla f(x^{(t)}))\|^2]$$

$$\le \frac{2}{T}\sum_t \mathbb{E}[\|G^\eta(x^{(t)}, g(x^{(t)}))\|^2] + \frac{2}{T}\sum_t \mathbb{E}[\|G^\eta(x^{(t)}, g(x^{(t)})) - G^\eta(x^{(t)}, \nabla f(x^{(t)}))\|^2]$$

$$\le \frac{4\beta(f(x^{(0)}) - f(x^*))}{T} + 4\varepsilon^2 + \frac{2}{T}\sum_t \mathbb{E}[\|g(x^{(t)}) - \nabla f(x^{(t)})\|^2]$$

$$\le \frac{4\beta(f(x^{(0)}) - f(x^*))}{T} + 6\varepsilon^2$$

$$\square$$

**Lemma G.2** (Theorem 6.39 from [Bec17]). *Let $g : \mathcal{E} \to (\infty, \infty]$ be a proper closed and convex function. Then for any $x, y \in \mathcal{E}$, the following three claims are equivalent:*

  *1. $y = \text{prox}_g(x)$*

2. $x - y \in \partial g(u)$

3. $\langle x - y, u - y \rangle \le g(u) - g(y)$ *for any* $u \in \mathcal{E}$

**Lemma G.3.** *Suppose $f$ is $M$-gradient dominated and $\beta$-smooth, and*

$$\frac{1}{T} \sum_t \left\| G^\eta(x^{(t)}, \nabla f(x^{(t)})) \right\|^2 \le \varepsilon^2,$$

*where $G^\eta(x, g)$ is the gradient mapping defined in Def. G.1. Also, suppose $\|x - x'\|_2 \le r$ for all $x, x' \in \mathcal{X}$. Then*

$$\min_{t \in [T]} \left\{ f(x^*) - f(x^{(t)}) \right\} \le rM(\eta\beta + 1)\varepsilon. \tag{28}$$

*Proof of Lemma G.3.* If $f$ is gradient dominated, for any $t \in [T]$ we have

$$f(x^*) - f(x^{(t)}) \le M \max_{x' \in \mathcal{X}} \left\langle \nabla f(x^{(t)}), x' - x^{(t)} \right\rangle.$$

Applying Lemma G.4 with $-f$, we have

$$\nabla f(x^{(t)}) \in N_{\mathcal{X}}(x^{(t)}) + \mathcal{B}\left( (\eta\beta + 1)\|G^\eta(x^{(t-1)}, \nabla f(x^{(t-1)}))\| \right)$$

From the definition of the normal cone, we have $\langle v, x' - x^{(t)} \rangle \le 0$ for any $v \in N_{\mathcal{X}}(x^{(t)})$ and $x' \in \mathcal{X}$. Then for any $x' \in \mathcal{X}$,

$$\left\langle \nabla f(x^{(t)}), x' - x^{(t)} \right\rangle \le (\eta\beta + 1) \left\| G^\eta(x^{(t-1)}, \nabla f(x^{(t-1)})) \right\| \|x - x^{(t)}\| \le (\eta\beta + 1)r \left\| G^\eta(x^{(t-1)}, \nabla f(x^{(t-1)})) \right\|$$

Combining the above inequalities,

$$\min_{t \in [T]} \left\{ f(x^*) - f(x^{(t)}) \right\} \le (\eta\beta + 1)Mr \min_{t \in [T]} \|G^\eta(x^{(t-1)}, \nabla f(x^{(t-1)}))\| \le (\eta\beta + 1)Mr\varepsilon.$$

$\square$

**Lemma G.4** (Lemma 3 from [GL16]). *Let $f : \mathbb{R}^d \to (-\infty, \infty)$ be be $\beta$-smooth over a convex set $\mathcal{X}$ For any $t \in [T]$, consider $x^{(t+1)} = \mathrm{Proj}_{\mathcal{X}} \left( x^{(t)} - \eta \nabla f(x^{(t)}) \right)$. Then*

$$-\nabla f(x^{(t+1)}) \in N_{\mathcal{X}}(x^{(t+1)}) + \mathcal{B}\left( (\eta\beta + 1)\|G^\eta(x^{(t)}, -\nabla f(x^{(t)})\| \right),$$

*where $N_{\mathcal{X}}$ is the normal cone of $\mathcal{X}$ and $\mathcal{B}(r) = \{x \in \mathbb{R}^d : \|x\|_2 \le r\}$.*

*Proof of Lemma G.4.* Projected gradient descent can be equivalently written as [Bec17]

$$\mathrm{Proj}_{\mathcal{X}} \left( x^{(t)} - \eta \nabla f(x^{(t)}) \right) = \operatorname*{argmin}_{x \in \mathbb{R}^d} \left[ f(x^{(t)}) + \left\langle \nabla f(x^{(t)}), x - x^{(t)} \right\rangle + \frac{1}{2\eta} \|x - x^{(t)}\|_2^2 + I_{\mathcal{X}}(x) \right],$$

where $I_{\mathcal{X}}(x) = 0$ if $x \in \mathcal{X}$, and $+\infty$ otherwise, is the indicator function for $\mathcal{X}$. Then by the subgradient optimality condition, we have

$$0 \in \nabla f(x^{(t)}) + \tfrac{1}{\eta}(x^{(t+1)} - x^{(t)}) + N_{\mathcal{X}}(x^{(t+1)})$$

With some rearrangement, this implies that

$$-\nabla f(x^{(t+1)}) \in N_{\mathcal{X}}(x^{(t+1)}) + \nabla f(x^{(t)}) - \nabla f(x^{(t+1)}) + \tfrac{1}{\eta}(x^{(t+1)} - x^{(t)})$$

which implies the lemma statement since

$$\|\nabla f(x^{(t)}) - \nabla f(x^{(t+1)}) + \tfrac{1}{\eta}(x^{(t+1)} - x^{(t)})\| \le \beta\|x^{(t)} - x^{(t+1)}\| + \tfrac{1}{\eta}\|x^{(t)} - x^{(t+1)}\|$$
$$\le (\eta\beta + 1)\|G^\eta(x^{(t)}, \nabla f(x^{(t)}))\|$$

using the $\beta$-smoothness of $f$ in the first inequality, and Definition G.1 in the second. $\square$

**Lemma G.5.** *Suppose $f$ is $\beta$-smooth and that at each iteration $t$, we have $g^{(t)}$ from a gradient oracle such that $\mathbb{E}\left[g^{(t)}|x^{(t)}\right] = \nabla f(x^{(t)})$ and $\mathbb{E}\left[\|\nabla f(x^{(t)}) - g^{(t)}\|^2|x^{(t)}\right] \leq \varepsilon^2$ for all $t \in [T]$. Then gradient ascent using $\{g^{(t)}\}$ satisfies*

$$\frac{1}{T}\sum_{t=1}^{T}\mathbb{E}\left[\left\|\nabla f(x^{(t)})\right\|^2\right] \leq \frac{2\beta(f_0 - f^*)}{T} + \varepsilon^2. \tag{29}$$

*Proof of Lem. G.5.* From the $\beta$-smoothness of $f$,

$$f(x^{(t+1)}) \leq f(x^{(t)}) + \left\langle\nabla f(x^{(t)}), x^{(t+1)} - x^{(t)}\right\rangle + \frac{\beta}{2}\|x^{(t+1)} - x^{(t)}\|^2$$

$$= f(x^{(t)}) - \eta\left\langle\nabla f(x^{(t)}), g^{(t)}\right\rangle + \frac{\beta\eta^2}{2}\|g^{(t)}\|^2$$

$$= f(x^{(t)}) - \eta\left\langle\nabla f(x^{(t)}), \nabla f(x^{(t)}) - \nabla f(x^{(t)}) + g^{(t)}\right\rangle + \frac{\beta\eta^2}{2}\|\nabla f(x^{(t)}) - \nabla f(x^{(t)}) + g^{(t)}\|^2$$

$$= f(x^{(t)}) + \left(\frac{\beta\eta^2}{2} - \eta\right)\|\nabla f(x^{(t)})\|^2 + \left(\beta\eta^2 - \eta\right)\left\langle\nabla f(x^{(t)}), g^{(t)} - \nabla f(x^{(t)})\right\rangle + \frac{\beta\eta^2}{2}\|\nabla f(x^{(t)}) - g^{(t)}\|^2$$

Taking the expectations of both sides conditioned on $x^{(t)}$ (prior histories), we have

$$\mathbb{E}[f(x^{(t+1)})|x^{(t)}] \leq f(x^{(t)}) + \left(\frac{\beta\eta^2}{2} - \eta\right)\|\nabla f(x^{(t)})\|^2 + \frac{\beta\eta^2\varepsilon^2}{2}$$

Substituting $\eta = 1/\beta$, unrolling through iterations, and using the law of total expectation gives the result. $\square$

