# OpenReview forum: "Occupancy-based Policy Gradient: Estimation, Convergence, and Optimality"
_NeurIPS.cc/2024/Conference — NeurIPS 2024 poster_

### Official Review · Reviewer_n45j · 2024-07-12

**Soundness:** 3
**Presentation:** 3
**Contribution:** 2
**Rating:** 6
**Confidence:** 2

**Summary:**

With a focus on "occupancy-based" methods for RL, the authors propose a
model-free, policy gradient method for policy optimization in finite-horizon
MDPs via occupancy estimation. Guided by the representation of return in terms
of the state-visitation probability and reward, they express the gradient of
the return in terms of gradient of the log of the state-visitation probability.
They propose to estimate the latter term via squared loss regression (see
eq 2). This estimate is then used to approximate the gradient of the return
which is then plugged in to the usual policy gradient objective. In the online
RL setting, they provide gradient estimation guarantees and convergence to a
near-optimal policy. For offline RL, due to insufficient coverage of the
offline dataset, they provide guarantees on the estimate of the gradient of the
clipped return.

**Update**: I have revised my score based on the following considerations:
1. The authors satisfactory response to the rebuttal,
2. The unique approach to policy gradients, which may be insightful in advancing research on policy gradient methods,
3. While rigidity of their assumption in the offline setting may limit its broader applicability, it also leaves room for further improvement.
4. The authors willingness to address the other reviewers suggestions on notation and presentation.

**Strengths:**

1. To the best of my knowledge, the authors propose a novel perspective to policy gradients in online and offline RL based on computing the return gradient by first estimating the gradient of occupancy measures.

2. Their method for the online setting is technically sound with local and global convergence claims supported by theoretical analysis and relevant assumptions.

**Weaknesses:**

For the offline setting, the authors introduce new assumptions in the appendix which appear to influence their main result in Theorem 16. Precisely, in the proof of Theorem 16, the authors make use of Lemmas which refer to assumption 7 on page 37 and assumption 8 on page 38 to respectively control the MLE estimation error and density ratio estimation error. As such the paper is not self-contained and the results are rather difficult to verify.

**Questions:**

Just some additional comments:

1. Line 70: should be $h'\geq h$.
2. In Line 151, do you mean $\psi^{\pi}$ instead of $\psi$? Also, the first expression $d^{\pi}=\mu(s)^{T}\psi$ seems inconsistent.

**Limitations:**

None. This is a purely theory paper.

---

> ### Author Rebuttal · Authors · 2024-08-07
>
> Thank you very much for your feedback.
>
> ---
>
> **re: "Introduce new assumptions in the appendix which appear to influence their main results"**
>
> The omission of Asm. 7 and 8 from the preconditions of Theorem 16 was an oversight, rather than a deliberate choice to obscure dependencies. We will include Asm. 7 and 8 in the preconditions of Theorem 16 for the next revision, and space allowing, will move the definitions of Asm. 7 and 8 (and surrounding discussions) into the main text.
>
> We made the difficult decision of not including Asm. 7 and 8 (or Algs. 4 and 5) in the main text due to space constraints, given the breadth of our results in both offline and online policy gradient. Our reasoning was that
> - Alg. 4 (requiring Asm. 7) and Alg. 5 (requiring Asm. 8) are algorithms from established papers and have been analyzed therein.
> - They are orthogonal to the novelty of our paper, which uses algorithms from the aforementioned works as subroutines.
> - Most importantly, we wanted to highlight the distinguishing parts of our analysis given limited space.
>
> In addition, they are weaker in some sense than Asm. 5 (expanded on below), which we mentioned briefly in L310, “We focus on discussing Asm. 5 for the offline gradient function class, which requires a stronger level of expressiveness.” Given extra space, we will expand this discussion based on the following points sourced from the Appendix:
>
> > Assumption 8
>
> From L1002-1005: “The weight function class completeness assumption is shown Asm. 8, and is satisfied in low-rank MDPs using linear-over-linear function classes that have pseudo-dimension bounded by MDP rank. It can be seen as a 1-dimensional version of Asm. 5 where $\rho = 1$, and in that sense strictly weaker.”
>
> >Assumption 7
>
> This is the standard realizability guarantee for maximum likelihood estimation or supervised learning (it is analogous to [Theorem 21, AKKS20] and also required by [HCJ23]), and only requires that the function class includes the ground-truth function. In comparison, Asm. 5 and 8 involve multiple functions.
>
> ---
>
> **re: "Paper is not self-contained and the results are rather difficult to verify"**
>
> Aside from the above issue (for which our previous response proposes a self-contained fix), we believe that the paper is self-contained and verifiable. For all algorithms and results in the appendix we have included rigorous proofs, and assumptions with justification. However, we empathize with the sentiment of your comment in the sense that our results rely on algorithms from established papers as subroutines, which results in layers of analysis. With more space, we hope to provide more details and results on this in the main body, per the existing descriptions, guarantees, and proofs in Appendices D and E.
>
> ---
>
> **Questions**
>
> Yes, it should be $d^\pi = \mu(s)^\top \psi^\pi$. Thank you for catching these typos. We will correct them in our paper.
>
> ---
>
> **References**
>
> [AKKS20] Alekh Agarwal, Sham Kakade, Akshay Krishnamurthy, and Wen Sun. “Flambe: Structural complexity and representation learning of low rank mdps”. In: Advances in Neural Information Processing Systems (2020).
>
> [HCJ23] Audrey Huang, Jinglin Chen, and Nan Jiang. "Reinforcement learning in low-rank mdps with density features." International Conference on Machine Learning. PMLR, 2023.

---

> > ### Comment · Reviewer_n45j · 2024-08-08
> >
> > I thank the authors for their response. I am satisfied with their clarification of the Assumptions and willingness to move them (including Assumption 6) to the main text.
> >
> > I have also read other review responses.

---

### Official Review · Reviewer_K5ET · 2024-07-12

**Soundness:** 4
**Presentation:** 3
**Contribution:** 4
**Rating:** 7
**Confidence:** 3

**Summary:**

This paper introduces policy gradient algorithms that focus on
estimating the gradient of the *occupancy measure* with respect to the
policy parameters. In this framework, one can easily extend convergence
analysis beyond the standard RL objective (e.g., maximization of
expected return) to a much broader class of occupancy objectives, such
as imitation learning or pure exploration. In addition to an online
algorithm, the authors design an algorithm for the offline setting based
on pessimistic corrections to density ratios, and provide bounds on the
policy gradient error for both under function approximation. In the case
of the offline algorithm, this paper demonstrates convergence bounds
under weaker assumptions than existing methods. In the case of the
online algorithm, the paper further provides conditions under which
their policy gradient algorithm achieves global optimality.

**Strengths:**

This paper is very interesting and extemely thorough. I really like the idea of
estimating the occupancy gradient in order to generalize over objectives for
sequential decision-making; these seems quite powerful, and has the potential of
being very impactful. Effectively, here the authors have provided a
meta-algorithm (with convergence bounds) for a much broader class of algorithms
than standard policy gradient methods, which can unify several fields of research.
The Offline OccuPG method was really neat, and makes great
use of the existing FORC method for estimating clipped density ratios.

**Weaknesses:**

The biggest weakness for the paper, in my opinion, is that it could
benefit from extra clarification at many points. Realistically, this was
probably omitted in favor of making room for more content (the paper is
very dense), but I suggest the authors add some guiding discussions.

Particularly, I think the following should be clarified:

- **Motivation**: My first instinct after having read the draft was
  "this was very interesting to read, but how is the field of RL better
  off in light of these results?". The conclusion and introductory
  sections place a lot of emphasis on the fact that this paper presents
  the first algorithms/convergence results based on estimating the
  occupancy measure, but it's not immediately clear why this is so
  impactful. Ultimately, my impression is that the following two points
  (especially the first) are the real winners:
  1.  The ability to optimize a general class of functionals;
  2.  Weaker assumptions for convergence with the offline algorithm.

  I believe emphasizing these points more, and motivating them, can
  really help the reader appreciate the results, but I found these
  points to get a little lost in the sea of math.
- **Comparison to existing work**: While I understand that this paper is
  the first to analyze convergence of PG methods based on occupancy
  measures, the paper focuses largely on the application of optimizing
  expected return. As such, as someone that is not intimately familiar
  with convergence results for PG methods, it would have been helpful to
  have a more concrete comparison between the bounds presented in this
  work and existing ones.
- **Misc. writing/clarity issues**: Some math and/or algorithmic details
  were ambiguous and/or unclear, listed below. This made the logic a
  little difficult to follow at times.

## Minor issues

On line 28, "In answer" -\> "In response".

In Algorithm 1, $\mathcal{D}^{\mathrm{grad}}$ is not properly defined. On
line 2 of the algorithm, it is implied that it is seeded the same way as
$\mathcal{D}^{\rm reg}$, but on line 7 it appears that
$\mathcal{D}^{\rm grad}$ is supposed to contain reward data (unlike
$\mathcal{D}^{\rm reg}$).

In Algorithm 1, I think the definition of $\mathcal{D}^{\mathrm{reg}}$ has a
typo. It says $\mathcal{D}_h^{\mathrm{reg}} = \{(s\_h, a\_h,
s\_{h+1})\}\_{i=1}^n$, but the index $i$ does not appear anywhere. My
guess is that it should really be something like
$\mathcal{D}\_h^{\mathrm{reg}} = \{(s\_h^{(i)}, a\_h^{(i)},
s\_{h+1}^{(i)})\}\_{i=1}^n$, where e.g. $\{s^{(i)}\_h\}$ is the sequence of
states encounted over the course of rollout $i$.

In Theorem 2, $\mathsf{pd}\_{\mathcal{G}}$ looks confusing (not clear
that it is $\mathsf{p}$ times $\mathsf{d}\_{\mathcal{G}}$ as opposed to a
variable called $\mathsf{pd}\_{\mathcal{G}}$). Is it necessary to use
sans serif font for $\mathsf{p}$?

In definition 46, I believe there is a notational issue. It says
$\mathcal{F}\subset\mathbb{R}^{\mathcal{X}}$, but then you have a term
$\mathrm{sign}(f(x_1^n - c))$ for $f\in\mathcal{F}$. But $x_1^n -
c\not\in\mathcal{X}$. Are you actually mapping $f$ over the dimensions
of $x_1^n - c$? Also, there is a typo here, $c$ should be $v$.

**Questions:**

In Corollary 6, what does it mean to run OccuPG with initial
distribution $\nu$?

On line 238, you defined $\bar{\pi}_h = (\pi\land
C^\mathbf{a}_h\pi^D_h)$. I am having some trouble interpreting this.
Firstly, should it not be $(\pi_h\land C^\mathbf{a}_h\pi^D_h)$? Then, if
that's the case, is this equivalent to

$$\begin{align*}
\bar{\pi}_h(a\mid s) = \min\{\pi_h(a\mid s), C^a_h\pi^D_h(a\mid s)\}?
\end{align*}$$

Given that the text focuses almost entirely on the problem of maximizing
expected return, how do the bounds given in this paper compare to those
of existing results? Is there any benefit to running OccuPG if you only
care about expected return (especially in the online setting)?

How tight are your convergence bounds, particularly in the case of
optimizing general functionals?

In Lemma 5, when should one expect to have $\mathcal{C}^{\pi^*}<\infty$?

**Limitations:**

No major issues in this regard. Two limitations come to mind that were not
thoroughly addressed:
1. For optimization of general functionals, convergence results depend on a
   Lipschitz assumption on the functional that is only discussed on the
   appendix. How generally is this assumption satisfied?
2. For the global convergence result (Lemma 5), there is a very strict
   assumption on the initial state distribution, which would not often be
   satisfied in practice (to my understanding). That said, other theoretical
   works have made similar assumptions.

---

> ### Author Rebuttal · Authors · 2024-08-07
>
> Thank you for your detailed and insightful feedback, and for your appreciation of our work. We will refine our presentation per your comments, and include additional discussions according to the points below.
>
> ---
>
> **Motivation**
>
> Yes, these are the main contributions of the online and offline algorithms. We will make them more prominent in our revision.
>
> ---
>
> **Comparison to previous work**
>
> For both the online and offline settings, our results can be divided into two parts, estimation and optimization. Once the estimation and gradient domination conditions (e.g., Eq.(4) for online) are established, the optimization analyses largely follow from existing techniques in the literature.
>
>
> Overall, the dependencies on sample size $n$, Lipschitz constant $\beta$, and iterations $T$ match what we expect to see based on previous work and the optimization literature [Bec17, KU20, AKLM21, NZJZW22, BR24]. For coverage coefficients, the online version (Definition 3) is analogous to those in existing work (ours is finite-horizon while [AKLM21] is infinite-horizon). For offline gradient estimation, the results in [NZJZW22] pay for coverage of all policies (Assumption 6.1) while ours is single-policy (Theorem 16).
>
> Beyond that, our setting of occupancy-based gradient estimation is novel, so a direct comparison for the gradient estimation bounds is not readily available. In addition, few previous works provide end-to-end analysis incorporating both estimation/statistical error and optimization error (e.g., [AKLM21] establish global convergence with true gradients, while [NZJZW22] estimate off-policy gradients but do not use them for policy gradient).
>
> ---
>
> **Questions**
>
> > In Corollary 6, what does it mean to run OccuPG with initial distribution $\nu$?
>
> This means that the initial state is drawn from $\nu$, i.e., $s_1 \sim \nu$. The rest of the trajectory is generated by rolling out the policy in the true MDP.  We will revise our paper to make this more explicit.
>
> > Definition of $\bar\pi_h$ in L238
>
> We believe our use of the $h$ subscript might have caused some confusion (discussed below), but overall yes. By $\bar\pi_h  = \left(\pi \wedge C_h^{\mathsf{a}}\pi^D_h \right)$ we do mean $\bar\pi_h(a_h|s_h) = \min \left\lbrace \pi(a_h|s_h), C_h^{\mathsf{a}} \pi^D_h(a_h|s_h) \right\rbrace$, and we will make this more explicit in L238.
>
> With regards to $\pi(a_h|s_h)$ vs. $\pi_h(a_h|s_h)$, for notational compactness we assumed that the same state cannot appear in multiple timesteps (L60-62). So $\pi(a_h|s_h)$ only refers to the policy’s action at timestep $h$, and is de facto equivalent to $\pi_h(a_h|s_h)$.
>
> In the general offline dataset from Def. 7, where each $h$ has a different dataset, $\pi^D_h$ was intended to indicate the data collection policy for timestep $h$. This might have caused some confusion, and we will clarify our notation accordingly.
>
> > Benefit to running OccuPG for expected return (esp. online)?
>
> In the online setting with expected return, we view OccuPG as complementary to value-based PG, in the sense that the former can be beneficial when occupancy modeling is more feasible or aligned with existing inductive biases (e.g., on occupancy representations). However, it is possible that calculating $\nabla\log d^\pi$ is more challenging than auto-differentiating through the value-based policy gradient. This is something that we are interested in exploring experimentally.
>
> More importantly, OccuPG serves as the conceptual framework for our offline PG algorithm Off-OccuPG, and here we do see improvements in sample complexity because, unlike previous works [LSAB19, NZJZW22], we get away with single-policy coverage in gradient estimation.
>
> > Tightness of convergence bounds, particularly in the case of optimizing general functionals
>
> This is an interesting question, and currently we are not sure. For occupancy gradient estimation, we do not expect that the rate of $n^{-½}$ can be improved, and our dependence in $\mathsf{p}$ matches similar works on (off-policy) gradient estimation [NZJZW22]. It’s possible that $H$ factors can be reduced by more sophisticated handling of error compounding.
>
> For general functionals, the generality of our bound (obtained through the Lipschitz factor, which doesn’t take into account properties of the objective such as curvature) means that it is likely not tight.
>
> > In Lemma 5, when should one expect to have $\mathcal{C}^{\pi^*} < \infty$?
>
> $\mathcal{C}^{\pi^*} < \infty$ if we have access to some distribution over states $\nu \in \Delta(\mathcal{S})$, such that $\nu(s) \gg \sum_h d^{\pi^*}(s)$ for all $s$. In other words, it places nonzero mass on all states visited by the optimal policy.
>
> At the worst, one can always guarantee $\mathcal{C}^{\pi^*}$ is finite (though potentially very large) by setting $\nu = \mathrm{uniform}(\mathcal{S})$ to be uniform over all states. In practice, one may be able to craft a more refined $\nu$ that results in smaller $\mathcal{C}^{\pi^*}$ given some expert or domain knowledge.
>
> ---
>
> **References**
>
> [AKLM21] Alekh Agarwal, Sham M Kakade, Jason D Lee, and Gaurav Mahajan. “On the theory of policy gradient methods: Optimality, approximation, and distribution shift”. In: The Journal of Machine Learning Research 22.1 (2021)
>
> [Bec17] Amir Beck. First-order methods in optimization. SIAM, 2017.
>
> [BR24] Jalaj Bhandari and Daniel Russo. “Global optimality guarantees for policy gradient methods”. In: Operations Research (2024)
>
> [LSAB19] Yao Liu, Adith Swaminathan, Alekh Agarwal, and Emma Brunskill. “Off-policy policy gradient with state distribution correction”
>
> [KU20] Nathan Kallus and Masatoshi Uehara. “Statistically efficient off-policy policy gradients”. In: International Conference on Machine Learning. PMLR. 2020
>
> [NZJZW22] Chengzhuo Ni, Ruiqi Zhang, Xiang Ji, Xuezhou Zhang, and Mengdi Wang. “Optimal Estimation of Policy Gradient via Double Fitted Iteration”. In: International Conference on Machine Learning. PMLR. 2022

---

> > ### Comment · Reviewer_K5ET · 2024-08-08
> >
> > Thanks for the authors for the great and detailed response.
> >
> > Upon reading the rebuttal and the other reviews, I maintain my view that this is a really nice paper that should be accepted for publication.

---

### Official Review · Reviewer_2KwR · 2024-07-17

**Soundness:** 3
**Presentation:** 3
**Contribution:** 2
**Rating:** 3
**Confidence:** 2

**Summary:**

This paper investigates the convergence of model-free policy gradient (PG) methods that compute the gradient through occupancy.

**Strengths:**

The convergence analysis and the theorem appear to be sound.

**Weaknesses:**

In my opinion, the paper makes overclaims. The convergence depends on a coverage coefficient, which removes the need for exploration. Therefore, the obtained bound does not indicate the number of samples needed for exploration when the initial policy is not good. Additionally, the proposed algorithm does not address the problem of exploration.

**Questions:**

See the 'Weakness' part.

**Limitations:**

See the 'Weakness' part.

---

> ### Author Rebuttal · Authors · 2024-08-07
>
> Thank you very much for your comments. However, we respectfully disagree that our paper over-claims its results. The reviewer’s comment that “the coverage coefficient… removes the need of exploration” likely results from confusion over the term “exploration”, whose meaning in the PG literature is different from that in PAC-RL. Our use of the term is aligned with the literature, and our results’ dependence on the coverage coefficient is also standard. Results of a similar nature are commonly found in the literature, and we elaborate on the details below.
>
>
> ---
>
>
> **re: "The convergence depends on a coverage coefficient, which removes the need for exploration"**
>
> In (online) policy gradient analysis, the coverage coefficient $\mathcal{C}^{\pi^*}$ (Lemma 5) expresses the difficulty of the exploration problem in policy optimization. L184-188 discusses this in-depth.
>
> There may be some confusion over terminology because our algorithm, along with routine PG methods such as REINFORCE and PPO, do not actively explore in the sense of PAC exploration (e.g., UCB and other optimism-in-face-of-uncertainty algorithms). Rather, exploration here refers to the difficulty of finding a good policy within the policy class from online interactions with the environment.
>
> This usage of the phrase “exploration” matches foundational works in the PG literature, for which $\mathcal{C}^{\pi^*}$ is also the standard notion of online PG complexity. They are identical to Definition 3.1 from the seminal work of [AKLM21], and the paragraph above it (modulo differences in infinite and finite horizon MDPs). Definition 3 of [BR24] is another point of comparison. All of these coefficients depend on how well an initial state distribution covers the optimal policy’s occupancy.
>
> That said, we should have defined “exploration” more explicitly in the abstract/introduction to reduce confusion, and will revise accordingly. This is a simple fix that can be lifted from L184-188, and that is aligned with standards and terminology from existing work.
>
> ---
>
> **re: "the obtained bound does not indicate the number of samples needed for exploration when the initial policy is not good"**
>
> First of all, the coverage coefficient is defined with respect to a _distribution over initial/starting states_. This is _not an initial policy, or even a policy_.
>
> Moreover, a good coverage coefficient does not necessarily imply a good initial policy; the initial policy can still have poor performance, and we rely on gradient ascent to find a better policy through online interactions. This is what the PG literature means by “exploration”.
>
> ---
>
> **re: "Additionally, the proposed algorithm does not address the problem of exploration"**
>
> As argued above, the coverage coefficient in our online bounds characterizes the difficulty of exploration faced by policy gradient optimization algorithms, on par with the existing literature.
>
> However, developing algorithms that actively explore to improve policy optimization is a valuable direction of future work, given that in practice the initial state distribution may not be very exploratory (aka have poor coverage over $d^{\pi^*}$ per Lemma 5).
>
> ---
>
> **Coverage coefficient in offline setting**
>
> Lastly, our results cover both online and offline PG, and the latter makes up half of the paper (pages 6-9). In the offline learning setting, the learner cannot interact with the environment and must learn from the given data, hence _the problem of exploration simply does not exist there_. As with all offline results, a coverage coefficient is expected in the guarantee, reflecting the quality of the offline data (since, if the data is poor, there is nothing one can do). In fact, the kind of coverage coefficient we use is already improved over previous _offline_ PG results; previous results require all-policy coverage even just for estimating the gradient [LSAB19, NZJZW22], whereas our estimation guarantee only depends on single-policy coverage (Theorem 16).
>
> ---
> **References**
>
> [AKLM21] Alekh Agarwal, Sham M Kakade, Jason D Lee, and Gaurav Mahajan. “On the theory of policy gradient methods: Optimality, approximation, and distribution shift”. In: The Journal of Machine Learning Research 22.1 (2021), pp. 4431–4506.
>
> [BR24] Jalaj Bhandari and Daniel Russo. “Global optimality guarantees for policy gradient methods”. In: Operations Research (2024)
>
> [LSAB19] Yao Liu, Adith Swaminathan, Alekh Agarwal, and Emma Brunskill. “Off-policy policy gradient with state distribution correction”. In: arXiv preprint arXiv:1904.08473 (2019).
>
> [NZJZW22] Chengzhuo Ni, Ruiqi Zhang, Xiang Ji, Xuezhou Zhang, and Mengdi Wang. “Optimal Estimation of Policy Gradient via Double Fitted Iteration”. In: International Conference on Machine Learning. PMLR. 2022, pp. 16724–16783.

---

### Decision · Program_Chairs · 2024-09-25

**Decision:**

Accept (poster)

**Comment:**

In this paper on policy gradient for finite-horizon MDPs with general function approximation, the authors propose a novel approach based on the estimation of the gradients of occupancy measures.
Under (variants of) standard assumptions, they establish iteration and sample complexity upper bounds for finding stationary points and optimal policies. They also consider nonlinear performance functionals and the offline learning setting.

The reviewers agree on the originality and the soundness of the proposed approach. Indeed, although occupancy measures have been employed before in the PG literature (as discussed in appendix A of the paper), the perspective presented here is new and leads to a satisfying compromise between sample and computational complexity.

The reviewers also raised some important issues, especially regarding presentation, that were mostly solved in the discussion phase.
Indeed, the paper is very dense and some parts need polishing.
I stress the importance of incorporating the fixes and clarifications in the revised manuscript. I will summarize the most important ones below, but solving the several minor issues pointed out by the reviewers is also important to improve overall clarity:

- Add a comparison with existing results from the PG literature, as suggested by Reviewer K5ET. Regarding this, I suggest to include the following simple but important observation: Alg. 1 needs a total of O(\epsilon^{-4}) samples to find an approximate stationary point with (not squared!) gradient norm smaller than epsilon, the same as REINFORCE (Yuan et al. 2022, "A general sample complexity analysis of vanilla policy gradient").

- Expand on the motivation of using an occupancy-based approach compared to more standard PG methods, as suggested by Reviewer K5ET

- Use the extra space to make the theory more self-contained, as suggested by Reviewer n45j

Additionally, as pointed out by Reviewer n45j in the discussion phase, the low-rank MDP example is a bit misleading. In low-rank MDPs, one typically assumes to know state-action features, while $\mu(s')$ is unknown. The alternative setting proposed here should be motivated better, or the authors should acknowledge that the completeness assumption may be a strong one.

I also suggest to add a few words on possible infinite-horizon extensions.

Regarding the concern of Reviewer 2KwR on concentrability coefficients, I agree with the authors that it is misdirected, as it applies to a significant part of the PG theory literature. However, I suggest to add a paragraph on the meaning of "exploration" to avoid further misunderstandings, as proposed in the authors' response.

Overall, I think that this paper would be interesting for the community and will be ready for publication after implementing the fixes suggested by the reviewers and summarized here, which I think can be reasonably done for the camera ready version.